# Multitemporal characterisation of a proglacial system: a multidisciplinary approach.

Elisabetta Corte[1], Andrea Ajmar[2], Carlo Camporeale[1], Alberto Cina[1], Velio Coviello[3,5], Fabio Giulio Tonolo[4], Alberto Godio[1], Myrta Maria Macelloni[1], Stefania Tamea[1], and Andrea Vergnano[1]

[1]Department of Environment, Land and Infrastructure Engineering, Politecnico di Torino, Turin, 10129, Italy
[2]Interuniversity Department of Regional and Urban Studies and Planning, Politecnico di Torino, Turin, 10125, Italy
[3]Research Institute for Geo-Hydrological Protection, CNR, Padova, Italy
[4]Department of Architecture and Design, Politecnico di Torino, Turin, 10125, Italy
[5]†deceased

**Correspondence:** Elisabetta Corte (elisabetta.corte@polito.it)

**Abstract.**

The recession of Alpine glaciers causes an increase in the extent of proglacial areas and leads to changes in the water discharge and sediment balance (morphodynamics and sediment transport). Although the processes occurring in proglacial areas are relevant not only from a scientific point of view but also for the purpose of climate change adaptation, there is a lack of studies on the continuous monitoring and multitemporal characterization of these areas. This work offers a multidisciplinary approach that merges the contributions of different scientific disciplines such as hydrology, geophysics, geomatics, and water engineering to characterise the Rutor glacier and its proglacial area. We surveyed the glacier and its proglacial area since 2020 with uncrewed and crewed aerial surveys; we determined the bathymetry of the most downstream proglacial lake and the thickness of the sediments deposited on its bottom. Water depth at four different locations within the hydrographic network of the proglacial area and the bedload at the glacier snout were continuously monitored. The synergy of our approach enables the characterisation, monitoring, and understanding of a set of complex and interconnected processes occurring in a proglacial area.

## 1 Introduction

Global warming is entailing a rapid decline of the cryosphere globally. Mountain snow cover and glaciers, respond directly and rapidly to climate change making them key indicators of global warming. The intensity and frequency of precipitation are changing and part of the precipitation has shifted from solid to liquid due to the increased temperatures (e.g. in the European mountains). This shift and the increasing number of dry and warm winter days in European Alps reduce snow accumulation. In addition, rising air temperature in spring increases snow melt, modifying the local water balance (Carrer et al., 2023; Gizzi et al., 2022). Snowfall and ice/snow melt impact glacier mass balance. As a consequence of global warming, the glaciers within the European Alps are subject to reduction in surface area and ice mass (Sommer et al., 2020).

Most glaciers reached their Holocene maximum extent at the end of the Little Ice Age (LIA) and have receded since then (Grove, 2004). With LIA being a cooler period in the Holocene, lasting from years 1300s to 1850s (Ivy-Ochs et al., 2009). The decline of snow cover and glaciers exposes more land and water surfaces to solar energy, leading to decreasing albedo and to weathering, resulting in increased erosion. Glaciers produce a considerable amount of sediments (Hallet et al., 1996), the size of which ranges from large boulders to fine sands, silt, and clay (Hallet et al., 1996; Carrivick and Tweed, 2021). Depending on dynamic and thermodynamic conditions, glaciers have the ability to entrain sediment and erode bedrock. Even when the entrainment capacity is reduced, the glacier retains the ability to deform the sediment (Alley et al., 1997).

Using the terminology defined by Slaymaker (2011), the area encompassing the glacier outline at the end of the LIA and the present-day glacier terminus is the proglacial area. Proglacial areas are considered systems in transition from glacial to non-glacial conditions and are therefore natural laboratories that allow the investigation of the early stages of newly exposed soil development, vegetation succession, and associated soil stability and sediment fluxes (Matthews, 2019). Due to global warming and glacial retreat, disequilibrium occurs between sediment delivery from the glacier and fluvial reworking in proglacial areas (Slaymaker, 2011). Their evolution depends on the interaction between geomorphic processes and vegetation succession. On the one hand, plant colonization stabilizes glacial sediment and reduces sediment fluxes; on the other hand, geomorphic processes disturb and limit vegetation succession (Curry et al., 2006; Moreau et al., 2008; Eichel, 2019).

Studies investigating multiple processes within a proglacial area, on a larger scale than a single landform or a hillslope, at different time-frames are not frequent (Hilger and Beylich, 2019). The integration of all the processes involved in the sediment budget requires a catchment-wide identification, mapping, and quantification of all relevant sediment transport processes, a localization and monitoring of the storage elements in the sediment transport system, and a localization of their interaction areas (Hilger and Beylich, 2019). Carrivick and Tweed (2021) state that the remobilization of sediment within the proglacial area mainly determines sediment yield in a proglacial area. Guillon et al. (2018), in their study of the Bosson Glacier (FR), found that sediment sources vary according to season; sediment remobilisation within the sandur is the dominant source of sediment in autumn, while during the melt season the main export of sediment comes from the glacial source. Further efforts in integrating multiparametric observations and enhancing interdisciplinary scientific collaboration are needed to predict sediment dynamics in a warming world (Zhang et al., 2022)

Sediment availability is strongly governed by morphology (Cavalli et al., 2018). The land-system elements of a proglacial area have different geomorphic functions and are heterogeneously distributed. These elements can act like sediment sources, stores (short-term storage landforms), and sinks (long-term storage landforms) (Matthews, 2019). In Alpine catchments, runoff depends on rainfall events, snow, and glacier melt (Camporese et al., 2014). Glacier retreat in response to the local climate is heterogeneous in space and time and so is the water regime. Sediment yield depends on runoff and sediment availability which are both highly variable in space and time (Heckmann and Schwanghart, 2013; Hooke, 2000; Carrivick and Tweed, 2021). Furthermore, the connection among water discharge, bedload, and suspended sediment transport exhibits variability over the years and within seasons, influenced by climatic conditions as highlighted in previous studies (Mao et al., 2018; Coviello et al., 2022).

In this work, to the best of our knowledge, we present the first public dataset of a proglacial area that is the result of hydrological, geophysical, geomatics, and water engineering monitoring. This dataset is the result of a multidisciplinary approach and represents the input data to assess the water and sediment balance in the Rutor proglacial area and the morphodynamics occurring in recently exposed soils. The synergy among different disciplines has allowed for achieving a holistic viewpoint in the observation of the evolutive phenomena of the Rutor proglacial area.

## 2   Materials and methods

### 2.1   Site description

The Rutor glacier lies at the head of the Dora Baltea Valley in La Thuile, near the French-Italian border in northwestern Italy. It is mainly oriented to the northwest and at an altitude ranging from 2540 m a.s.l. to 3486 m a.s.l., with an average altitude close to the average value for Alpine glaciers, as retrieved from the Global Land Ice Measurements from Space (GLIMS) database
(GLIMS Consortium, 2005; Raup et al., 2007). The Rutor glacier is among the glaciers with the largest surface area in the Alps, and it is the third largest glacier in the Aosta Valley (GLIMS Consortium, 2005; Raup et al., 2007). At present, it has a surface area of 7.5 km$^2$ and its front is formed by three tongues (Figure 1) that were once united (Figure 2). Villa et al. (2007) determined the Rutor glacier retreat and volume changes from the mid-19$^{th}$ century to 2004. Since 2005, the regional environmental protection agency (ARPA) of Valle d'Aosta has been monitoring the mass balance of the Rutor glacier, which
with the exception of the years 2013, 2014 and 2016 has always been negative, resulting in a cumulative mass balance from 2005 to 2017 of -12252 mm w.e. (ARPA Valle d'Aosta, 2014). Since its maximum extent in LIA (Orombelli, 2005; Villa et al., 2007), the glacier has lost approximately 34% of its surface area. The retreat and lowering of the glacier surface are not uniform but more pronounced in the eastern tongue (Villa et al., 2007).

    The entire Rutor proglacial area spans approximately 4 km$^2$ (Villa et al., 2007). This area holds significant importance for
investigating sediment dynamics in proglacial systems, owing to its geomorphological diversity and pristine condition resulting from minimal human impact. Notably, the presence of L4, which acts as a basin closure within the proglacial area, collects all mobilized sediment within the region. Since the end of LIA, the Rutor glacier has retreated, leading to a progressive increase in the proglacial area. The glacier recession has exposed topographic depressions which determined changes in stream networks and the formation of several proglacial lakes. These lakes act as sediment sinks, interrupting sediment transfer from the glacier
outlet to the lowlands. The altitude from the lowest proglacial lake to the glacier terminus (middle tongue snout) ranges from about 2390 m a.s.l. to 2660 m a.s.l. The land-system elements within the Rutor proglacial area include steep slopes, outwash plains (sandurs), and single and braided channels, while the alluvial channel beds and banks vary in size from fine sands, silt, and clays to boulders. L1 has a single outflow which, after a distance of 830 meters, flows into a sandur. This sandur is fed by the meltwater of the entire glacier and has a surface area of about 0,1 km$^2$. Due to a topographic barrier, the water is forced to
flow downstream the outwash plain through a single channel. When the water level in the sandur rises, the above-mentioned topographic barrier determines the formation of the L2 lake (2504 m a.s.l.). The water flows from L2 to the L4 proglacial lake (Seracchi lake, 2387 m a.s.l.), through a steep creek with an elevation jump of about 100 m.

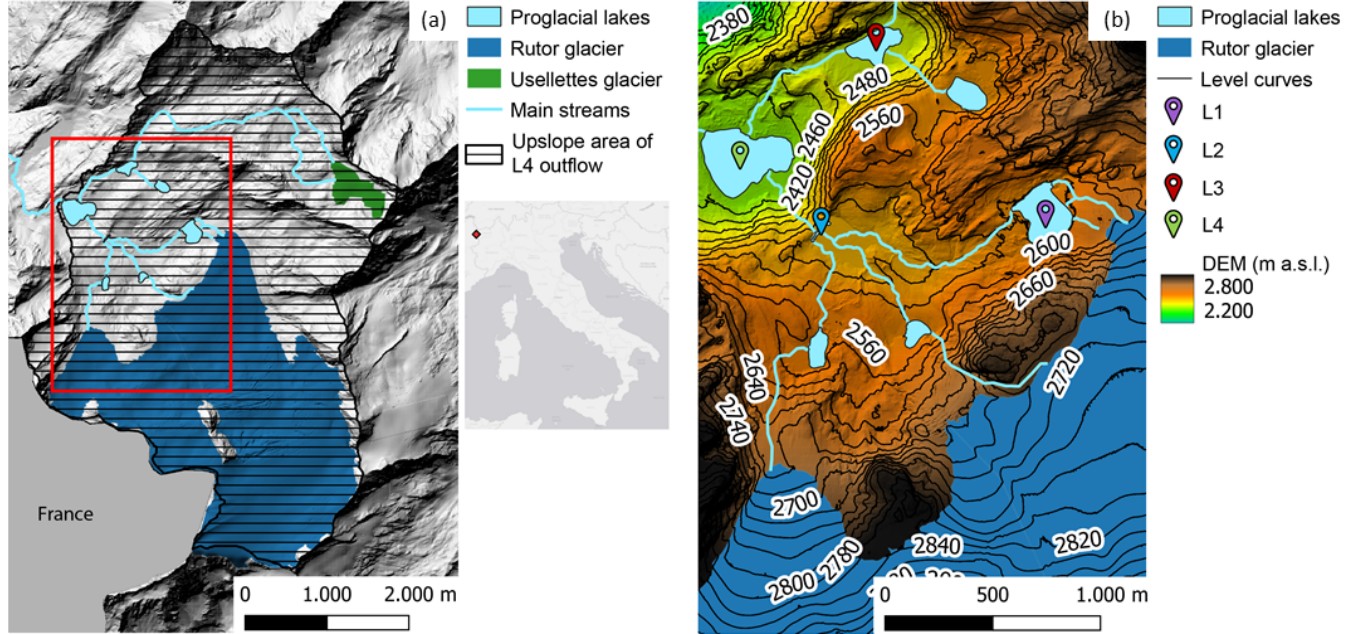

**Figure 1. (a)** Hillshade based on the Digital Surface Model (DSM) as of 2008 of the Rutor glacier and the L4 lake catchment. The upslope area of L4 outflow (hatched area with continuous black lines) has been mapped using the 2008 model of Valle d'Aosta (SCT Geoportale, Regione autonoma Valle d'Aosta). The inset shows the location in Italy. **(b)** DSM as of 2021 of the Rutor proglacial area and locations of L1, L2, L3, and L4 proglacial lakes.

The outflows of L2 and L3 (Santa Margherita Lake) are the only two surface inflows of the L4 Lake, whose outflow feeds the majestic Rutor waterfalls. The L4 lake collects all meltwater from the Rutor glacier and is the major and the most downstream proglacial lake of the analyzed area. Its outflow cross-section is quite stable and allows to easily measure the lake outflow. Since the main processes involving the water and sediment budget of the Rutor proglacial area occur upstream and within L4, the study focuses on the basin area upstream of the outflow control section of L4, with an overall catchment area of 18,12 km$^2$, whose 43% is glacierized (see Figure 1(a)).

The characteristics of the study area described above can be easily observed through a WebGIS available at https://arcg.is/Tyeju0 (last access: 15 February 2024).

Among all the lakes of the area, the Santa Margherita Lake – here named L3 (2422 m a.s.l.) – was the most monitored in the past because of catastrophic outburst floods (Baretti, 1880; Sacco, 1917), which began in the first half of the XV century, showing that the glacier at the time had already retreated (Sacco, 1917).

The past evolution of L3 lake testifies the changes that the whole area had gone through due to the glacier retreat since the end of the LIA. These changes have been reported in several documents (e.g., Sacco, 1917; Baretti, 1880; Valbusa and Peretti, 1937), that allow reconstructing the changes of the glacier and its proglacial area.

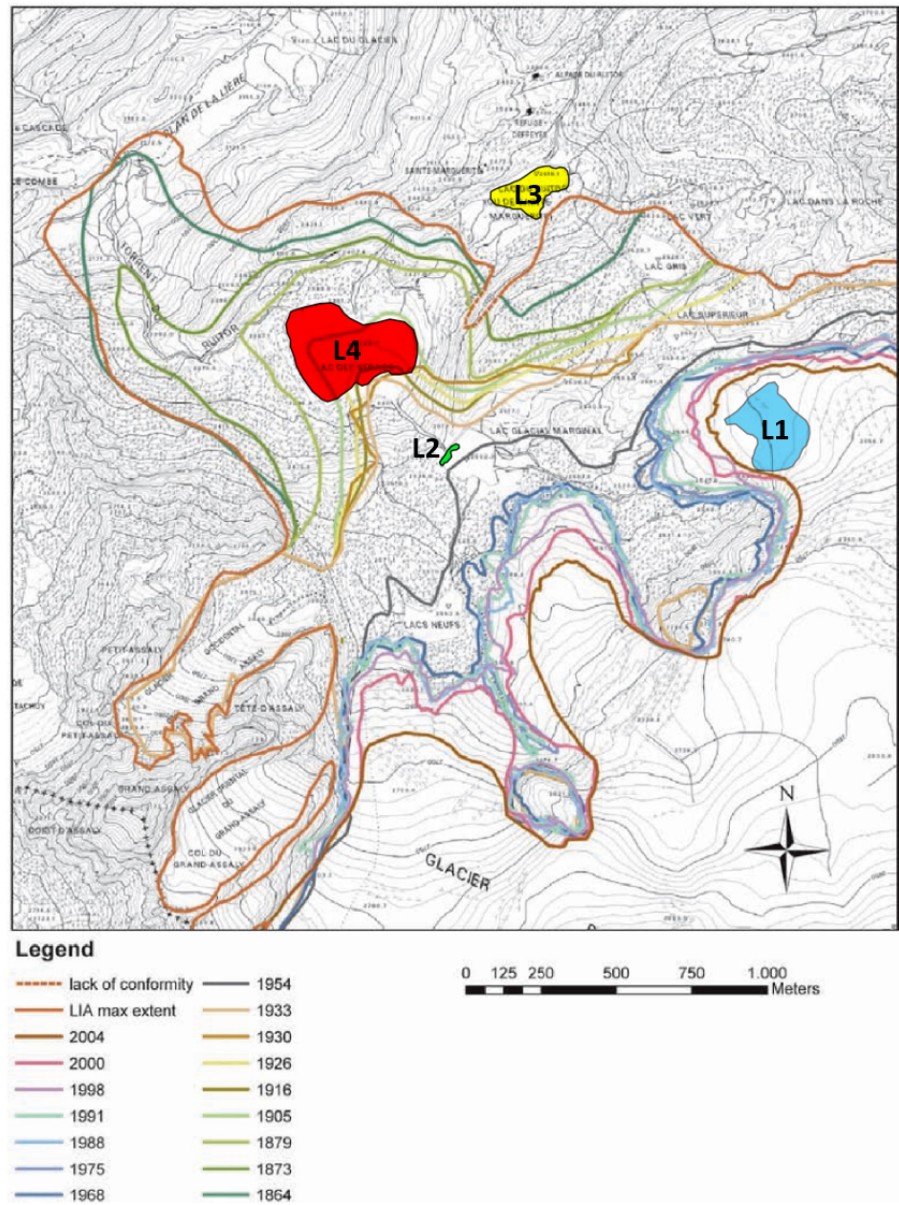

**Figure 2.** Reconstruction of the Rutor glacier terminus from its maximum extent in LIA to 2004 (modified from Villa et al., 2007). The areas highlighted in blue, green, yellow, and red indicate the current extent of lakes L1, L2, L3, and L4 respectively.

## 2.2 Multidisciplinary framework

The assessment of the water balance and sediment budget implies the identification of the different physical processes involved, their geomorphic function, their contribution to the overall sediment production and their effectiveness in supplying sediment

to the mainstream (Hilger and Beylich, 2019). Quantifying the sediment budget of proglacial areas is a challenging task due to the multitude of processes involved and their spatial and temporal variability. Most studies either focus on a single landform or hillslope at different times (e.g., Laute and Beylich, 2014; Curry et al., 2006), or they measure river-basin scale production rates at the outlet of the basin (e.g., Hicks et al., 1990; Müller, 1999; Bogen et al., 2015). The following paragraph provides a concise overview of the monitoring methods used in three distinct studies concerning different proglacial areas.

Guillon et al. (2018) combined sedimentary measurements with precipitation data to understand present-day suspended sediment storage and erosion processes during a melt season. They measured water depth and turbidity, deriving water discharge and suspended sediment concentration respectively, in three different stations. Orwin and Smart (2004) characterized a proglacial channel over a 9-week ablation period by continuously measuring the water depth and turbidity in different gauging stations distributed within the proglacial area. Confirming that sediment yield varies spatially and temporally within a proglacial area. Delaney et al. (2018) assessed erosion rates and processes in an alpine proglacial area through digital surface models (DSMs), reservoir bathymetry and a glacial-hydrological model (GERM). Water discharge measurements were determined by the water level at the reservoir located at the basin outlet. The first two reported studies (Guillon et al., 2018; Orwin and Smart, 2004) provided an explanation for the variation in space and time of proglacial suspended sediment flux but they did not assess the landscape evolution of the geomorphological features in the whole proglacial area, whereas in the latter reported study (Delaney et al., 2018) the sediment processes in the whole proglacial area was identified but the water discharge was directly measured only at the basin outlet. The studies presented are of important value for the understanding of the dynamics in proglacial areas, but there is a lack of studies in the literature involving repeated surveys (e.g. photogrammetric flights) and continuous monitoring (e.g. flow measurements) at several points in the proglacial and glacial area.

**Table 1.** Timetable of continuous measurements and field surveys, in the first column the measured quantities/surveys and at the bottom the timeline. Each colour and symbol are characteristic of the measuring station and surveyed area respectively. The arrows between the dates indicate that the measurements are continuous between the dates.

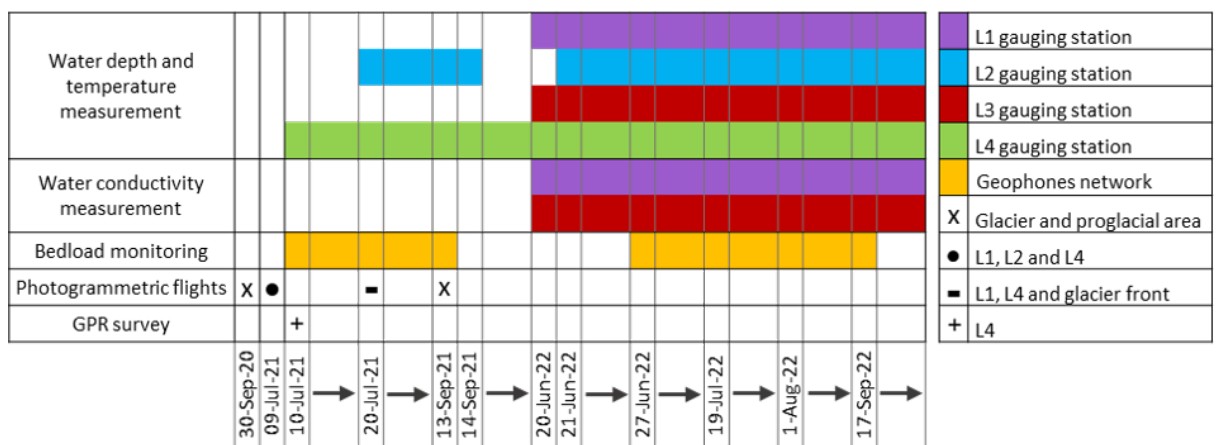

Since 2005, the local environmental agency (ARPA) of Aosta Valley has been monitoring the mass balance of the Rutor glacier through direct in situ measurements. Starting from 2020, the Glacier Lab of the Turin Polytechnic integrated ARPA

surveys with geophysical and geomatics measurements. Since the summer of 2021, the area monitored by the Glacier Lab has increased from 25.2 to 34.5 km$^2$ to include the proglacial area.

Our monitoring activities at the Rutor glacier can be categorized into multitemporal and continuous surveys. An overview of these monitoring activities is provided in Table 1.

## 2.3 Geomatic survey: aerial acquisitions and GNSS positioning

The Rutor glacier was monitored with different geomatics techniques supported by different surveying campaigns, with two aims: i) to provide a common 3D reference system to properly manage all the spatial and temporal datasets of the different research groups involved in the glacier monitoring, and ii) to enable the 4D (3D over the time) monitoring of the extent and morphology of the glacier surface. The geomatics surveys started in 2020 and include both uncrewed and crewed aerial photogrammetric flights as well as topographic measurements in the field. The geomatics surveys were carried out in parallel with the activities of the other Glacier Lab teams, to acquire in-situ data and enable the implementation of integrated multidisciplinary monitoring activities.

In the 2020-2021 period, the surveys described in Table 2 have been carried out. Among those, during the summer campaigns in 2021, three photogrammetric flights (one on $9^{th}$ July 2021 and two on $20^{th}$ July 2021) were carried out with the DJI Phantom 4 RTK UAV multirotor platform (using the UAV's built-in camera equipped with a 1" RGB sensor) to survey the proglacial lakes.

**Table 2.** Photogrammetric flights carried out on the study area between 2020 and 2021.

| Platform | Date of acquisition | Covered area | Extent (km$^2$) | Average flight height (m) | GSD (m) | n. of Images | n. of GCPs | n. of CPs | Image overlap |
|---|---|---|---|---|---|---|---|---|---|
| Aerial | 30/09/2020 | Glacier and a portion of the proglacial area | 25.2 | 818 | 0.07 | 867 | 18 | 7 | 60% |
| UAV | 9/07/2021 | L1, L2, and L4 | 2.6 | 126 | 0.03 | 1480 | 8 | 6 | 80% |
| UAV | 20/07/2021 | L1 and L4 | 0.4 | 89.2 | 0.02 | 369 | 5 | 2 | 80% |
| UAV | 20/07/2021 | Glacier front and lower part | 1.1 | 159 | 0.04 | 623 | Direct georeferencing | Direct georeferencing | 80% |
| Aerial | 13/09/2021 | Glacier and proglacial area | 34.5 | 877 | 0.06 | 1100 | 9 | 4 | 60% |

After the summer of 2021, at the end of the hydrological year 2020/21, crewed photogrammetric flights were carried out by the Digisky company over the glacier and proglacial area, using a medium-format PhaseOne camera iXM-RS150F installed onboard an ultralight aircraft. The crewed aerial flight was carried out with a P90e light aircraft. Its handling allows easy flight

altitude changes to maintain a constant GSD. The camera has a focal length of 50 mm, a sensor size of 40 x 53.5 mm and a resolution of 151.3 MP. The 2021 photogrammetric survey was the repetition of a previous flight, carried out at the end of September 2020 with the same aerial platform and sensors but with a smaller coverage (without a complete coverage of L2, L3, and L4 lakes).

The dataset acquired during the UAV and aerial photogrammetric flights were processed to obtain a 3D model of the terrain and additional cartographic products, i.e. orthophotos and DSMs. A standard Structure-from-Motion (SfM) photogrammetric approach was adopted, following a consolidated workflow using the software Agisoft Metashape, i.e.:

– Image alignment, to estimate interior/exterior (relative) orientation parameters, generating a relative sparse point cloud using feature detection and matching and SfM-based bundle block adjustment with self-calibration.

– Collimation of Ground Control Points (GCPs, not relevant in case of a direct georefencing approach), to re-estimate interior/exterior (absolute) orientation parameters refining the SfM-based bundle block adjustment with self-calibration, to generate a georeferenced sparse point cloud.

– Evaluation of residuals on GPCs and Check Points (CPs) and iteration of the previous step in case of anomalies in the residuals.

– Generation of a dense point cloud.

– Generation of a Digital Surface Model (with respect to a cartographic plane) and Orthoimagery.

The photogrammetric processing reports generated by Metashape software for the two crewed aerial flights (the only ones used for elevation analyses) are available in the supplementary material section (https://doi.org/10.5281/zenodo.11144390, Corte et al. (2024)). These reports include information on processing parameters settings, survey data details, camera calibration, camera locations, ground control points, and check points.

During the 2021 field activities, a total of 32 artificial photogrammetric markers, either squared (0.5 m x 0.5 m) plastered markers or crosses painted on stable rocks, were positioned (or painted) and measured with a Real-Time Kinematic (RTK) and static Global Navigation Satellite System (GNSS) positioning approach, using 3 Spectra Precision SP80 GNSS receivers (static data have been processed with RTKLIB software). The markers were distributed on the proglacial area (to ensure stability over time), around L4, and along the L1 until the glacier front on the eastern tongue. Among the 32 markers, 12 larger markers (1 m x 1 m) were positioned during the September 2021 campaign around the top part of the glacier area, to enable a straightforward identification on aerial images. The markers placed in 2021 have been used as both GCP and independent CP (details in Table 2) for the 2021 crewed aerial survey. Considering that the focus is on relative displacements rather than on absolute values, 25 natural GCPs and CPs have been then identified on the 2021 orthomosaic and DSM to orient the 2020 crewed aerial imagery and assess its 3D positional accuracy (considering that the artificial markers were not yet available in 2020). A GPC/CP based approach has been used also for UAV surveys (two markers used for the aerial dataset were also leveraged for the UAV datasets), except for one UAV survey where, exploiting the RTK capabilities of the UAV GNSS receiver,

a direct georeferencing approach has been adopted (considering it was not possible to place markers in the glacier front for safety reasons). Direct Georeferencing refers to the orientation of remotely sensed imagery without using GCP, exploiting Real Time Kinematic (RTK) or Post Processing Kinematic (PPK) approaches. The RTK- or PPK-based approach enables the generation of metric products with 3D positional precision and accuracy in the range of few centimeters (Chiabrando et al., 2019; Teppati Losè et al., 2020a, b).

Since the camera positions of aerial flights were not geo-tagged with proper accuracy, it was necessary to exploit the artificial markers to georeference the 3D model accurately over the entire glacier area. The cartographic reference system adopted for all the 3D models is ETRF2000/UTM32N; the ellipsoidal height was reduced to orthometric height by applying the Italian geoid model ITALGEO05. Due to the availability of a suitable number of well-distributed ground control points, the 2021 aerial survey was considered the reference model (referred to as 'Model Zero') to be used for multitemporal analyses. The 2020 survey was, therefore, co-registered (i.e., georeferenced in the same reference system, enabling the overlap of all the derivative products) with the 2021 survey.

Since one of the main objective is the evaluation of relative displacements between multitemporal 3D models and considering that the main focus is on the elevation component, multitemporal DSMs were compared using a pixel-by-pixel approach. It has to be highlighted that, being the focus on the entire glacier area and considering that UAV flights cover only the terminus of one of the glacier tongues, the elevation analysis described in this manuscript is related solely to the crewed aerial dataset. In particular, the aerial DSM 2020 was subtracted from the aerial DSM 2021 to calculate the altimetric differences, referred to as Difference of DSM (DoD). The following sections is focused on the assessment of the positional accuracy of the datasets derived by the aerial photogrammetric flights, focusing on the elevation.

### 2.3.1 DSM Validiton: DoD and LoD estimation

To properly assess the photogrammetric results (i.e. in this specific case the DSMs and their differences), it is necessary to apply "suitable statistics to identify systematic error (bias) and to estimate precision" and to propagate "uncertainty estimates into the final data products" (James et al., 2019). When comparing DSMs from two different epochs (t0 and t1), their DoD need to be evaluated with reference to its Limit of Detection (LoD), i.e. the threshold for the significance of the DoD values.

$$DoD = Z_{t1} - Z_{t0} \qquad (1)$$

The estimation of the LoD, with a certain level of significance, can be conducted either i) by analyzing altimetric residuals at specific independent CPs or ii) through the analysis of areas that remain unchanged over time, where significant altimetric variations are not expected (i.e stable terrain) (Paul et al., 2017). Each method has its own advantages and disadvantages. For this particular multidisciplinary study, it has been decided to determine LoD based on the analysis of residuals at CPs. This decision was made both to adopt a cautious approach (the relative LoD value at 95% is slightly less favorable compared to that derived from stable areas) and because it was not feasible to identify uniformly distributed stable areas across the entire region covered by the photogrammetric surveys.

The first approach relies on the statistical analysis of altimetric residuals ($\Delta Z$) at some independent CPs, which are points with known coordinates that, unlike GCPs, have not been used during the photogrammetric image orientation parameter estimation phase. Preliminarily, it is important to highlight the difficulty in acquiring independent CPs with field measurements in glacial areas and generally in high mountain zones, both due to the limited presence of suitable areas for establishing points and the intrinsic difficulties and related constraints in movements. Despite having a limited sample of CPs, it is possible to perform a statistical analysis to assess the altimetric accuracy and precision of DSMs, particularly by calculating the mean (to analyze possible systematic errors) and standard deviation of $\Delta Z$ respectively. Naturally, these statistics are also calculated for the planimetric residuals $\Delta X$ and $\Delta Y$, since significant planimetric systematic errors would also influence altimetric precision. The results of such analysis can be summarized in a table structured similarly to Table 3.

**Table 3.** Means and standard deviations of the residuals $\Delta X$, $\Delta Y$, and $\Delta Z$ calculated on GCP and CP.

| | | Residuals GCPs [cm] | | | Residuals CPs [cm] | | |
|---|---|---|---|---|---|---|---|
| n. GPPs | n. CPs | $\mu_{\Delta X} \pm \sigma_{\Delta X}$ | $\mu_{\Delta Y} \pm \sigma_{\Delta Y}$ | $\mu_{\Delta Z} \pm \sigma_{\Delta Z}$ | $\mu_{\Delta X} \pm \sigma_{\Delta X}$ | $\mu_{\Delta Y} \pm \sigma_{\Delta Y}$ | $\mu_{\Delta Z} \pm \sigma_{\Delta Z}$ |

Where:

$$\Delta K = K_{Photogrammetry} - K_{(G)CP} = Residual K \tag{2}$$

$$\mu_{\Delta K} = \text{Mean } \Delta K \tag{3}$$

$$\sigma_{\Delta K} = \text{Standard Deviation } \Delta K \tag{4}$$

$$K = \{X, Y, Z\}, \text{ i.e. } 3D \text{ coordinates components} \tag{5}$$

Mean values significantly lower than the standard deviation imply, in the case of $\Delta Z$ residual analysis, where a distribution with a mean of zero is expected, the absence of systematic errors, which - if present - should be corrected before proceeding with the estimation of the LoD. The LoD - at a 95% confidence level - of the differences between two multitemporal DSMs (acquired at epochs $t_0$ and $t_1$) can thus be derived from the variance propagation law applied to $\sigma_{\Delta Z}$ (last column of Table 3), as demonstrated in Brasington et al. (2000) and Lane et al. (2003). In particular, under the assumption of normal distribution and independence of variables:

$$\sigma_{\text{DoD}} = \sqrt{\sigma_{\Delta Z_{t1}}^2 + \sigma_{\Delta Z_{t0}}^2} \tag{6}$$

$$\text{LoD}_{95\%} = 2\sigma_{\text{DoD}} \tag{7}$$

In this context, t1 and t0 designate the specific temporal points considered for the multitemporal analysis, indicating the respective acquisition dates of the source data utilized in producing the DSMs.

Alternatively, or in support of, the analysis based on CPs, the estimation of the LoD can be conducted using differences in DSMs over time, derived from multitemporal analysis on areas of the terrain considered "stable" over time. These stable areas must be meticulously delineated outside of glacial regions susceptible to melting or more generally subject to changes. Except for potential outliers, in stable terrain, the values of DoD should tend towards zero (considering the stability of the areas and the altimetric precision of the DSMs). For determining the LoD of the DoD, even with the approach based on stable areas, it is necessary to statistically analyze the altimetric differences. In particular, the distribution of DoD values, unlike the CP-based approach, will involve a large number of points (even millions of points), significantly increasing statistical significance. Preliminarily, however, it is necessary to remove any outliers to evaluate whether the distribution of DoD follows a normal distribution with a mean of zero (given that the DoD concerns stable terrain) and estimate its variance. Outliers may arise from either localized terrain variations not identified during the delineation of stable areas or potential errors in the automatic correlation of the photogrammetric process in regions with "monotone" texture. The process of excluding outliers can be carried out by removing the tails of the DoD distribution, for example, by eliminating the top and bottom 5% of the distribution tails. Once the normality of the distribution is confirmed, it is possible to compute the mean and standard deviation of the DoD ($\sigma_{DoD}$) on the sample without the top and bottom 5% of the distribution tails. In analogy to the CP-based approach, the LoD can be then calculated as two times the $\sigma_{DoD}$ (see eq.7).

## 2.4 Geophysical survey

The bathymetry of Seracchi Lake and the thickness of the sediments deposited on its bottom were determined by using a Ground Penetrating Radar (GPR) (Sambuelli et al., 2015) supported by Time Domain Reflectometry (TDR) measurements (He et al., 2021), as reported in more detail in Vergnano et al. (2023).

Both systems are based on the principle of the propagation of high-frequency electromagnetic signals, in the bandwidth between 30 MHz and 1 GHz. The signal propagation in natural media depends on the electromagnetic properties of the media (dielectric permittivity and electrical conductivity). In a low-conductivity material, the signal propagates with a velocity related to the dielectric permittivity, according to $v = c/\sqrt{\varepsilon}$, where $c$ is the electromagnetic wave velocity in vacuum and $\varepsilon$ is the relative dielectric permittivity of the material (Psarras, 2018). The velocity is usually estimated in the time domain: a signal pulse is excited by an antenna (in GPR) or TDR device, and it propagates into the medium; part of the energy carried out by the signal is scattered back (or reflected) when a contrast of electromagnetic impedance is encountered. The amount of energy that is reflected depends on the contrast of electrical conductivity or dielectric permittivity between two different media. The backscattered signal is then collected by an antenna (receiving antenna in GPR) or by an oscilloscope in the case of TDR devices. In the GPR, the amplitude of the signal that is backscattered at the interface between two different media defines the reflectivity of the target. The GPR approach for detecting the bathymetry of a lake is based on the reflectivity of the lake bottom, based on the contrast of dielectric permittivity between water and sediments of the lake bottom.

The dielectric permittivity of water depends on the temperature, and it is slightly affected by salinity; typical values at low temperatures are around 80 (relative values, referring to the dielectric permittivity of vacuum), corresponding to an e.m. waves velocity in water of around 0.033 m/ns. In our case, with a 6-degree temperature and a relative permittivity of 83.3, the wave velocity was estimated to be 0.0327 m/ns. High porosity sediments could exhibit dielectric permittivity in the range between 35 and 40. This means that the water-sediments interface should exhibit good reflectivity, given the contrast of dielectric permittivity values.

The GPR antenna, manufactured by IDS GeoRadar s.r.l., had a central frequency of 200 MHz, which provides the best possible resolution while avoiding the energy dispersion that occurs in water at frequencies higher than 200 MHz (Bradford et al., 2007). The GPR system was installed on an inflatable rowing boat and the boat was moved to cover the whole area of the lake. The GPR sections acquired were processed according to a set of standard processing steps, performed in Reflexw software (Sandmeier, 2021; Vergnano et al., 2023), and reported in Appendix B.

The x-y-z locations of the first interface, representing the lake bottom, detected in all the GPR sections, were interpolated with a linear triangulation-based method (griddata function of MATLAB) to produce a bathymetry map (Figure 11, which also displays the sediment thickness distribution and the electrical conductivity measurements). The perimeter of the lake, retrieved from the 6-cm-resolution orthophoto acquired on the day of the geophysical survey, was useful to fix the 0-depth in the interpolation process.

The TDR probe, installed on a rod, was inserted in the lake bottom sediments at several locations and measured their electrical conductivity and dielectric permittivity. The valence of the TDR survey is double:

– to corroborate the interpretation of the GPR sections, because the punctual values of dielectric permittivity give an estimation of the electromagnetic wave velocity in the sediment (v), necessary to convert the GPR travel times into thickness of the sediments itself. Also, the dielectric permittivity can be used to define the expected reflectivity of the water-sediments interface and to estimate the sediment porosity, since they are considered fully saturated.

– to assess the spatial variability of the type of sediments by measuring their electrical conductivity. In fact, the electrical conductivity of the lake sediments depends on the porosity, water salinity and temperature, and texture of the sediments; the electrical conductivity is a good indicator of the presence of finer material, as the bulk electrical conductivity usually increases due to the contribution of the surface electrical conduction of the finer particles.

To validate the GPR and TDR measurements, geotechnical analyses (grain size distribution and Atterberg limits) were performed on a few sediment samples collected at the locations shown in Figure 11 (see Vergnano et al. (2023) for more details).

## 2.5 Hydraulic monitoring

The hydrography of the Rutor proglacial area is made complex by a sequence of flat and steep areas, by the presence of several proglacial lakes differently connected and by the contribution from three tongues of the glacier. In order to assess the partial and total surface runoff, four instruments were installed to measure the water depth at different locations in the study area. The

location of these water pressure gauges was determined by the accessibility and the geometry of the channel or lake and the presence of stable rocks or banks on which to install the instruments.

Two types of instruments were installed: i) a self-contained, water logger and transmitter measuring water level and temperature (OTT ecoLog 1000); ii) a combined measurement of water level, temperature, and conductivity (OTT CTD). The four locations of the gauges stations, from upstream to downstream, are L1 emissary, L2, L3 emissary, and the outflow of L4 (Figure 3). Water depth allows one to retrieve from direct velocity measurements the water discharge, and conductivity measurements allow water characterization for surface or groundwater flow, which is a matter of interest for L3 and L4. Therefore, the two OTT CTDs were installed in L1 and L3 emissaries. The ecoLog1000 and CTDs instruments were first installed in July 2021 and June 2022, respectively; the measuring periods of each sensor are shown in Table 1.

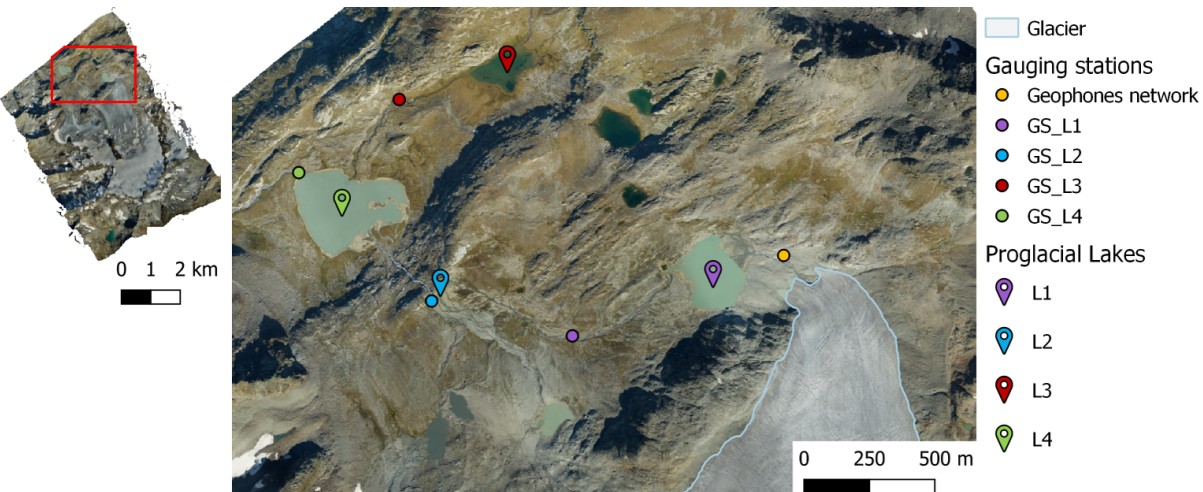

**Figure 3.** Aerial orthophoto of the Rutor proglacial area acquired on 13/09/2021 and the snout of the Rutor eastern tongue. The red polygon in the upper left orthophoto shows the position of the area enlarged in the right figure. The lakes (L1,L2,L3, and L4), the gauging stations and the geophones network are indicated.

The upslope areas of the 4 sensors installed are: 5.3 km$^2$ for L1 gauging station, 12.6 km$^2$ for L2 gauging station, 4.9 km$^2$ for L3 gauging station, and 18 km$^2$ for L4 gauging station.

Since the area covered by the photogrammetric flights (2020 and 2021) excluded a portion of the upstream area of L1 and L3 gauging stations, these areas were determined using the 2008 DSM of Aosta Valley (SCT Geoportale, Regione autonoma Valle d'Aosta).

At all the cross sections of the gauging stations, with the exception of L2, the flow velocity was measured with a current meter. At L2, due to the high flow velocity during the summer season, direct measurements are not safe for the operator. To derive the discharge from the water level measurements, a stage-discharge (or rating) curve has to be developed. In the summer of 2021 and 2022, a set of nine flow velocity measurements were taken with an Acoustic Doppler Velocimeter (ADV) current meter in the cross-section of gauging station L4. The velocity-based discharge measurements $Q$ were related to the

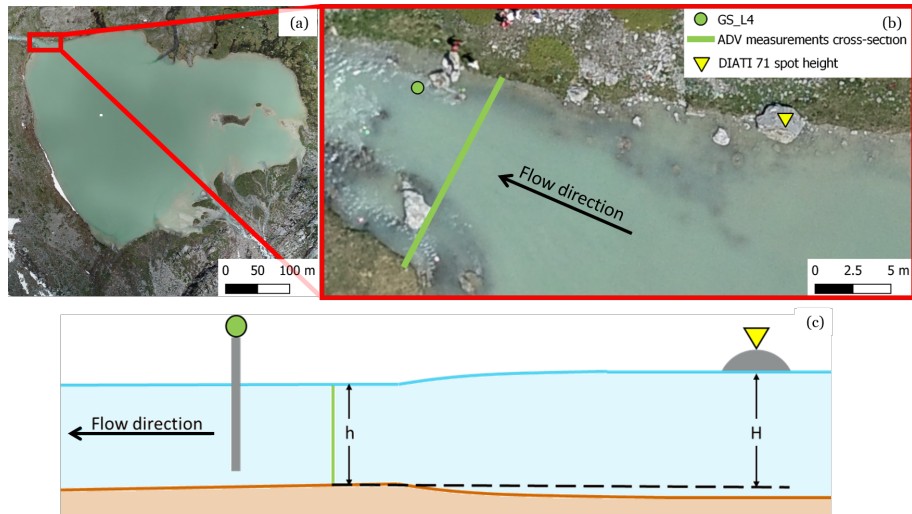

**Figure 4.** (a) Orthophoto of L4 (July 2021); (b) zoom in of the L4 outflow where the ADV measurements were taken and the DIATI 71 spot height is shown; (c) the longitudinal cross-section of the L4 outfall shows the reference water depth of the emissary (h) and the reference total head measured in the lake (H).

corresponding water depth $h$ measured at the gauge (Figure 4), to plot the stage-discharge diagram (Figure 5(a), further details on the procedure are given in Appendix A).

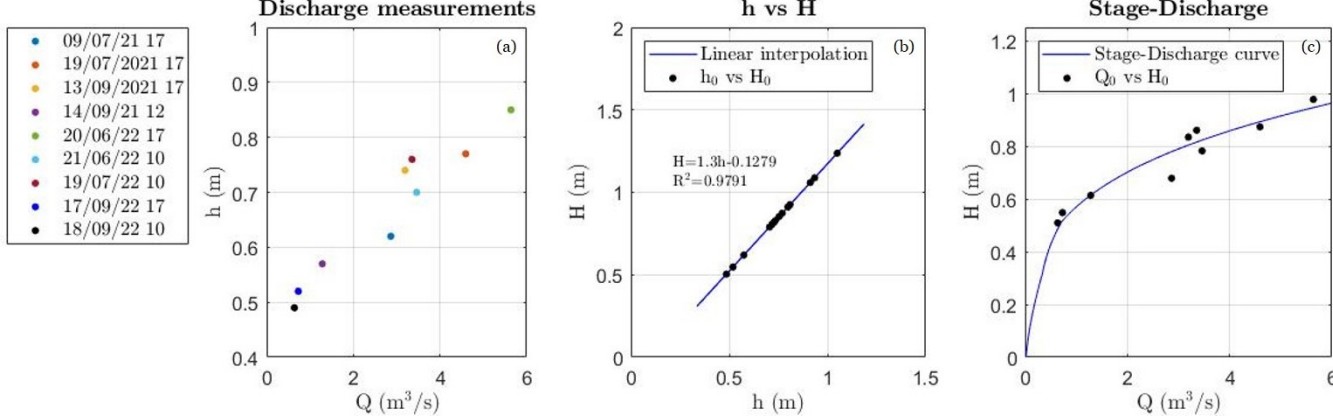

**Figure 5. (a)** Discharge measurements in L4 emissary and the corresponding water depth (h). **(b)** Measurements of total head H and the corresponding water depth h. The linear interpolation equation and the coefficient of determination ($R^2$) are reported. **(c)** Discharge measurements and the corresponding total head H and the stage-discharge curve for L4 emissary.

L4 is the largest and the most downstream lake of the area collecting the whole meltwater of the Rutor glacier and the suspended sediment of the upstream area. Monitoring the water level and the outflow of L4 is crucial to assess the water and

sediment budget of the Rutor proglacial area. Due to backwater effects at the outflow, the water levels in the lake and the control cross-section are not identical, but strictly related. Therefore, to monitor the stage of the lake, a relationship between the continuously recorded water level at the gauging station and the water level in the lake far from the gauging station was determined. A spot height was placed on a rock near the shore of the lake (Figure 4); the water level of the lake was assessed by measuring the altitude difference with the spot height (DIATI 71) using a laser level and a levelling staff. A total of 15 altitude difference measurements were taken during the 2022 summer campaigns. The position of the instrument at L4 gauging station and the geometry of the L4 outfall cross-section were measured with an RTK positioning approach (Table 4). This made it possible to determine the position of the measuring point of the instrument and to establish a reference elevation against which to assign the water depth in the outfall cross-section (h) and the water depth in the lake (H). The elevation of the bed of the L4 emissary, where the ADV measurements were taken, is considered the reference elevation; the water depth in the outfall (h) and L4 (H) was assessed by subtracting the orthometric elevation of the bed of the L4 emissary from their geodetic elevation.

**Table 4.** Orthometric height of spot height DIATI 71, L4 gauging station measuring point and reference elevation.

|  | Orthometric height (m a.s.l.) |
| --- | --- |
| DIATI 71 | 2388.14 |
| Measuring point at L4 gauging station | 2386.50 |
| Reference elevation | 2386.12 |

The best fitting of the relation between the water depth measurements at the gauge $h$ and the Hydraulic head in the lake $H$ was found to be linear (H=1.3h-0.1279, $R^2 \sim 0.98$; see Figure 5(b)). The stage-discharge diagram (h-Q) and the linear fitting (h-H) were used to calibrate the lake outflow curve, i.e., the relationship between the hydraulic head ($H$) in the lake and the flowing discharge ($Q$) (Figure 5(c)).

## 2.6 Bedload monitoring

Quantitative sediment transport estimation in proglacial streams is challenging due to frequent geomorphic changes associated with snow cover/melt and glacier dynamics. A growing number of studies investigate the use of seismic techniques to obtain continuous, indirect measurements of bedload transport (e.g., Bakker et al., 2020; Coviello et al., 2018; Schmandt et al., 2013). Geophones installed near a stream channel detect seismic waves produced by two different seismic sources: coarse particles impacting on the channel bed and flow turbulence. We use a low-cost and easy-to-install geophone network to investigate the temporal variability of the hydro-sedimentary export from the snout of the Rutor glacier. Data are recorded with a DATA-CUBE3 (solar power supply, 24-bit converter, GPS-based time synchronization) configured with an amplifier gain of 16, with a sampling frequency of 200 Hz and stored on site. On 10 July 2021, we deployed a temporary monitoring network composed of three single-component geophones (4.5 Hz) installed along the proglacial stream draining the eastern tongue of the Rutor glacier. The geophones were installed a few meters from the right bank of the channel, about 200 m downstream of the glacier

snout (Figure 6). The monitored channel reach (main channel in Figure 6) features a wetted perimeter of about 10 m and a slope of 2°. An ephemeral stream channel crosses the area monitored by Geo 1, which is the sensor located at the smallest distance from the main channel, (about 3 m). This ephemeral stream is a tributary of the main channel and likely activates during intense rainstorm events. On the other side of the ephemeral stream are installed Geo 2 and Geo 3, at a distance of 6 m and 8 m from the main channel, respectively.

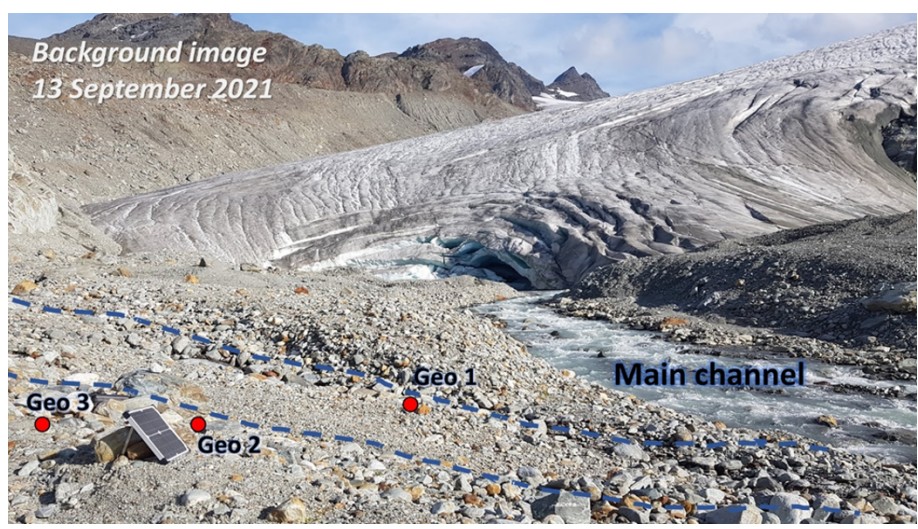

**Figure 6.** View on the monitored reach of the proglacial stream draining the eastern tongue of the Rutor glacier. Red dots indicate the location of the geophones, and the dashed blue line is the limits of the ephemeral stream flowing into the main channel.

The counts exported by the DATA-CUBE3 are converted to vertical ground velocity considering logger and geophone sensitivities according to the specifications of the manufacturer. The power spectral density is determined as the ratio of the square of the absolute value of the Fourier transform to the time window (Bakker et al., 2020). Raw seismic signals were filtered in the band 5-95 Hz and then the envelope was calculated as the average of the absolute value of the filtered signal over a time window of 1-min.

During the 2022 season, we performed direct measurements of bedload transport at the glacier mouth by means of portable samplers during a day of intense glacier melt (14 July) and at the end of the monitoring season (16 September). Bedload traps (4 mm mesh size, 20 × 30 cm opening, (Bunte et al., 2004)) were deployed simultaneously at 2 positions. Measured unit bedload rates feature a large variability ranging from 0.02 to 16.2 kg/m/min in a few hours, as already observed in glacierized basins (Coviello et al., 2022). Bedload samples were sieved and weighed to obtain the grain size distribution. The total bedload
transport rate $Q_s$ (kg/min above 4 mm) for each sampling period (ranging from 2 to 30 min) was estimated as width-weighted averages based on the available positions sampled.

## 3  Results

The dataset derived from the results presented in the following sections is also accessible in the WebGIS mentioned above through which it is possible to find the link to the open repository according to the location of the monitored/surveyed point.

### 3.1  Orthophotos and DSMs products

As described in section 2.3, both aerial and UAV photogrammetric data have been processed to generate orthoimages and DSMs to support glacier monitoring from a multidisciplinary perspective.

2021 UAV orthoimages and DSMs are characterized by a spatial resolution lower than 4 cm: the mosaic of such metric products provides a very detailed model of the area covering the path of the water melted from the eastern glacier tongue towards the proglacial lakes (see Figure 7, right). The spatial resolution of 2020 and 2021 aerial orthoimages is slightly lower (around 7 cm) compared to UAV ones, while the aerial DSMs have a resolution of about 20 cm. Figure 7 clearly shows the larger coverage of the aerial orthoimage (left) with respect to the UAV one (right). As previously mentioned, the limited coverage of the UAV products is the main reason why the elevation changes analysis is based on the aerial DSMs only. Nevertheless, UAV orthoimagery has been used (in addition to support the geophysical and hydraulic analyses) to assess the planimetric retreat of the Eastern glacial front. More specifically, a multitemporal orthoimage comparison was performed using the photogrammetric products described above and considering the following additional datasets, namely a) the 2012 color orthomosaic available as WMS through the Italian national geoportal (http://www.pcn.minambiente.it/mattm/, last access: 23 February 2024) and b) a Pléiades orthophoto acquired in August 2017. Figure 8 shows that the front is receding annually: the glacier tongue front has receded by more than 200 m in 9 years and about 100 m from 2017 to 2021.

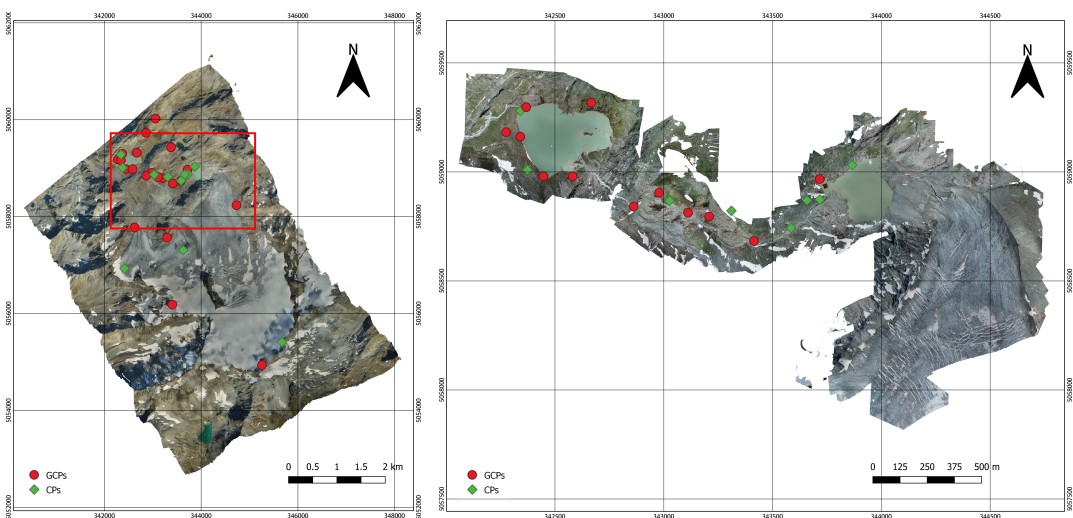

**Figure 7.** Aerial orthophoto as of September 2021 (left) and UAV high-resolution mosaic of orthophotos of $9^{th}$ and $21^{th}$ July 2021 (right)

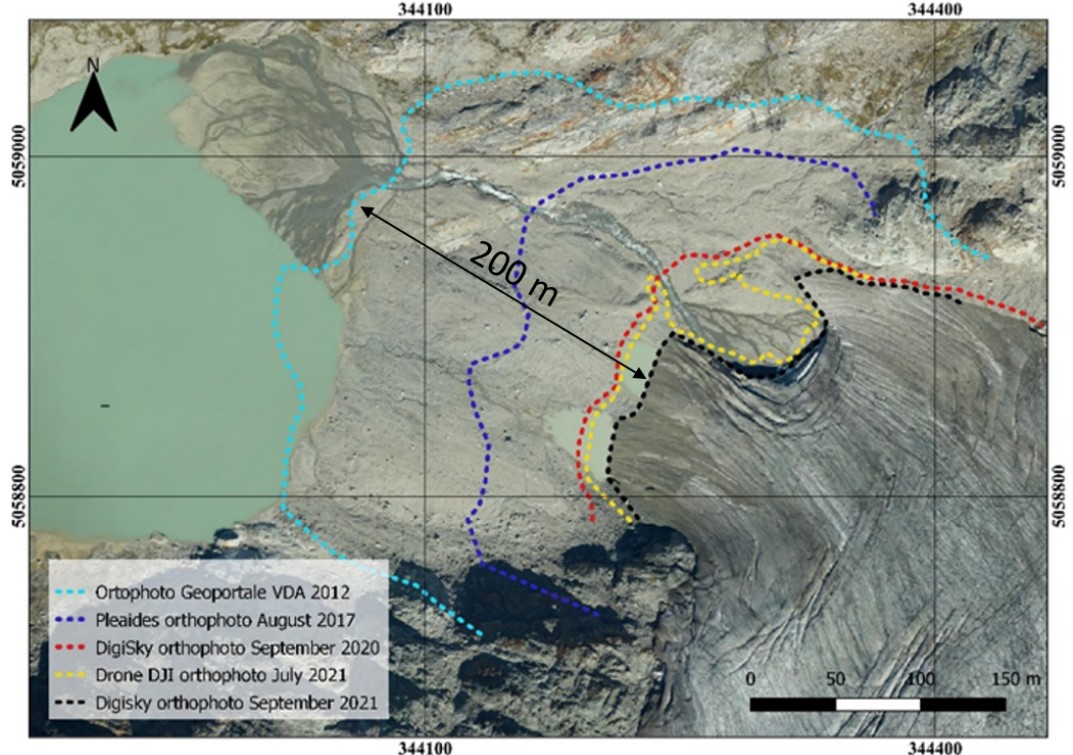

**Figure 8.** Multitemporal analysis of eastern glacial front retreat: front lines plotted on September 2021 ortophoto.

## 3.2 DoD analysis and LoD estimation

Glacier surface elevation differences were estimated by subtracting the 2020 aerial DSM to the 2021 one, to quantify glacier ablation and displacement (Figure 9). Using the approach described in section 2.3.1, means and standard deviations of the residuals $\Delta X$, $\Delta Y$, and $\Delta Z$ have been computed for both GCP and CP (Table 5), enabling the possibility to evaluate the planimetric and altimetric accuracies and precision of the photogrammetric products.

From Table 5, the mean value of $\Delta Z$ calculated on CPs (which is expected to be close to zero, being related to residuals) is not significant even at a 68% probability ($1\sigma$), as it is lower than the value of the $\Delta Z$ standard deviation. This suggests the absence of altimetric systematic errors for both DSMs. The planimetric error exhibits weak significance, which can be partially attributed to the precision of marker collimation. Given the good altimetric accuracy and precision achieved, it was not deemed necessary to make planimetric corrections. Referring to the previously described formulas, at a 68% probability ($1\sigma$), the value of the LoD for the Differences of Differences (DoD) is 22 cm.

$$LoD = \sigma_{DoD} = \sqrt{\sigma^2_{\Delta Z_{Aerial2020}} + \sigma^2_{\Delta Z_{Aerial2021}}} = \sqrt{18.0^2 \text{ cm}^2 + 12.6^2 \text{ cm}^2} = 22.0 \text{ cm} \tag{8}$$

**Table 5.** Means and standard deviations of the residuals ΔX, ΔY, and ΔZ calculated on GCP and CP. Values of interest for estimating the LoD of the DoD between aerial photogrammetric data from 2021 and 2020 are highlighted in bold.

| | n. GPPs | n. CPs | Residuals GCPs [cm] | | | Residuals CPs [cm] | | |
|---|---|---|---|---|---|---|---|---|
| | | | $\mu_{\Delta X} \pm \sigma_{\Delta X}$ | $\mu_{\Delta Y} \pm \sigma_{\Delta Y}$ | $\mu_{\Delta Z} \pm \sigma_{\Delta Z}$ | $\mu_{\Delta X} \pm \sigma_{\Delta X}$ | $\mu_{\Delta Y} \pm \sigma_{\Delta Y}$ | $\mu_{\Delta Z} \pm \sigma_{\Delta Z}$ |
| Aerial 2021 | 9 | 4 | $0.0 \pm 3.7$ | $0.0 \pm 3.5$ | $0.0 \pm 6.5$ | $-6.0 \pm 5.4$ | $-0.5 \pm 1.8$ | **4.5±12.6** |
| Aerial 2020 | 18 | 7 | $1.4 \pm 21.0$ | $-0.6 \pm 13.5$ | $0.1 \pm 10.9$ | $-3.6 \pm 20.1$ | $10.8 \pm 11.2$ | **-3.9±18.0** |
| UAV $9^{th}$ July 2021 | 8 | 6 | $-1.2 \pm 1.4$ | $-0.4 \pm 2.0$ | $0.8 \pm 1.8$ | $1.8 \pm 2.4$ | $1.4 \pm 4.0$ | -3.2±2.9 |
| UAV $20^{th}$ July 2021 | 5 | 2 | $-2.5 \pm 2.7$ | $-9.7 \pm 22.7$ | $4.2 \pm 8.6$ | $1.5 \pm 0.4$ | $0.9 \pm 0.6$ | 1.3±1.2 |

In the assumption of normality of the distribution, it becomes 44 cm ($2\sigma$) at a 95% probability.

$$LoD_{95\%} = 2\sigma_{DoD} = 44 \, \text{cm} \tag{9}$$

We are aware that this value is derived from a limited sample of CPs (Table 5, second column). However, practically speaking, it is not feasible to increase the number of points significantly due to the difficulties, time constraints, and hazards of a high mountain environment, as previously described. Even with considerable time and effort, the number of CPs could only vary by a few units.

Neverthless, we also tested the stable terrain-based approach, identifying three polygons (outlined in black in Figure 9(a)) corresponding to an approximately 1 km$^2$ area downstream of the glacier fronts. These polygons were identified through the photointerpretation of aerial orthophotos and field experience to exclude local variations in the terrain, specifically those related to fluvial dynamics (all periglacial lakes and water-covered surfaces were excluded). As outlined, the statistical calculations (outlined in Table 6) were conducted both across the entire dataset and with exclusion of the top and bottom 5% of the distribution to mitigate any potential outliers. The mean and root mean square values computed over 95% of the dataset (approximately 45 million points) exhibit a marginal decrease compared to those calculated across the entire dataset. This suggests accurate identification of stable areas and limited presence of outliers, a notion further supported by the similarity between mean and median values (within a range of approximately 4 cm). Furthermore, the Fisher indices $\gamma$ and $\gamma_2$, which quantify skewness and kurtosis respectively, converge towards zero (with a probability close to 1), indicating a distribution of the DoD that approximates normality. Excluding the top and bottom 5% of the distribution and referring to the previously described formulas, at a 68% probability ($1\sigma$), the value of the LoD for the DoD is 18 cm. Having confirmed the normality of the distribution, it becomes 36 cm ($2\sigma$) at a 95% probability:

$$LoD_{95\%} = 2\sigma_{DoD} = 36 \, \text{cm} \tag{10}$$

**Table 6.** Statistical analysis of DoD distribution between aerial DSMs for the years 2021 and 2020 in stable terrain.

| | DoD on stable terrain $DSM_{Aerial2021} - DSM_{Aerial2020}$ |
|---|---|
| Median (entire sample) | 0.012 m |
| $\mu_{DoD} \pm \sigma_{DoD}$ (entire sample) | -0.028 m $\pm$ 0.240 m |
| $\mu_{DoD} \pm \sigma_{DoD}$ (without tails 5%) | -0.025 m $\pm$ 0.182 m |
| Skeweness – Fisher $\gamma$ (without tails 5%) | 0.356 |
| Kurtosis – Fisher $\gamma_2$ (without tails 5%) | -0.968 |

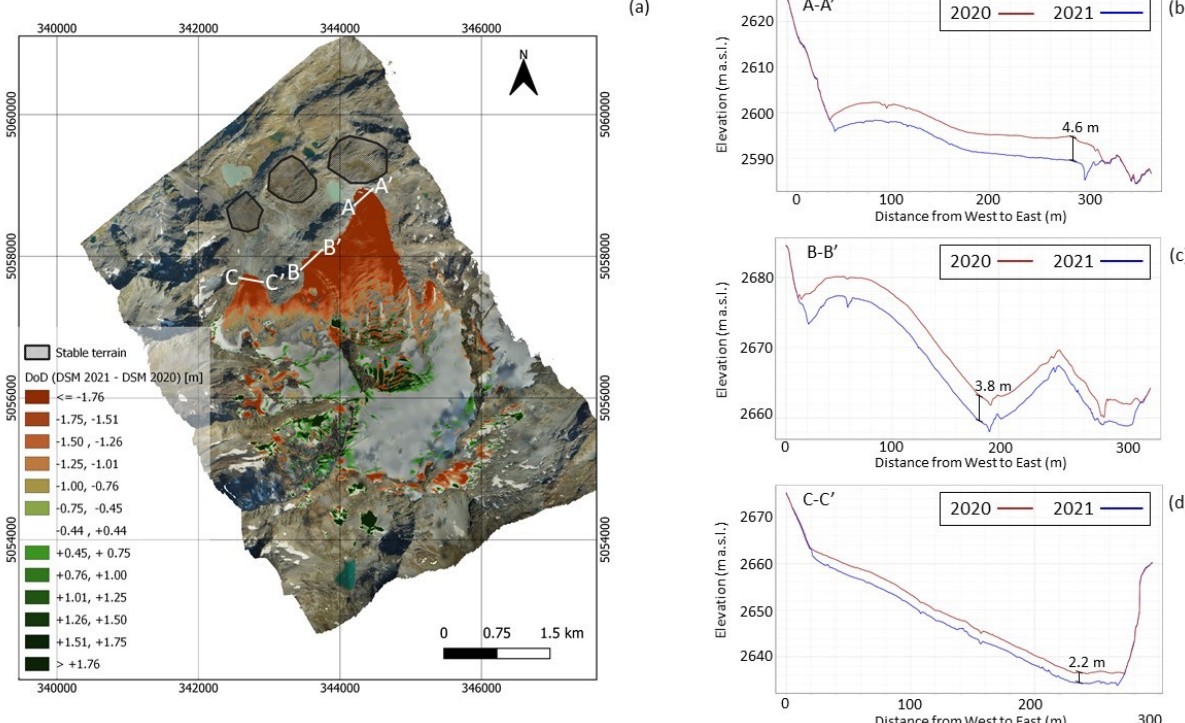

**Figure 9.** (a) 2021-2020 DoD. The white lines refer to the cross-sections A-A', B-B', and C-C', whose 2020 (red) and 2021 (blue) elevation profiles are shown in panels (b), (c), and (d) respectively.

Taking a cautious approach and recognizing the challenge of identifying uniformly distributed stable areas across the entire surveyed region, this study has opted to use the LoD value derived from CP analysis, which stands at 44 cm (slightly higher than that obtained from stable areas). It is worth noting, however, that if representative stable areas covering the entire monitored region could be identified, the corresponding estimation of the LoD for the DoD would be considered more robust than the CP-based approach, as it would be derived from a highly representative sample of the DoD population over an area of approximately 1 km$^2$. The DoD derived from the photogrammetric flights of 2021 and 2020 are illustrated in Figure 9(a) using

color intervals. To ensure clarity, a color scale with 25 cm intervals (approximately matching the DoD precision) was selected. Notably, values falling within the range of -44 cm to +44 cm are omitted from the figure, as they fall within the LoD for the

425 DoD calculated based on CPs. It is apparent that the phenomenon of annual melting is less pronounced at higher elevations, where snow accumulation persists (indicated by positive DoD values in greenish colors), and becomes more prominent downstream, particularly at the glacier fronts (indicated by negative DoD values in reddish colors). These differences can reach more than 4 meters, as also demonstrated by profile A-A' traced along the eastern glacier front (see Figure 9(b)).

### 3.3 Bathymetry and lake bed sediment distribution

The outcome of the GPR survey is a series of georeferenced x-depth sections of the lake. The radar reflections depict two main interfaces: the "water - fine sediments" interface, which represents the lake bottom, and a second deeper interface, which separates the fine sediments from the underlying ground layer. In the example in Figure 10, the first (lake bottom) interface starts at near 0 depth at the left border of the picture, deepening until 250 ns (3.5 m) in the centre of the picture, and then ascending to 0 depth in the right part. The average water depth of the lake was 3.9 meters and the maximum depth was around

11 meters in July 2021.

The deeper interface in Figure 10 is fairly distinguishable and runs parallel to the first interface, deepening until 350 ns. In the left and centre of the image, the sediment thickness is more than double compared to the right part of the image. Under the second interface, many sparse reflections are visible, thus the underlying layer is probably not formed by compact rock, but by coarse debris or sediments. The second deeper interface was interpreted as the bottom of the fine sediment layer. To convert

the radar two-way travel times to the thickness of this layer, we needed an estimation of the signal propagation velocity in the fine sediments.

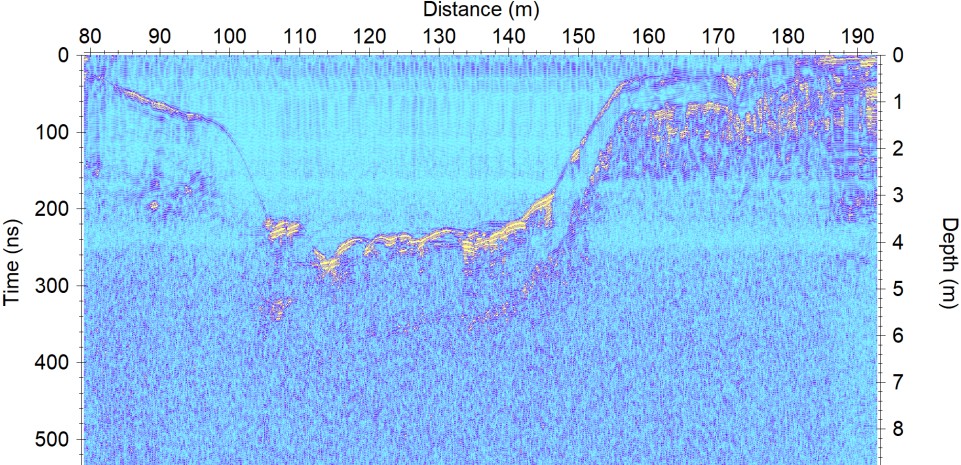

**Figure 10.** Example of a GPR section of the Seracchi Lake. Relevant reflections are the water – fine sediments interface and, deeper, the fine sediments - coarse sediments interface).

The TDR probe measured a fairly uniform average relative electrical permittivity of 36 +- 3, which was converted to a propagation velocity of about 0.05 m/ns. Similarly to bathymetry, an interpolation process produced a final map of the thickness distribution. Figure 11 shows that a major sediment accumulation has happened in the zones near the glacier inflows (from the Southeast). Aside from this, the fine sediment layer is quite homogeneously distributed all around the lake, with an average of 1.6 meters of thickness. Unfortunately, the zones where the water was deeper than 6-7 meters could not be penetrated with sufficient energy and the second interface was lost. This is the main limitation of the GPR survey, which restricts its range of applicability to other proglacial lakes, is the depth of investigation. We expect that after 15 meters of depth even the first interface could not be detected anymore, in similar conditions (200 MHz antenna, low-conductivity water). The TDR probe

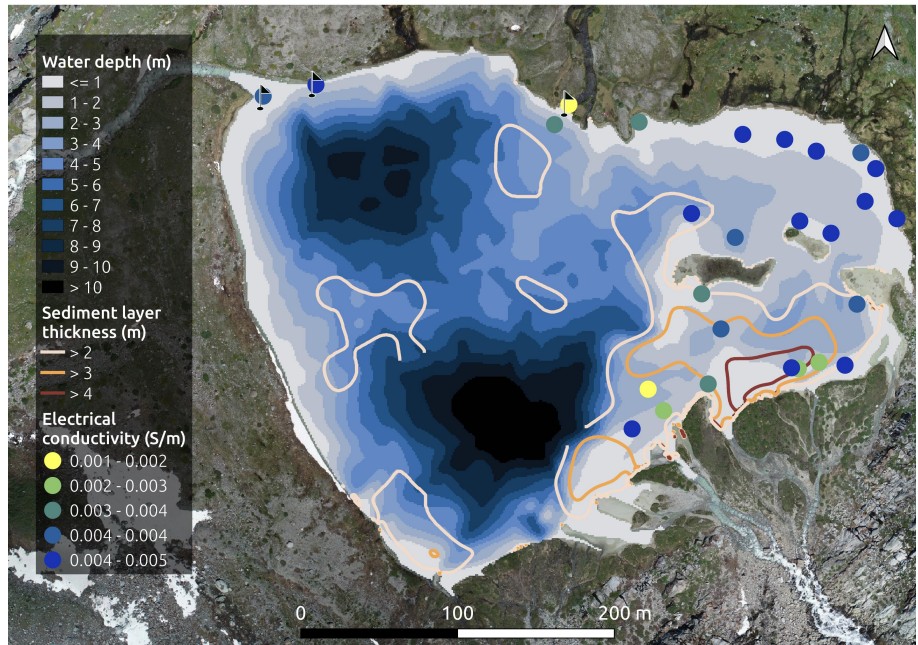

**Figure 11.** Results of the GPR and TDR geophysical survey. In a blue colour scale, the bathymetry of the lake. The brown contour lines indicate the areas where the sediment layer is thicker (in particular near the inflows from the glacier). The yellow-to-blue points indicate the TDR measurements of electrical conductivity. The electrical permittivity is not shown here but it is fairly uniform (average = 36). The three black flags indicate the locations of manual sediment samplings. Colour scale according to Crameri et al. (2020)

measured, other than the permittivity, also the electrical conductivity of the sediments. The locations of measurements, which also correspond to the permittivity measurements, are shown in Figure 10. This property had a uniform value, except in small areas near the inflows. This means that the type of sediment in those zones is different from the rest of the lake. Thanks to sediment sampling (locations in Figure 10) and grain size distribution analysis, together with the electrical conductivity distribution, we reconstructed that the fine sediment layer is fairly uniform around the lake and contains around 50 % of clayish-sized material, while near the inflows there is coarser gravel because the flow velocity does not allow the fine particles to sediment.

### 3.4 Hydrometric monitoring

The investigation at the L4 gauging station involved: i) a set of 9 velocity-based discharge measurements which allowed the stage-discharge diagram (h-Q) to be assessed; ii) a set of 15 elevation difference measurements which led to the linear fitting (h-H) and the lake outflow curve (H-Q) (Figure 5).

Thanks to these results it has been possible to reconstruct the high-resolution (10-minute acquisition time) temporal sequence of discharge flowing from the lake and primarily driven by the glacier melt. Figure 12(a) shows this temporal sequence.

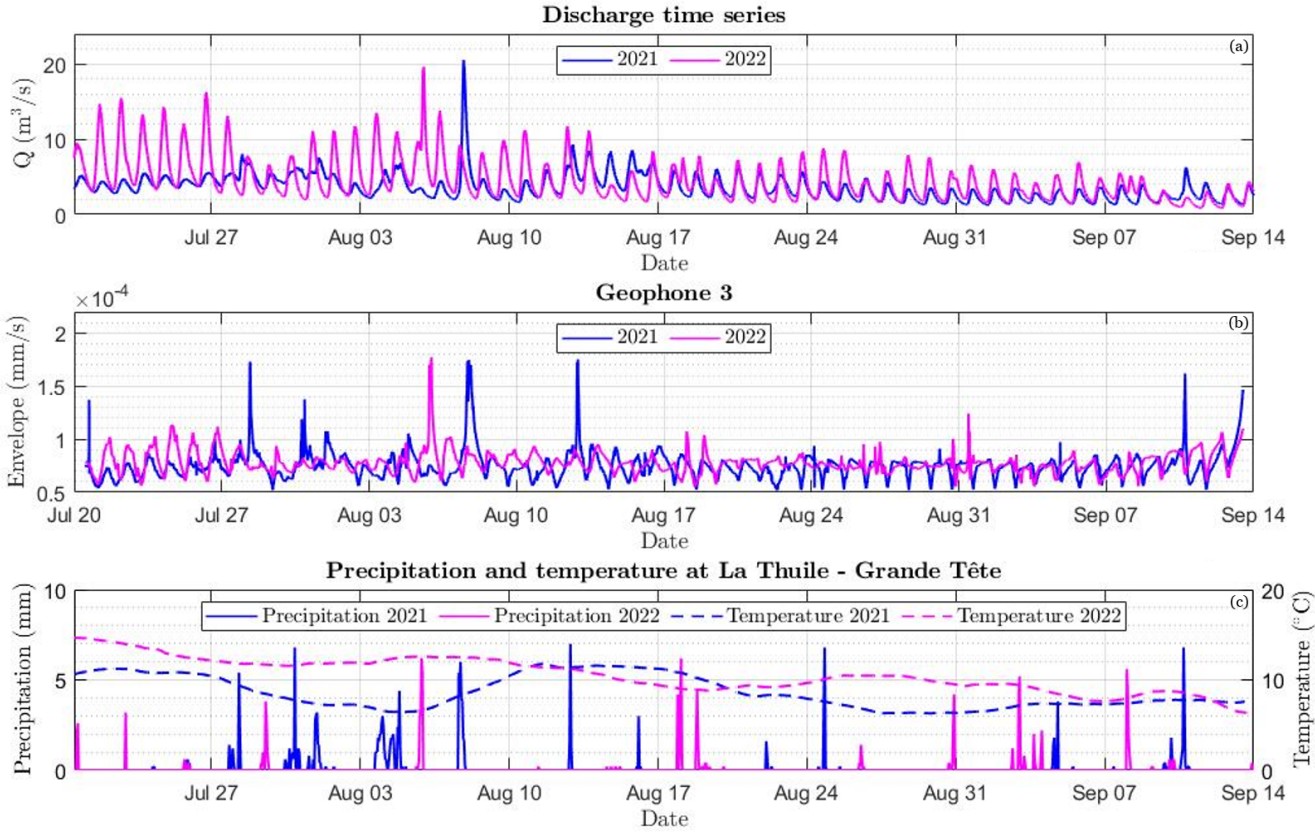

**Figure 12. (a)** Discharge time series in L4 emissary in 2021 and in 2022. **(b)** Geophone 3 signal envelope in 2021 and in 2022 calculated on a time window of 1-min .**(c)** Daily precipitation (solid line) and 10-day moving mean of air temperature (dashed-dotted lines) measured at La Thuile-Grande Tête weather station in 2021 and in 2022.

Using meteorological data from the Grande Tête weather station managed by ARPA Valle d'Aosta (data available at https://presidi2.regione.vda.it/str_dataview_download, last access: 19 January 2023), we observed that water level and water temperature are strongly correlated with air temperature, and this correlation is higher in summer. The summer of 2022 has been warmer than that of 2021. Consequently, the average flow measured in July and August in the L4 outfall in 2022 is higher

than that measured in 2021 for the same period (by as much as 26%). The difference between the amplitude of the water level fluctuation in 2022 and 2021 is more pronounced in early summer (Figure 12(a)), due to the different air temperature in May, which was on average 5°C higher in 2022 than in 2021 (Figure 12(b)), and to an earlier discharge of meltwater in 2022 than in
the previous year.

The water discharge caused by glacier melt has a strong daily periodicity driven by solar radiation and thermal energy, perturbed occasionally by rainfall events. Unlike the contribution of glacier melt to water discharge, the contribution of rainfall is not periodic within the day, thus altering the otherwise daily periodic flow pattern in glacier-fed watercourses. The auto-correlation functions of the water depth time series measured at L4 and L2 highlight the daily periodicity that is strongly
related to the glacier melt (Figure 13 (a) and (b)). However, the amplitude of these functions in 2021 is smaller than in 2022 and their daily means cross the zero axis earlier (about 5 days in L4 and 2 days in L2). This fact can be attributed to the different sizes and numbers of precipitation events in the two years (Figure 12). Rainfall in 2021 was more frequent than in 2022, and the July-August cumulative rainfall was 238,6 mm and 82,6 mm in 2021 and 2022, respectively. Accordingly, as the frequency of precipitation increases, the auto-correlation function decreases.

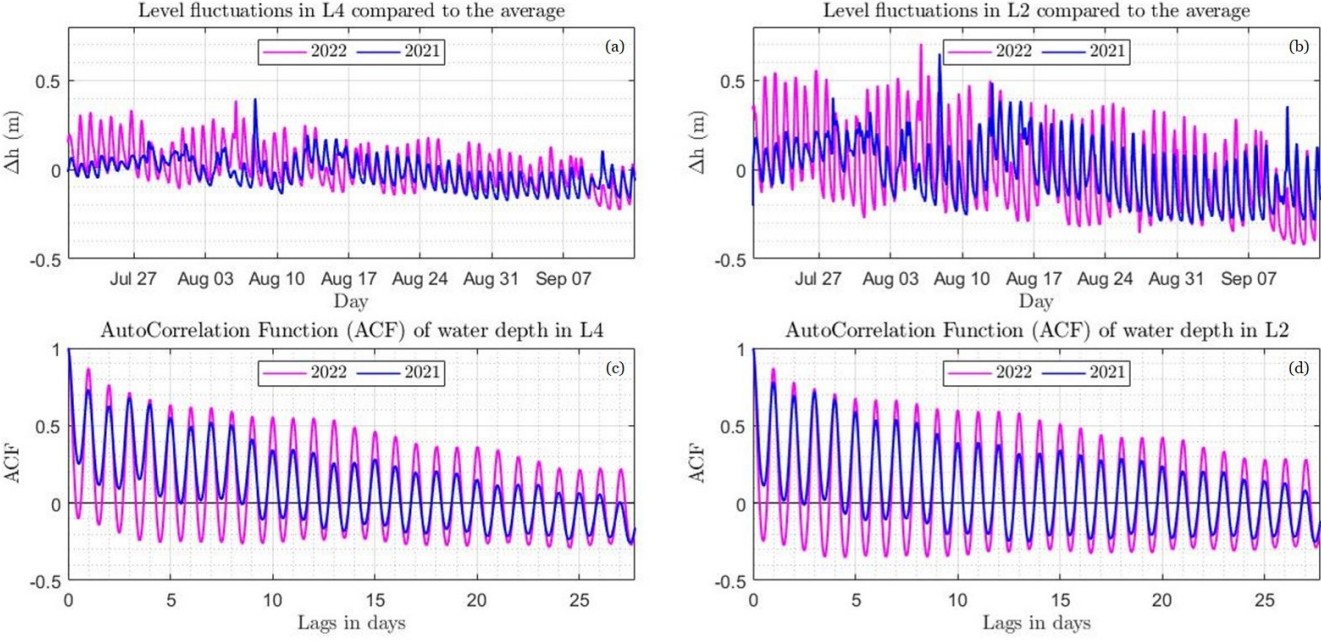

**Figure 13.** Level fluctuations recorded in 2021 (blue) and 2022 (magenta) by measuring stations L4 **(a)** and L2 **(b)** and the corresponding autocorrelation function in the panels **(c)** and **(d)** respectively.

 **3.5    Bedload monitoring**

Preliminary results show how an array of single-component geophones installed close to the flow path can detect both daily and longer-period fluctuations in bedload and water flow. The geophone signal mirrors well the flow of daily cycles with fluctuations within a period of 24 hours and permits the identification of time intervals characterized by intense transport (Figure 12b). Results highlight the signal fluctuations and suggest that intense runoff with bedload transport occurred during specific days (i.e., 13 July, 28 July, 4 August, 7 August, and 12-13 August). The larger flood event was detected on 7 August 2021 (Figure 14), during which a marked increase of the seismic power was observed (i.e., one order of magnitude) compared to time periods characterized by low water flow and no bedload transport.

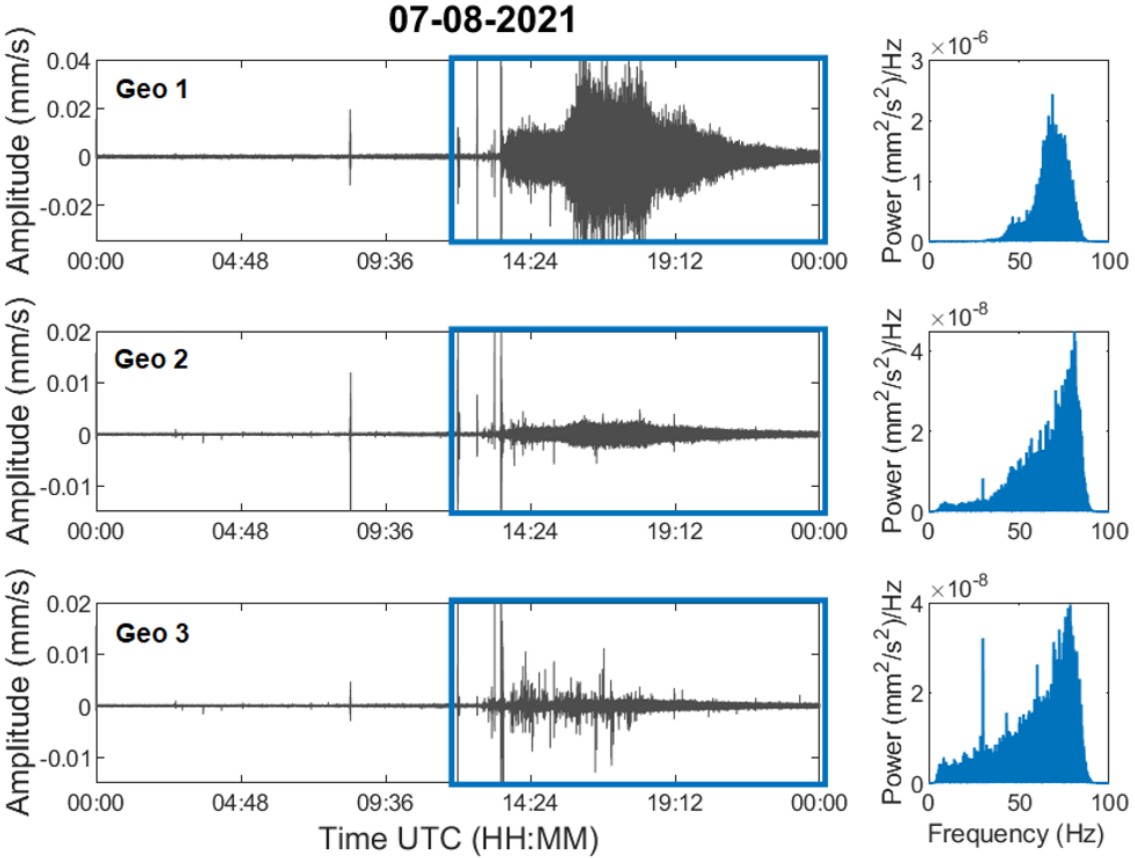

**Figure 14.** Waveforms recorded on 7 August 2021 and power spectra of a specific portion of the signal (blue boxes, from noon to midnight UTC).

It is assumed that the geophone signal (Figure 12b) permits the identification of time intervals characterised by intense transport since, in correspondence with the peaks of the envelope, the power increases for high values of frequency (Figure 14).

Indeed, the power in the lower bands is attributed to turbulent fluid flow (Schmandt et al., 2013) while that in the higher bands to bedload (Schmandt et al., 2013; Bakker et al., 2020).

In 2021, we observed, through direct inspection of the flow field, the absence of bedload transport in three days (10 July, 20 July and 13 September). The dataset of direct measurements will be expanded in 2023 and used to calibrate the seismic data and extract quantitative information on the bedload export from the glacier.

## 495  4   Data availability

Eight different datasets were produced. These datasets are listed below and accessible in a WebGIS (available at https://arcg. is/Tyeju0, last access: 23 February 2024) through which the link to the open archive can be found according to the location of the monitored/surveyed point:

- The orthophotos and DSMs database related to the 2020 aerial survey is available on the Zenodo repository at https: //zenodo.org/records/8089499 (Corte et al. (2023d));

- The orthophotos and DSMs database related to the 2021 aerial survey is available on the Zenodo repository at https: //zenodo.org/records/10100968 (Corte et al. (2023b));

- The orthophotos and DSMs database related to the 2021 UAV survey is available on the Zenodo repository at https: //zenodo.org/records/10074530 (Corte et al. (2023b));

- The Rutor glacier surface area database obtained from the orthophoto of September 2021 is available on the Zenodo repository at https://zenodo.org/records/10101236 (Corte et al. (2023b));

- The footprints of the various glacial fronts obtained from the elaborated cartographic products database is available on the Zenodo repository at https://doi.org/10.5281/zenodo.7713146 (Corte et al. (2023c))

- The bathymetry and sediment thickness of L4 database is available on the Zenodo repository at https://doi.org/10.5281/ 510      zenodo.7682072 (Corte et al. (2023a));

- The dataset of the water depth measured by the instrument installed at gauging stations L1, L2, L3, and L4 and the relationship between the water depth and the wetted area at gauging station L4 is available on the Zenodo repository at https://zenodo.org/record/7697100 (Corte et al. (2023e));

- The geophones monitoring database is available on the Zenodo repository at https://doi.org/10.5281/zenodo.7708800 515      (Corte et al. (2023f)).

The images used for the creation of the photogrammetric products used for the DoD analysis are not available in the datasets due to licensing restrictions. The ownership of the images is not exclusive to the authors but is shared with the organization responsible for their acquisition, Digisky. For this reason, it is not possible to share this dataset as well. To obtain the images,

the purposes of the request need to be analyzed, and an agreement on their use must be signed directly with the authors and Digisky. We are open to sharing the dataset if it is agreed upon with Digisky.

Our objective is to increase and update the dataset by continuing to monitor and survey the Rutor glacier and its proglacial area over the years through this multidisciplinary approach.

## 5 Conclusions

At present (to the best of our knowledge) appears to be a lack of studies in the literature on proglacial areas involving multitemporal geospatial surveys with continuous monitoring of melt water runoff during the ablation period, which merge the contributions of different disciplines. At the same time, very few cases exist of continuous monitoring of streamflow at high frequency and high altitude.

In this work, a multidisciplinary and multitemporal approach was presented to characterise the Rutor glacier and its proglacial area. Multidisciplinary analyses are fundamental in the study of complex environment. It is instructive to summarise some direct examples of the synergies involved in a multidisciplinary approach for the investigated area. Firstly, the comparison of multitemporal 3D geospatial data (taking the related DoD LoD into account) determined that the eastern tongue is losing mass faster than the others, leading to the intensification of measurements at L1 and nearby the eastern tongue of the glacier. Secondly, the orthoimage based on the photogrammetric UAV surveys carried out at the same time as the geophysical survey, enabled the accurate extraction of the lake perimeter, which - integrated with the data acquired from the GPR - resulted in an accurate bathymetry of the lake and allowed to get the exact outline of the zero-depth points at the time of the investigation. Thirdly, continuous hydraulic monitoring at the L4 gauging station and the relationship between the water depth measured by the sensor and the depth of the lake provided the volume change of L4 over time. In addition, combining the bathymetry map with the DSM of the surrounding area will enable the determination of the water volume of L4 when the water level is higher than at the time of the geophysical survey. Lastly, the extracted products of the crewed aerial photogrammetric flights allowed the Environmental Agency (ARPA VDA) to develop the mass balance for the hydrological years under consideration. The comparison of different DSMs sets the basis for continuous monitoring over time, in which the 2021 model will serve as a reference for future comparisons. The mass balance of the Rutor glacier can also be determined through the application of a hydrological model calibrated with the water discharge time series obtained from this study. It is important to stress that the accurate georeferencing of all the acquired data with respect to the same datum plays a crucial role in the data integration phase and in enabling multitemporal analyses.

Future modelling of the water flow and sediment transport in L4 may be based on the bathymetry map combined with the inflow and outflow measurements. The GPR and TDR surveys, with a few ground-proof sediment sampling, evidenced that in about 140 years since the birth of the lake, a fine sediment layer thick 1.6 m on average was deposited on the lake bottom. Sediment transport deserves further investigation, because it may change due to the rapid shrinking of the Rutor glacier, whose bedrock erosion is the source of the fine sediment found in the lake. An approach to model these changes could involve temporal

monitoring of water turbidity as a proxy of the concentration of suspended sediment in the various inflows and outflows of the interconnected water bodies.

The multidisciplinary approach and the dataset herein presented enable the characterisation, monitoring, and understanding of a set of complex processes that take place in the studied area, allowing the authors to shed light on interconnected phenomena with a broader perspective than a single scientific discipline approach. Indeed, the results of a combined effort often go beyond the sum of each contribution.

## Appendix A: Stage-discharge relationship for L4

The procedure followed to measure the velocity is reported in ISO 748:2007. The methods used to determine the discharge from current-meter measurements are classified in ISO 748:2007 as the graphical method and the arithmetic method. The latter, which is more suitable for computations carried out in the field, includes two methods: the mean-section method and the mid-section method. The discharge was determined by applying both arithmetic methods and averaging the two results.

The power law is the one that best represents the stage-discharge measurements (Figure 5):

$$Q = 12.118 \times h^{4.0042}, \quad R^2 = 0.925. \tag{A1}$$

The lowest water discharge measured corresponds to a water depth in the cross-section of 0.49 m. In fact, the power law describes well the $Q - h$ relationship when $h$ is greater than half a metre. For shallower water depths, the power law returns a flow rate that is too low for the geometry of the cross-section considered. Consequently, for $h < 0.49$ m, the stage-discharge curve was obtained by taking into account the geometry of the cross-section.

Water discharge can be written as a function of the wetted area of the cross-section:

$$Q = k \times \Omega^m, \tag{A2}$$

where $Q$ is the discharge, $k$ is a flow resistance coefficient, $\Omega$ is the wetted area and $m$ is a coefficient dependent on the cross-section geometry. To obtain the expression of the coefficient $m$, the stage-discharge relationship and Chézy equation were expanded using the Taylor series and set equal each other, thus obtaining:

$$m = \frac{5}{2} - \frac{2}{3} \frac{\Omega_0}{B_0} \left( \frac{dB}{d\Omega} \right)_{\Omega_0}. \tag{A3}$$

This coefficient depends on the wetted area ($\Omega$) and the wetted perimeter ($B$) of the cross-section.

The geometry of the cross-section of L4's emissary was determined through an RTK survey. The measurements of the three coordinates of the points within the cross-section bed were with steps of about 20 cm along the cross direction.

When visualizing the cross-section geometry and the curve that describes how the wet perimeter changes with the wet area (Figure A1), it is clear that two different stage-discharge relationships, corresponding to two different water depth intervals,

**Figure A1.** The wetted area and the corresponding wetted perimeter for water depths between $h = 0$ and $h = 52$ cm.

must be considered:

$$Q = k_1 \times \Omega(h)^{m_1}, \quad 0 \text{ m} \leq h \leq 0.34 \text{ m} \tag{A1}$$

$$Q = k_2 \times \Omega(h)^{m_2}, \quad 0.34 \text{ m} \leq h \leq 0.49 \text{ m.} \tag{A2}$$

For each interval, the coefficient $m$ was calculated as the mean of all the values determined at each point within the corresponding interval. Considering the first interval from $h_1$ to $h_n$ and the second from $h_{n+1}$ to $h_l$, the coefficients were calculated according to:

$$m_1 = \frac{1}{n} \sum_{i=1}^{n} \frac{5}{2} - \frac{2}{3} \frac{\Omega_i}{B_i} \left( \frac{dB}{d\Omega} \right)_{\Omega_i}, \quad 0 \text{ m} \leq h \leq 0.34 \text{ m} = 1.487 \tag{A3}$$

$$m_2 = \frac{1}{l-n} \sum_{i=n+1}^{l} \frac{5}{2} - \frac{2}{3} \frac{\Omega_i}{B_i} \left( \frac{dB}{d\Omega} \right)_{\Omega_i} = 1.0609, \quad 0.34 \text{ m} \leq h \leq 0.49 \text{ m.} \tag{A4}$$

The $k$ coefficients were calculated by imposing the continuity stage-discharge relationship respectively at $h = 0.34$ cm and $h = 0.49$ cm, thus obtaining $k_1 = 0.232$ and $k_2 = 0.257$. The definitive stage-discharge relationship is given by three different relationships corresponding to three different water depth intervals:

$$Q = 0.232 \times \Omega^{1.487}, \quad 0 \text{ m} \leq h \leq 0.34 \text{ m} \tag{A5}$$

$$Q = 0.257 \times \Omega^{1.069}, \quad 0.34 \text{ m} \leq h \leq 0.49 \text{ m} \tag{A6}$$

$$Q = 12.118 \times h^{4.0042}, \quad h \geq 0.49 \text{ m.} \tag{A7}$$

## Appendix B: Normality testing - Skewness and Kurtosis

Following this operation, it is possible to verify whether the median, a "robust" estimator", and mean of the DoD are similar (an indicator of the absence of outliers) and assess the normality of the distribution. The test for normality of a sample of observations can be conducted by considering the kth-order moments relative to the mean of the deviation variable:

$$m_{k,m} = \int_{-\infty}^{\infty} (x - m)^k f(x) dx \tag{B1}$$

With k=3, the moment is termed Skewness, and with k=4, it is called Kurtosis. They provide information on the asymmetry and peakedness of the distribution under examination relative to the normal distribution. With these, we can define the Fisher indices by considering:

- Skewness coefficient:

$$\gamma = \frac{m_3}{m_2^{\frac{3}{2}}} \tag{B2}$$

- Kurtosis coefficient:

$$\gamma_2 = \frac{m_4}{m_2^2} - 3 \tag{B3}$$

If the distribution approaches normality, these coefficients tend towards zero. We can evaluate their proximity to zero by testing the z-scores, normalized with respect to their standard error. The null hypothesis assumes that the gamma indices are equal to zero. For the text concerning skewness, let us consider:

$$z_1 = \frac{\gamma}{s_1} = \frac{\gamma}{\sqrt{\frac{6}{n}}} \tag{B4}$$

For the test concerning kurtosis, let us consider:

$$z_2 = \frac{\gamma_2}{s_2} = \frac{\gamma_2}{\sqrt{\frac{24}{n}}} \tag{B5}$$

At the normalized z-value, we can search for the probability of accepting the null hypothesis on tables of the normal distribution.

## Appendix C: GPR data processing

The GPR profiles were processed in Reflexw software (Sandmeier, 2021) according to the following processing steps:

– "Move startime", to delete the data acquired before the radar impulse transmission. To recognize automatically the timing of the transmitted impulse, the ReflexW processing "Correct max phase" was used.

    – "Make equidistant traces", to make every trace distant 0.1 m between each other, to compensate for the variable speed of the boat.

    – "subtract mean - dewow", a filter that subtracted to each trace the average of that trace, on a 5-ns timewindow, to correct
for instrumental voltage shifts.

    – "Bandpass butterworth", a bandpass filter which cut the frequencies lower than 50 MHz and higher than 260 MHz

    – "subtracting average", a filter that subtracts the average trace from each trace, on a 100-traces window, to eliminate horizontally coherent noise.

    – "energy decay", a simple time-axis gain function which equalizes the energy of the traces, which generally decreases
over time.

*Author contributions.*

Conceptualization & Supervision (AC, CC, AG, ST); Investigation (All authors); Data Curation (EC, VC, MMM, AV); Visualization (AA, EC, MMM, FGT, AV); EC prepared the manuscript with contributions from all co-authors.

*Competing interests.*

The authors declare that no competing interests are present

*Acknowledgements.* We thank Umberto Morra di Cella (ARPA - Valle d'Aosta) for the collaboration and interesting discussions, and Francesco Comiti and Matthias Bonfrisco (Free University of Bozen) for supporting bedload measurements. The research has been funded by the European Union with the Alcotra-Interreg project R.I.T.A and the Italian Ministry for Education and Research (MIUR), through the project "Department of Excellence".

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
