# Peer review of "Multitemporal characterisation of a proglacial system: a multidisciplinary approach."

_Earth System Science Data, 2023_

## Referee Comment (RC2)

[referee-annotated manuscript omitted]

---

## Author Comment (AC1)

We wish to thank Referee 1 for the careful reading and the valuable comments.

We provide below a summary of the key changes implemented and a point-by-point response to all the raised queries (in italic).

**General comments**

**a. Section 2.2.1 Geomatic survey:** *The section does not provide a detailed explanation of the photogrammetric processing, and it is not supported by the relevant literature. In my opinion, this section needs to be improved. For instance, the name "SfM-MVS photogrammetry" is never mentioned throughout the article and no effort is made to explain the steps needed in a rigorous photogrammetric study. A few examples: the authors do not mention how the images were collected (e.g., drone flight geometries), how many ground control points they used, or which software or freeware they used to process the images. Finally, the author do not provide any information on the quality of the DSMs, therefore questioning if their results are reliable or not. This important weakness needs to be addressed.*

We are grateful to the reviewer for the specific feedback. Section 2.2.1 will be integrated with all required details about the photogrammetric workflow supported by the relevant literature.

Additional details on the Drone and Crewed Aerial Photogrammetric surveys will be provided and an updated Table 2 will be integrated in the article, detailing the number of Ground Control Points (GCP) and Check Points (CP). Line 138 will be updated adding a remark about the difficulty of retrieving or positioning/measuring control point coordinates in glacier environments, as in the following link: https://zenodo.org/record/8089499 .

| Photogrammetric flight | Date of acquisition | Covered area | Extent (km$^2$) | Average flight height (m) | GSD (m) | Number of images | Number of GCPs | Number of CPs |
|---|---|---|---|---|---|---|---|---|
| Aerial | 30/09/2020 | Glacier and a portion of the proglacial area | 25.2 | 818 | 0.07 | 867 | 18 | 7 |
| Drone | 9/07/2021 | L1, L2 and L4 | 2.6 | 126 | 0.03 | 1480 | 8 | 6 |
| Drone | 20/07/2021 | L1 and L4 | 0.4 | 89.2 | 0.02 | 369 | 5 | 2 |
| Drone | 20/07/2021 | Glacier front and lower part | 1.1 | 159 | 0.04 | 623 | Direct georeferencing | |
| Aerial | 13/09/2021 | Glacier and proglacial area | 34.5 | 877 | 0.06 | 1100 | 9 | 4 |

**b. Section 2.2.4 Bedload monitoring:** *As per the geomatics, I believe there is the need of providing a more detailed explanation of the bedload monitoring since the use of seismometers in bedload*

*studies is relatively recent. It would be very useful to know in more detail how the data were processed and the steps required to go from the raw signal to the results presented here.*

We are thankful to the reviewer for the general and detailed comments. As requested, section 2.2.4 will be modify as follows:

- the sentence on line 252 "Data are recorded with a DATA-CUBE3 (solar power supply, 24-bit converter, GPS-based time synchronization) with a sampling frequency of 200 Hz and stored on site." will be changed to "Data are recorded with a DATA-CUBE3 (solar power supply, 24-bit converter, GPS-based time synchronization) configured with an amplifier gain of 16, with a sampling frequency of 200 Hz and stored on site."
- in line 261 we will add "The counts exported by the DATA-CUBE[3] are converted to vertical ground velocity considering logger and geophone sensitivities according to the manufacturer's specifications. The power spectral density is determined as the ratio of the square of the absolute value of the Fourier transform to the time window (Bakker et al., 2020)."

In this way, we believe that the procedure is clearer and the added reference should provide all details explaining the steps in data processing.

**c. Data availability:** *The 2020 orthophoto and DSM are not available on Zenodo (https://zenodo.org/record/7713299). Are you going to include them in future?*

Thanks for pointing out that also the availability of 2020 metric products could be valuable for the readers. They have now been uploaded to Zenodo at the following link: https://zenodo.org/record/8089499
* * *
**Detailed comments:**

**1. Lines 26 – 28:** *"Alpine glacier retreat is leading to increased exposure of formerly glaciated terrain, entailing the colonization of plants and animals, and changes in morphodynamics and sediment transfer."*

*Consider adding one or more citations here.*

We will add the following 3 citations:

- Mainetti A, D'Amico M, Probo M, Quaglia E, Ravetto Enri S, Celi L and Lonati M (2021) Successional Herbaceous Species Affect Soil Processes in a High-Elevation Alpine Proglacial Chronosequence. Front. Environ. Sci. 8:615499. doi: 10.3389/fenvs.2020.615499
- Brambilla, M. and Gobbi, M.: A century of chasing the ice: delayed colonisation of ice-free sites by ground beetles along glacier forelands in the Alps, Ecography, 37, 33–42, https://doi.org/https://doi.org/10.1111/j.1600-0587.2013.00263.x, 2014

**2. Lines 29 – 30:** *"Little Ice Age (LIA)"*

*You already defined the acronym; perhaps just use LIA instead of "Little Ice Age (LIA)".*

We will update the manuscript and use LIA instead of "Little Ice Age (LIA)".

**3. Lines 35 – 36:** *"On the one hand, plant colonization stabilizes glacial sediment and reduces sediment fluxes; on the other hand, geomorphic processes disturb and limit vegetation succession."*

*Consider adding one or more citations here.*

We will add the following 3 citations:

-   Eichel, J.: Vegetation Succession and Biogeomorphic Interactions in Glacier Forelands, pp. 327–349, Springer International Publishing, Cham, https://doi.org/10.1007/978-3-319-94184-4_19, 201
-   Moreau, M., Mercier, D., Laffly, D., and Roussel, E.: Impacts of recent paraglacial dynamics on plant colonization: A case study on midtre Lovénbreen foreland, Spitsbergen (79°N), Geomorphology, 95, 1-2: 48-60., https://doi.org/10.1016/j.geomorph.2006.07.031, 2008
-   Curry, A. M., Cleasby, V., and Zukowskyj, P.: Paraglacial response of steep, sediment-mantled slopes to post-'Little Ice Age' glacier recession in the central Swiss Alps, J. Quat. Sci., 21, 211–225, https://doi.org/10.1002/jqs.954, 2006

**4. Lines 54 – 55:** *"Sediment yield depends on water discharge and sediment availability which are both highly variable in space and time."*

*Consider adding one or more citations here.*

We will add the following 2 citations:

-   Heckmann, T. and Schwanghart, W.: Geomorphic coupling and sediment connectivity in an alpine catchment — Exploring sediment cascades using graph theory, Geomorphology, 182, 89–103, https://doi.org/https://doi.org/10.1016/j.geomorph.2012.10.033, 2013
-   Hooke, R. L.: Toward a uniform theory of clastic sediment yield in fluvial systems, GSA Bulletin, 112, 1778–1786, https://doi.org/10.1130/0016-7606(2000)112

**5. Line 148:** *"manned photogrammetric flights"*

*I would move away from "manned" and describe those as crewed or airborne.*

We are going to replace the term Manned/Unmanned with Crewed/Uncrewed. Additionally, it will be detailed at line 150 that "The Crewed aerial flight was carried out with a P90e light aircraft. Its handling allows easy flight altitude changes to maintain a constant GSD"

**6. Lines 154 – 159:**

*How many targets did you use in total? Did you deploy the targets only in 2021? Why did you not consider collecting independent checkpoints for quality assessment?*

The artificial targets were deployed and measured in 2021 and independent Check Points have been used for quality assessment. The derived metric products (orthoimagery and DSM) generated using as GCP/CP the targets described above are used as reference data to extract GCPs and CPs for all the available datasets.

Section 2.2.1 (line 159) will be integrated with the following sentences.

"The 1 m x 1 m markers deployed in 2021 have been used as both GCPs and independent CPs (details in Table 2) for the 2021 crewed aerial survey. Considering that the focus is on relative displacements rather than on absolute values, natural GCPs and CPs have been then identified on the 2021 orthomosaic and DSM to orient the 2020 crewed aerial imagery and assess its 3D positional accuracy (considering that the artificial markers were not yet available in 2020).

The same approach has been used for drone surveys (except for one drone survey where, exploiting the RTK capabilities of the drone GNSS receiver, a direct georeferencing approach has been adopted)

Section 3.1 will be integrated with the following sentence (at line 268) and table.

"Table **XX** shows the planimetric and altimetric errors calculated on both GCPs and CPs. The "reference" dataset is characterised by a planimetric accuracy (CPs) of 8 cm and a vertical accuracy of 11 cm. The 3D accuracies of the other datasets are calculated considering the 2021 dataset as ground truth".

Table XX - planimetric and altimetric errors calculated on both GCPs and CPs.

| Flight | n° GCP | n° CP | Residuals GCP [cm] | | Residuals CP [cm] | |
|---|---|---|---|---|---|---|
| | | | RMS plan | RMS H | RMS plan | RMS H |
| aerial 2021 | 9 | 4 | 4,7 | 6,1 | 7,7 | 11,1 |
| aerial 2020 | 18 | 7 | 24.3 | 10.6 | 9.5 | 16.7 |
| drone 9th July 2021 | 8 | 6 | 2,6 | 1,9 | 5,4 | 7,8 |
| drone 20th July 2021 | 5 | 2 | 22,7 | 8,8 | 1,8 | 1,5 |

**7. Line 160:** *"Unlike drone flights which were oriented exploiting a direct georeferencing approach"*

*What do you mean with "direct georeferencing"? Did you use the camera positions alone? If yes, why? The GPS onboard of the DJI P4 is of poor quality for high-precision photogrammetric surveys, and it is a standard practice to use ground targets in SfM-MVS studies (particularly to reduce the occurrence of systematic deformations in DEMs).*

Firstly, we would like to point out that only one of the drone surveys has been oriented by means of a Direct Georeferencing approach (having used a DJI Phantom 4 with RTK capabilities and a GNSS base station positioned on a point of known coordinates and sending RTK corrections through RTCM radio transmissions), and the highlighted sentence will be amended accordingly.

A few details about the direct georeferencing approach, including relevant references, will be added at line 160, namely:

"Direct Georeferencing refers to the orientation of remotely sensed imagery without the use of GCP, exploiting Real Time Kinematic (RTK) or Post Processing Kinematic (PPK) approaches. The RTK- or PPK-based approach enables the generation of metric products with 3D positional precision and accuracy in the range of few centimeters (Teppati Losé et al., 2020; Chiabrando et al., 2019)"

We will add the following 3 references:

- Teppati Losè, L., Chiabrando, F., and Giulio Tonolo, F.: Boosting the Timeliness of UAV Large Scale Mapping. Direct Georeferencing Approaches: Operational Strategies and Best Practices, ISPRS International Journal of Geo-Information, 9, https://doi.org/10.3390/ijgi9100578, 2020
- Chiabrando, F., Giulio Tonolo, F., and Lingua, A.: UAV DIRECT GEOREFERENCING APPROACH IN AN EMERGENCY MAPPING CONTEXT. THE 2016 CENTRAL ITALY EARTHQUAKE CASE STUDY, Int. Arch. Photogramm. Remote Sens. Spatial Inf. Sci., XLII-2/W13, 247–253, https://doi.org/10.5194/isprs-archives-XLII-2-W13-247-2019, 2019
- Teppati Losè, L., Chiabrando, F., and Giulio Tonolo, F.: ARE MEASURED GROUND CONTROL POINTS STILL REQUIRED IN UAV BASED LARGE SCALE MAPPING? ASSESSING THE POSITIONAL ACCURACY OF AN RTK MULTI-ROTOR PLATFORM, Int. Arch. Photogramm. Remote Sens. Spatial Inf. Sci., XLIII-B1-2020, 507–514, https://doi.org/10.5194/isprs-archives-XLIII-B1-2020-507-2020, 2020.

The flight over the front was oriented with direct georeferencing and its positional accuracy was estimated based on the above-mentioned references (considering that no stable CP can be identified over the glacier area).

**8. Lines 163 – 166:** *"Due to a large number of well-distributed ground control points, the 2021 aerial survey was considered the reference model (referred to as 'Model" Zero') to be used for multitemporal analyses. The 2020 survey was, therefore, co-registered (i.e., georeferenced in the same reference system, enabling the overlap of all the derivative products) with the 2021 survey."*

*This suggests that you did not use any target in 2020 (see comment 6), am I right? How did you co-register the 2020 survey? Could you please explain the co-registration procedure? Could you provide statistics on the quality of the co-registration?*

As detailed in a previous reply, for 2020 survey 18 GCPs e 7 CPs have been used. The co-registration is granted by the fact that the GCPs have been identified on the 2021 final products. The relevant positional accuracy statistics will be reported in a new table (the Table X mentioned above) in the revised version of the manuscript.

**9. Lines 282 – 284:** *"The aerial DSMs were preliminary compared to the LiDAR DSM as of 2008 available on Valle d'Aosta Geoportal to verify the consistency of the produced model, checking the stability of the periglacial rocky areas. Subsequently, 2021 and 2020 DSMs were subtracted to quantify glacier ablation and displacement"*

*I would move this section into the methods, and explain how you compared the DSMs. In the results, it would be more informative to provide the statistics of such a comparison (e.g., mean error, std of error – not the RMSE) in order to demonstrate that your DSMs were free of systematic (mean error close to 0) and random (std of error close to 0) errors.*

We'll insert the following sentence in the methods section (2.2.1) at line 167. Additionally, the reference to the 2008 dataset will be removed, not being one of the main datasets described in the manuscript (following the advice of the reviewer in comment 11).

"Reiterating that the main focus is on evaluating relative displacements, the DSMs have been compared using a pixel-by-pixel approach. Specifically, the height from the 2021 DSM (i.e. the pixel value) was subtracted from the 2020 DSM one. The overall comparison enables the evaluation of the changes of the glacier surfaces, while the comparison limited to stable areas enables a further validation of the elevation products. To define the stable areas, we considered areas not covered by ice, snow or water: in order to retrieve a statistically relevant comparison dataset, we included in the stable areas also outwash plains, areas that may have been affected by geomorphological changes mainly due to water erosion and deposits. Since the stable areas are used as relative validation of the DSM, this choice is conservative since it may worsen the statistics." The outline of the stable areas will be inserted in Figure 8a.

In the results section (3), in addition to the result related to the glacier monitoring, a dedicated table related to the difference of DSMs over stable areas will be included, showing the statistics requested by the reviewer, with and without the exclusion of outliers in the DSMs. The median of differences is -0.098 m, the mean differences and standard deviation -0.082 m ± 0.788 m and the mean differences and standard deviation at 95% confidence level is -0.072 m ± 0.141 m. The Table XXX will be as follows.

Table XXX - Elevation differences (DSM 2020 - DSM 2021) on stable areas.

| Elevation differences (DSM 2020 - DSM 2021) on stable areas | |
|---|---|
| Median | -0.098 m |
| Mean | -0.082 m ± 0.788 m |
| Standard deviation (95% confidence level) | 0.072 m ± 0.141 m |

**10. Figure 8a**:

*I am a little concerned about the way you presented the DSM of difference. First, why did you not use a bivariate scale (from –X to +X)? A bivariate scale would help a lot in my opinion. Second, why did you not apply a Limit of Detection? The use of a Limit of Detection is a common practice in DoDs, and allows showing changes that are statistically significant (e.g., at 68 or 95% confidence limits). Lastly, from the DoD presented in Fig. 8a it seems that the whole study area experienced at least some movements in Z, is that really possible? Did you check for systematic deformations (e.g., doming, datum shift) in your DSMs? There is the need of providing statistics that illustrate the quality of your DSMs, e.g. the mean error in Z (i.e., systematic errors) and the std of error in Z (i.e., random errors) in respect to reference point altitudes or independent check points.*

• bivariate scale (from –X to +X): We appreciate the advice and we have modified the image accordingly, introducing a bivariate scale and the Limit of Detection (95%) thresholds in figure 8a.

[Figure]

**Figure 8. (a)** Difference between the DSM of 2020 and 2021. The black line refers to the cross-section A-A', whose 2020 (red) and 2021 (blue) elevation profiles are shown in panel **(b)**, with a zoom-in on the central tongue of the glacier in panel **(c)**.

• Limit of Detection: Thank you for your suggestion. The text will be integrated with a discussion on the suggested LoD approach at line 285: "When comparing two DSMs (i.e. Difference of DSMs, DoD) it's crucial to distinguish the information (actual vertical displacement) from the noise. To this purpose, the Limit of Detection approach has been adopted. The vertical error of the two DSMs propagates when calculating their difference. From the standard deviation of the DSM, it is possible to calculate the standard deviation of the difference exploiting the error propagation theory. The vertical precision (based on CPs) of the DSMs is 11 cm for 2021 and 16.7 cm 2020. The DoD LoD at 68% confidence level is 20 cm. Figure 8a shows the differences between the 2021 and 2020 DSMs starting from a LoD threshold of 95% = 40 cm." - (ref. Azmoon, B., Biniyaz, A., and Liu, Z.: Use of High-Resolution Multi-Temporal DEM Data for Landslide Detection, Geosciences, 12, https://doi.org/10.3390/geosciences12100378 , 2022.)

• systematic deformations:  Considering that 2020 aerial surveys were georeferenced using control points extracted from the 2021 one and the CPs horizontal residuals are comparable with the DSM resolution, we do not consider Datum Shift significant. Additionally, using the ETRF2000 reference system, the related displacements (about 1 mm/year) do not lead to a significant datum shift.

• in Fig. 8a: figure 8a will be replaced considering the Limit of Detection at 95% confidence level, as suggested.

**11. Line 289:** *"Additionally, a comparison with the 2008 DSM shows a lowering of glacier surface up to 50 meters in glacial front areas"*

*Where is this result presented?*

As previously mentioned (reply to comment 9), we'll remove the reference to the 2008 DSM (lines 282-283 and 289), not being the focus of the manuscript (we'll include a reference to such comparison in the conclusion section)

**12. Lines 290 – 293:** *"As far as very high-resolution satellite stereo pairs are concerned, they enable the extraction of 3D information with a lower vertical accuracy (metric level) with respect to aerial and drone data. Nevertheless, the coverage of a much larger area (in the range of hundreds of square kilometres) enables a multiscale and multiplatform approach to identify the most critical areas where to focus the monitoring activities in the field"*

*You do not present these results nor discuss them later, what is the point of including such a thing?*

As per the 2008 DSM, we do agree and we will remove line 167-170 and 290-293 related to satellite imagery.

**13. Lines 300 – 303:** *"The x-y-z locations of the first interface, representing the lake bottom, detected in all the GPR sections, were interpolated to produce a bathymetry map (Figure 10, which also displays the sediment thickness distribution and the electrical conductivity measurements). The perimeter of the lake, retrieved from the 6-cm-resolution orthophoto, was useful to fix the 0-depth in the interpolation process."*

*This section reads like methods. I would move it to the methods.*

We will re-arrange and move the paragraph under consideration to the methods section as follows.

- we will delete the sentence in lines 195-196: "The analysis of the GPR travel times provided the sections of the water depth and sediment thickness, which were interpolated into a bathymetric model.";
- we will add in line 195: "The GPR sections acquired were processed according to a set of standard processing steps, as detailed in Vergnano et al. (2023). The x-y-z locations of the first interface, representing the lake bottom, detected in all the GPR sections, were interpolated to produce a bathymetry map (Figure 11, which also displays the sediment thickness distribution and the electrical conductivity measurements). The perimeter of the lake, retrieved from the 6-cm-resolution orthophoto, was useful to fix the 0-depth in the interpolation process."

Nevertheless, the resulting interpolated lake bathymetry, being one of the research outputs, will be presented in the Result section.

**14. Lines 326 – 334:** *"The ecoLog1000 and CTDs instruments were first installed in July 2021 and June 2022, respectively. The measuring periods of each sensor are shown in a time: measured-quantity diagram in Table 1. At the L4 gauging station, a set of velocity-based discharge measurements (Q) taken in the summer of 2021 and 2022 were related to the corresponding water depth measured at the gauge (h), in order to plot the stage-discharge diagram (Fig. 11(a); details of the procedure followed to determine the stage-discharge relationship are given in Appendix A). Discharge measurements were also used to calibrate the lake outflow curve, i.e., the relationship between the hydraulic head (H) in the lake and the flowing discharge (see Fig. 11(c)). For this purpose, a linear fitting between the water depth at the gauge (h) and the Hydraulic head in the lake (H) was also calibrated (Fig. 11(b), R2 ~ 0.98), since the water levels in the lake and in the control cross-section in the stream are strictly related but not equal, due to the head-dependent outflow process and water speed."*

*This section reads like methods, I would therefore re-arrange this part.*

As suggested, we will re-arrange and move the paragraph under consideration to the methods section as follows:

- we will add in line 219 "The ecoLog1000 and CTDs instruments were first installed in July 2021 and June 2022, respectively; the measuring periods of each sensor are shown in a time: measured-quantity diagram in Table 1.";
- in line 231 we will change the sentence "Flow velocity measurements were taken with an Acoustic Doppler Velocimeter (ADV) current meter in the cross-section of gauging station L4 for a total of 9 surveys." To "In the summer of 2021 and 2022, a set of nine flow velocity measurements were taken with an Acoustic Doppler Velocimeter (ADV) current meter in the cross-section of gauging station L4."
- we will add before the paragraph on line 234: "The velocity-based discharge measurements $Q$ were related to the corresponding water depth $h$ measured at the gauge, in order to plot the stage-discharge diagram (Fig.4 (a)), further details on the procedure are given in Appendix A)."
- in line 236 we will change the sentence and add further details: "To monitor the water level in the lake, the relationship between the water level recorded continuously in the L4 gauging station and the water level in L4 was determined." to "Due to backwater effects at the outflow, the water levels in the lake and in the control cross-section are not identical, but strictly related. Therefore, in order to monitor the lake's stage, a relationship between the continuously recorded water level at the gauging station and the water level in the lake far from the gauging station, was determined."
- we will add in line 245 "The best fitting of relation between the water depth measurements at the gauge $h$ and the Hydraulic head in the lake $H$ was found to be linear (H=1.3h-0.1279, $R^2$~0.98; (Fig.4(b)). The stage-discharge diagram (h-Q) and the linear fitting (h-H) were used to calibrate the lake outflow curve, i.e., the relationship between the hydraulic head ($H$) in the lake and the flowing discharge ($Q$) (Fig.4(c)).".

Accordingly, we will amend the first paragraph of subsection "3.3 Hydrometric Monitoring" as follows: "The investigation at the L4 gauging station involved: i) a set of 9 velocity-based discharge measurements which allowed the stage-discharge diagram (h-Q) to be assessed; ii) a set of 15 elevation difference measurements which led to the linear fitting (h-H) and the lake outflow curve (H-Q) (Fig. 4)."

**14. Line 357:** *"permits the identification of time intervals characterized by intense transport"*

*Although bed load would easily occur at high discharges, is the signal you see necessarily related to intense transport events? Or, could the peaks be related to flow turbulence instead?*

To better clarify this, we have added in line 363 the following sentences: ''It is assumed that the geophone signal (Figure 12b) permits the identification of time intervals characterised by intense transport since, in correspondence with the peaks of the envelope, the power increases for high values of frequency (Figure 14). Indeed, the power in the lower bands is attributed to turbulent fluid flow (Schmandt et al., 2013) while that in the higher bands to bedload (Schmandt et al., 2013; Bakker et al., 2020).".

**15. Lines 358 – 359:** *"Raw seismic signals were filtered in the band 5-95 Hz and then the envelope was calculated as the average of the absolute value of the filtered signal over a time window of 1 min"*

*The sentence reads like methods, consider moving it in the appropriate section.*

We will move the sentence under consideration to the methods section below line 260, after the integration in the answer to general comment b.

**16. Lines 363 – 364:** *"In 2021, we directly observed the absence of bedload transport in three days (10 July, 20 July and 13 September)."*

*What does "directly observed" mean here? Could you be more precise?*

To clarify this concept, the sentence will be modified as follows: "In 2021, we observed, through direct inspection of the flow field, the absence of bedload transport in three days (10 July, 20 July and 13 September)."

**17. Lines 364 – 371**: *"During the 2022 season, we performed direct measurements of bedload transport at the glacier mouth by means of portable samplers on the occasion of one day of intense glacier melt (14 July) and at the end of the monitoring season (16 September). Bedload traps (4 mm mesh size, 20 × 30 cm opening, (Bunte et al., 2004)) were deployed simultaneously at 2 positions. Measured unit bedload rates feature a large variability ranging from 0.02 to 16.2 kg/m/min in a few hours, as already observed in glacierized basins (Coviello et al., 2022). Bedload samples were sieved and weighed to obtain the grain size distribution. The total bedload transport rate Qs (kg/min above 4 mm) for each sampling period (ranging 370 from 2 to 30 min) was estimated as width-weighted averages based on the available positions sampled."*

*This section really is about methods, I would therefore move into the methods.*

We will move this paragraph to the end of subsubsection "2.2.4 Bedload monitoring".

**18. Lines 416 – 417:** *"It is important to stress that the accurate georeferencing of all the acquired data with respect to the same Datum plays a crucial role in the data integration phase and in enabling the multitemporal analyses."*

*This is right, but what about systematic deformations that could well lead to erroneous multitemporal analysis? See comment 10.*

This comment has been addressed in a previous reply to comment 10.

---

## Author Response (AR1)

We wish to thank Referee 1 and Referee 2 for the careful reading and the valuable comments.

We provide below a summary of the key changes implemented and a point-by-point response to all the raised queries (in italic) starting with those of Referee 1 and then Referee 2.

The answer to follow will refer to the lines corresponding to the first version of the manuscript
* * *
**Referee 1**

**General comments**

**a. Section 2.2.1 Geomatic survey:** *The section does not provide a detailed explanation of the photogrammetric processing, and it is not supported by the relevant literature. In my opinion, this section needs to be improved. For instance, the name "SfM-MVS photogrammetry" is never mentioned throughout the article and no effort is made to explain the steps needed in a rigorous photogrammetric study. A few examples: the authors do not mention how the images were collected (e.g., drone flight geometries), how many ground control points they used, or which software or freeware they used to process the images. Finally, the author do not provide any information on the quality of the DSMs, therefore questioning if their results are reliable or not. This important weakness needs to be addressed.*

We are grateful to the Reviewer for the specific feedback. Section 2.2.1 has been integrated with all required details about the photogrammetric workflow supported by the relevant literature.

Additional details on the Drone and Crewed Aerial Photogrammetric surveys have been provided and an updated Table 2 has been integrated in the article, detailing the number of Ground Control Points (GCP) and Check Points (CP). Line 138 has been updated adding a remark about the difficulty of retrieving or positioning/measuring control point coordinates in glacier environments, as in the following link: https://zenodo.org/record/8089499 .

Table 2 - Photogrammetric flights carried out on the study area between 2020 and 2021.

| Photogrammetric flight | Date of acquisition | Covered area | Extent (km$^2$) | Average flight height (m) | GSD (m) | Number of Images | Number of GCPs | Number of CPs |
|---|---|---|---|---|---|---|---|---|
| Aerial | 30/09/2020 | Glacier and a portion of the proglacial area | 25.2 | 818 | 0.07 | 867 | 18 | 7 |
| UAV | 9/07/2021 | L1, L2 and L4 | 2.6 | 126 | 0.03 | 1480 | 6 | 6 |
| UAV | 20/07/2021 | L1 and L4 | 0.4 | 89.2 | 0.02 | 369 | 6 | 2 |
| UAV | 20/07/2021 | Glacier front and lower part | 1.1 | 159 | 0.04 | 623 | Direct georeferencing | Direct georeferencing |
| Aerial | 13/09/2021 | Glacier and proglacial area | 34.5 | 877 | 0.06 | 1100 | 9 | 4 |

**b. Section 2.2.4 Bedload monitoring:** *As per the geomatics, I believe there is the need of providing a more detailed explanation of the bedload monitoring since the use of seismometers in bedload studies is relatively recent. It would be very useful to know in more detail how the data were processed and the steps required to go from the raw signal to the results presented here.*

We are thankful to the Reviewer for the general and detailed comments. As requested, section 2.2.4 has been modify as follows:

- the sentence on line 252 "Data are recorded with a DATA-CUBE3 (solar power supply, 24-bit converter, GPS-based time synchronization) with a sampling frequency of 200 Hz and stored on site." has been changed to "Data are recorded with a DATA-CUBE3 (solar power supply, 24-bit converter, GPS-based time synchronization) configured with an amplifier gain of 16, with a sampling frequency of 200 Hz and stored on site."
- in line 261 we have added "The counts exported by the DATA-CUBE[3] are converted to vertical ground velocity considering logger and geophone sensitivities according to the manufacturer's specifications. The power spectral density is determined as the ratio of the square of the absolute value of the Fourier transform to the time window (Bakker et al., 2020)."

In this way, we believe that the procedure is clearer, and the added reference should provide all details explaining the steps in data processing.

**c. Data availability:** *The 2020 orthophoto and DSM are not available on Zenodo (https://zenodo.org/record/7713299). Are you going to include them in future?*

Thanks for pointing out that also the availability of 2020 metric products could be valuable for the readers. They have now been uploaded to Zenodo at the following link: https://zenodo.org/record/8089499

**Detailed comments:**

**1. Lines 26 – 28:** *"Alpine glacier retreat is leading to increased exposure of formerly glaciated terrain, entailing the colonization of plants and animals, and changes in morphodynamics and sediment transfer."*

*Consider adding one or more citations here.*

Thank you for your remark, as a result of the changes made to the text, also following the suggestions of the second rewier, this sentence is no longer present in the introduction.

**2. Lines 29 – 30:** *"Little Ice Age (LIA)"*

*You already defined the acronym; perhaps just use LIA instead of "Little Ice Age (LIA)".*

Thank you for pointing that out, we have updated the manuscript and used LIA instead of "Little Ice Age (LIA)".

**3. Lines 35 – 36:** *"On the one hand, plant colonization stabilizes glacial sediment and reduces sediment fluxes; on the other hand, geomorphic processes disturb and limit vegetation succession."*

*Consider adding one or more citations here.*

Thank you for your comment, we have added the following 3 citations:

- Curry, A. M., Cleasby, V., and Zukowskyj, P.: Paraglacial response of steep, sediment-mantled slopes to post-'Little Ice Age' glacier recession in the central Swiss Alps, J. Quat. Sci., 21, 211–225, https://doi.org/10.1002/jqs.954, 2006
- Eichel, J.: Vegetation Succession and Biogeomorphic Interactions in Glacier Forelands, pp. 327–349, Springer International Publishing, Cham, https://doi.org/10.1007/978-3-319-94184-4_19, 201
- Moreau, M., Mercier, D., Laffly, D., and Roussel, E.: Impacts of recent paraglacial dynamics on plant colonization: A case study on midtre Lovénbreen foreland, Spitsbergen (79°N), Geomorphology, 95, 1-2: 48-60., https://doi.org/10.1016/j.geomorph.2006.07.031, 2008

**4. Lines 54 – 55:** *"Sediment yield depends on water discharge and sediment availability which are both highly variable in space and time."*

*Consider adding one or more citations here.*

Thank you for your remark, we have added the following 2 citations:

- Carrivick, J. L. and Tweed, F. S.: Deglaciation controls on sediment yield: Towards capturing spatio-temporal variability, Earth-Sci. Rev., 221, 103 809, https://doi.org/https://doi.org/10.1016/j.earscirev.2021.103809, 2021.
- Heckmann, T. and Schwanghart, W.: Geomorphic coupling and sediment connectivity in an alpine catchment — Exploring sediment cascades using graph theory, Geomorphology, 182, 89–103, https://doi.org/https://doi.org/10.1016/j.geomorph.2012.10.033, 2013
- Hooke, R. L.: Toward a uniform theory of clastic sediment yield in fluvial systems, GSA Bulletin, 112, 1778–1786, https://doi.org/10.1130/0016-7606(2000)112

**5. Line 148:** *"manned photogrammetric flights"*

*I would move away from "manned" and describe those as crewed or airborne.*

Thanks for pointing that out. We have replaced the term Manned/Unmanned with Crewed/Uncrewed. Additionally, it has been detailed at line 150 that "The Crewed aerial flight was carried out with a P90e light aircraft. Its handling allows easy flight altitude changes to maintain a constant GSD"

**6. Lines 154 – 159:**

*How many targets did you use in total? Did you deploy the targets only in 2021? Why did you not consider collecting independent checkpoints for quality assessment?*

Thank you for your questions, the artificial targets were deployed and measured in 2021 and independent Check Points have been used for quality assessment. The derived metric products (orthoimagery and DSM) generated using as GCP/CP the targets described above are used as reference data to extract GCPs and CPs for all the available datasets.

Section 2.2.1 (line 159) has been integrated with the following sentences.

"Among the 32 markers, 12 larger markers (1 m x 1 m) were positioned during the September 2021 campaign around the top part of the glacier area, to enable a straightforward identification on aerial images. The markers placed in 2021 have been used as both GCPs and independent CPs (details in Table 2) for the 2021 crewed aerial survey. Considering that the focus is on relative displacements rather than on absolute values, 25 natural GCPs and CPs have been then identified on the 2021 orthomosaic and DSM to orient the 2020 crewed aerial imagery and assess its 3D positional accuracy (considering that the artificial markers were not yet available in 2020). A GPC/CP based approach has been used also for UAV surveys, except for one UAV survey where, exploiting the RTK capabilities of the UAV GNSS receiver, a direct georeferencing approach has been adopted (considering it was not possible to place markers in the glacier front for safety reasons). "

Section 3.1 has been integrated with the following sentence (at line 268) and table.

"Table 4 shows the planimetric and altimetric errors calculated on both GCPs and CPs. The "reference" dataset is characterised by a planimetric accuracy (CPs) of 7.7 cm and a vertical accuracy of 11.1 cm. The 3D accuracies of the other datasets are calculated considering the 2021 dataset as ground truth.".

Table 4 - Planimetric and altimetric errors calculated on both GCPs and CPs.

| Flight | Number of GCPs | Number of CPs | Residuals GCP [cm] | | Residuals CPs [cm] | |
|---|---|---|---|---|---|---|
| | | | RMS hor | RMS ver | RMS hor | RMS ver |
| Aerial 2021 | 9 | 4 | 4.7 | 6.1 | 7.7 | 11.1 |
| Aerial 2020 | 18 | 7 | 24.3 | 10.6 | 9.5 | 16.7 |
| UAV 9th July 2021 | 6 | 6 | 2.6 | 1.9 | 5.4 | 7.8 |
| UAV 20th July 2021 | 6 | 2 | 22.7 | 8.8 | 1.8 | 1.5 |

**7. Line 160:** *"Unlike drone flights which were oriented exploiting a direct georeferencing approach"*

*What do you mean with "direct georeferencing"? Did you use the camera positions alone? If yes, why? The GPS onboard of the DJI P4 is of poor quality for high-precision photogrammetric surveys, and it is a standard practice to use ground targets in SfM-MVS studies (particularly to reduce the occurrence of systematic deformations in DEMs).*

Thank you for raising these queries. Firstly, we would like to point out that only one of the drone surveys has been oriented by means of a Direct Georeferencing approach (having used a DJI Phantom 4 with RTK capabilities and a GNSS base station positioned on a point of known coordinates and sending RTK corrections through RTCM radio transmissions), and the highlighted sentence has been amended accordingly.

A few details about the direct georeferencing approach, including relevant references, have been added at line 160, namely:

"Direct Georeferencing refers to the orientation of remotely sensed imagery without using GCP, exploiting Real Time Kinematic (RTK) or Post Processing Kinematic (PPK) approaches. The RTK- or PPK-based approach enables the generation of metric products with 3D positional precision and accuracy in the range of few centimeters (Chiabrando et al., 2019; Teppati Losè et al., 2020a, b)."

We have added the following 3 references:

- Teppati Losè, L., Chiabrando, F., and Giulio Tonolo, F.: Boosting the Timeliness of UAV Large Scale Mapping. Direct Georeferencing Approaches: Operational Strategies and Best Practices, ISPRS International Journal of Geo-Information, 9, https://doi.org/10.3390/ijgi9100578, 2020
- Chiabrando, F., Giulio Tonolo, F., and Lingua, A.: UAV DIRECT GEOREFERENCING APPROACH IN AN EMERGENCY MAPPING CONTEXT. THE 2016 CENTRAL ITALY EARTHQUAKE CASE STUDY, Int. Arch. Photogramm. Remote Sens. Spatial Inf. Sci., XLII-2/W13, 247–253, https://doi.org/10.5194/isprs-archives-XLII-2-W13-247-2019, 2019
- Teppati Losè, L., Chiabrando, F., and Giulio Tonolo, F.: ARE MEASURED GROUND CONTROL POINTS STILL REQUIRED IN UAV BASED LARGE SCALE MAPPING? ASSESSING THE POSITIONAL ACCURACY OF AN RTK MULTI-ROTOR PLATFORM, Int. Arch. Photogramm. Remote Sens. Spatial Inf. Sci., XLIII-B1-2020, 507–514, https://doi.org/10.5194/isprs-archives-XLIII-B1-2020-507-2020, 2020.

The flight over the front was oriented with direct georeferencing and its positional accuracy was estimated based on the above-mentioned references (considering that no stable CP can be identified over the glacier area).

**8. Lines 163 – 166:** *"Due to a large number of well-distributed ground control points, the 2021 aerial survey was considered the reference model (referred to as 'Model" Zero') to be used for multitemporal analyses. The*

*2020 survey was, therefore, co-registered (i.e., georeferenced in the same reference system, enabling the overlap of all the derivative products) with the 2021 survey."*

*This suggests that you did not use any target in 2020 (see comment 6), am I right? How did you co-register the 2020 survey? Could you please explain the co-registration procedure? Could you provide statistics on the quality of the co-registration?*

Thank you for raising these queries. As detailed in a previous reply, for 2020 survey 18 GCPs e 7 CPs have been used. The co-registration is granted by the fact that the GCPs have been identified on the 2021 final products. The relevant positional accuracy statistics has been reported in a new table (the Table 2 mentioned above) in the revised version of the manuscript.

**9. Lines 282 – 284:** *"The aerial DSMs were preliminary compared to the LiDAR DSM as of 2008 available on Valle d'Aosta Geoportal to verify the consistency of the produced model, checking the stability of the periglacial rocky areas. Subsequently, 2021 and 2020 DSMs were subtracted to quantify glacier ablation and displacement"*

*I would move this section into the methods, and explain how you compared the DSMs. In the results, it would be more informative to provide the statistics of such a comparison (e.g., mean error, std of error – not the RMSE) in order to demonstrate that your DSMs were free of systematic (mean error close to 0) and random (std of error close to 0) errors.*

Thank you for your valuable comment. We have inserted the following sentence in the methods section (2.2.1) at line 167. Additionally, the reference to the 2008 dataset has been removed, not being one of the main datasets described in the manuscript (following the advice of the Reviewer in comment 11).

"Since the main objective is the evaluation of relative displacements, the DSMs were compared using a pixel-by-pixel approach. In particular, the height of DSM 2021 (i.e. the pixel value) was subtracted from that of DSM 2020. The overall comparison allows the evaluation of changes in the glacier surface, while the comparison limited to stable areas allows further validation of the elevation products. Stable areas were defined as those not covered by ice, snow or water. Outwash plains, which may have been affected by geomorphological changes (e.g. due to erosion and water deposits) between the time of the surveys, were also considered stable areas. These areas were included to obtain a statistically relevant comparison dataset. The stable areas were used for relative validation, therefore considering areas that may have changed between the time of the surveys is conservative, as it could worsen the statistics."

The outline of the stable areas has been inserted in Figure 8a.

In the results section (3), in addition to the result related to the glacier monitoring, a dedicated table related to the difference of DSMs over stable areas has been included, showing the statistics requested by the Reviewer, with and without the exclusion of outliers in the DSMs. The median of differences is -0.098 m, the mean differences and standard deviation -0.082 m ± 0.788 m and the mean differences and standard deviation at 95% confidence level is -0.072 m ± 0.141 m. The Table 5 has been added:

Table 5 - Elevation differences (DSM 2020 - DSM 2021) on stable areas.

| Elevation difference (DSM 2020- DSM 2021) on stable areas | |
|---|---|
| Median | -0.098 m |
| Mean | -0.082 m $\pm$ 0.788 m |
| Standard deviation (95% confidence level) | 0.072 m $\pm$ 0.141 m |

**10. Figure 8a**:

*I am a little concerned about the way you presented the DSM of difference. First, why did you not use a bivariate scale (from –X to +X)? A bivariate scale would help a lot in my opinion. Second, why did you not apply a Limit of Detection? The use of a Limit of Detection is a common practice in DoDs, and allows showing changes that are statistically significant (e.g., at 68 or 95% confidence limits). Lastly, from the DoD presented in Fig. 8a it seems that the whole study area experienced at least some movements in Z, is that really possible? Did you check for systematic deformations (e.g., doming, datum shift) in your DSMs? There is the need of providing statistics that illustrate the quality of your DSMs, e.g. the mean error in Z (i.e., systematic errors) and the std of error in Z (i.e., random errors) in respect to reference point altitudes or independent check points.*

• bivariate scale (from –X to +X): We appreciate the advice and we have modified the image accordingly, introducing a bivariate scale and the Limit of Detection (95%) thresholds in figure 8a.

[Figure]

**Figure 8. (a)** Difference between the DSM of 2020 and 2021. The black line refers to the cross-section A-A', whose 2020 (red) and 2021 (blue) elevation profiles are shown in panel **(b)**, with a zoom-in on the central tongue of the glacier in panel **(c)**.

• Limit of Detection: Thank you for your comment. The text has been integrated with a discussion on the suggested LoD approach at line 285: "When comparing two DSMs (i.e. Difference of DSMs, DoD) it's crucial to distinguish the information (actual vertical displacement) from the noise. To this purpose, the Limit of Detection approach has been adopted. The vertical error of the two DSMs propagates when calculating their difference. From the RMS of the DSM, it is possible to calculate the RMS of the difference exploiting the error propagation theory. The vertical precision (based on CPs) of the DSMs is 11.1 cm for 2021 and 16.7 cm 2020. The DoD Limit of Detection (LoD) at 68\% confidence level is 20 cm ($\sqrt{11.1^2 + 16.7^2}$).

• systematic deformations: Considering that 2020 aerial surveys were georeferenced using control points extracted from the 2021 one and the CPs horizontal residuals are comparable with the DSM resolution, we do not consider Datum Shift significant. Additionally, using the ETRF2000 reference system, the related displacements (about 1 mm/year) do not lead to a significant datum shift.

• in Fig. 8a: figure 8a has been replaced considering the Limit of Detection at 95% confidence level, as suggested.

**11. Line 289:** *"Additionally, a comparison with the 2008 DSM shows a lowering of glacier surface up to 50 meters in glacial front areas"*

*Where is this result presented?*

As previously mentioned (reply to comment 9), we'll remove the reference to the 2008 DSM (lines 282-283 and 289), not being the focus of the manuscript.

**12. Lines 290 – 293:** *"As far as very high-resolution satellite stereo pairs are concerned, they enable the extraction of 3D information with a lower vertical accuracy (metric level) with respect to aerial and drone data. Nevertheless, the coverage of a much larger area (in the range of hundreds of square kilometres) enables a multiscale and multiplatform approach to identify the most critical areas where to focus the monitoring activities in the field"*

*You do not present these results nor discuss them later, what is the point of including such a thing?*

Thank you for highlighting this issue. As per the 2008 DSM, we do agree and we have removed line 167-170 and 290-293 related to satellite imagery.

**13. Lines 300 – 303:** *"The x-y-z locations of the first interface, representing the lake bottom, detected in all the GPR sections, were interpolated to produce a bathymetry map (Figure 10, which also displays the sediment thickness distribution and the electrical conductivity measurements). The perimeter of the lake, retrieved from the 6-cm-resolution orthophoto, was useful to fix the 0-depth in the interpolation process."*

*This section reads like methods. I would move it to the methods.*

Thank you for your valuable comment. We have re-arranged and moved the paragraph under consideration to the methods section as follows.

- we have deleted the sentence in lines 195-196: "The analysis of the GPR travel times provided the sections of the water depth and sediment thickness, which were interpolated into a bathymetric model.";
- we added appendix B to include the steps performed in the software and we have added in line 195 the following paragraphs: "The GPR sections acquired were processed according to a set of standard processing steps, performed in Reflexw software (Sandmeier, 2021; Vergnano et al., 2023), and reported in Appendix B.

    The x-y-z locations of the first interface, representing the lake bottom, detected in all the GPR sections, were interpolated with a linear triangulation-based method (griddata function of MATLAB) to produce a bathymetry map (Figure 11, which also displays the sediment thickness distribution and the electrical conductivity measurements). The perimeter of the lake, retrieved from the 6-cm-resolution orthophoto acquired on the day of the geophysical survey, was useful to fix the 0-depth in the interpolation process."

Nevertheless, the resulting interpolated lake bathymetry, being one of the research outputs, has been presented in the Result section.

**14. Lines 326 – 334:** *"The ecoLog1000 and CTDs instruments were first installed in July 2021 and June 2022, respectively. The measuring periods of each sensor are shown in a time: measured-quantity diagram in Table 1. At the L4 gauging station, a set of velocity-based discharge measurements (Q) taken in the summer of 2021 and 2022 were related to the corresponding water depth measured at the gauge (h), in order to plot the stage-discharge diagram (Fig. 11(a); details of the procedure followed to determine the stage-discharge relationship are given in Appendix A). Discharge measurements were also used to calibrate the lake outflow curve, i.e., the relationship between the hydraulic head (H) in the lake and the flowing discharge (see Fig.*

*11(c)). For this purpose, a linear fitting between the water depth at the gauge (h) and the Hydraulic head in the lake (H) was also calibrated (Fig. 11(b), R2 ~ 0.98), since the water levels in the lake and in the control cross-section in the stream are strictly related but not equal, due to the head-dependent outflow process and water speed."*

*This section reads like methods, I would therefore re-arrange this part.*

Thank you for your remark, as suggested, we have re-arranged and moved the paragraph under consideration to the methods section as follows:

- we have added in line 219 "The ecoLog1000 and CTDs instruments were first installed in July 2021 and June 2022, respectively; the measuring periods of each sensor are shown in Table 1.";
- in line 231 we have changed the sentence "Flow velocity measurements were taken with an Acoustic Doppler Velocimeter (ADV) current meter in the cross-section of gauging station L4 for a total of 9 surveys." To "In the summer of 2021 and 2022, a set of nine flow velocity measurements were taken with an Acoustic Doppler Velocimeter (ADV) current meter in the cross-section of gauging station L4."
- we have added before the paragraph on line 234: "The velocity-based discharge measurements $Q$ were related to the corresponding water depth $h$ measured at the gauge (Figure 4 (a)), to plot the stage-discharge diagram (Figure 5(a)), further details on the procedure are given in Appendix A)."
- in line 236 we have changed the sentence and add further details: "To monitor the water level in the lake, the relationship between the water level recorded continuously in the L4 gauging station and the water level in L4 was determined." to "Due to backwater effects at the outflow, the water levels in the lake and the control cross-section are not identical, but strictly related. Therefore, to monitor the lake's stage, a relationship between the continuously recorded water level at the gauging station and the water level in the lake far from the gauging station was determined."
- we have added in line 245 "The best fitting of the relation between the water depth measurements at the gauge $h$ and the Hydraulic head in the lake $H$ was found to be linear (H=1.3h-0.1279, $R^2$~0.98; (Fig.4(b)). The stage-discharge diagram (h-Q) and the linear fitting (h-H) were used to calibrate the lake outflow curve, i.e., the relationship between the hydraulic head ($H$) in the lake and the flowing discharge ($Q$) (Fig.5(c)).".

Accordingly, we have amended the first paragraph of subsection "3.3 Hydrometric Monitoring" as follows: "The investigation at the L4 gauging station involved: i) a set of 9 velocity-based discharge measurements which allowed the stage-discharge diagram (h-Q) to be assessed; ii) a set of 15 elevation difference measurements which led to the linear fitting (h-H) and the lake outflow curve (H-Q) (Fig. 5)."

**14. Line 357:** *"permits the identification of time intervals characterized by intense transport"*

*Although bed load would easily occur at high discharges, is the signal you see necessarily related to intense transport events? Or, could the peaks be related to flow turbulence instead?*

Thank you for highlighting this aspect. To better clarify this, we have added in line 363 the following sentences: "It is assumed that the geophone signal (Figure 12b) permits the identification of time intervals characterised by intense transport since, in correspondence with the peaks of the envelope, the power increases for high values of frequency (Figure 14). Indeed, the power in the lower bands is attributed to turbulent fluid flow (Schmandt et al., 2013) while that in the higher bands to bedload (Schmandt et al., 2013; Bakker et al., 2020).".

**15. Lines 358 – 359:** *"Raw seismic signals were filtered in the band 5-95 Hz and then the envelope was calculated as the average of the absolute value of the filtered signal over a time window of 1 min"*

*The sentence reads like methods, consider moving it in the appropriate section.*

Thank you for the comment. We have moved the sentence under consideration to the methods section below line 260 (after the integration reported in general comment b).

**16. Lines 363 – 364:** *"In 2021, we directly observed the absence of bedload transport in three days (10 July, 20 July and 13 September)."*

*What does "directly observed" mean here? Could you be more precise?*

Thank you for the remark. To clarify this concept, the sentence has been modified as follows: "In 2021, we observed, through direct inspection of the flow field, the absence of bedload transport in three days (10 July, 20 July and 13 September)."

**17. Lines 364 – 371**: *"During the 2022 season, we performed direct measurements of bedload transport at the glacier mouth by means of portable samplers on the occasion of one day of intense glacier melt (14 July) and at the end of the monitoring season (16 September). Bedload traps (4 mm mesh size, 20 × 30 cm opening, (Bunte et al., 2004)) were deployed simultaneously at 2 positions. Measured unit bedload rates feature a large variability ranging from 0.02 to 16.2 kg/m/min in a few hours, as already observed in glacierized basins (Coviello et al., 2022). Bedload samples were sieved and weighed to obtain the grain size distribution. The total bedload transport rate Qs (kg/min above 4 mm) for each sampling period (ranging 370 from 2 to 30 min) was estimated as width-weighted averages based on the available positions sampled."*

*This section really is about methods, I would therefore move into the methods.*

Thank you for the comment. We have moved this paragraph to the end of subsubsection "2.2.4 Bedload monitoring".

**18. Lines 416 – 417:** *"It is important to stress that the accurate georeferencing of all the acquired data with respect to the same Datum plays a crucial role in the data integration phase and in enabling the multitemporal analyses."*

*This is right, but what about systematic deformations that could well lead to erroneous multitemporal analysis? See comment 10.*

Thank you for this valuable question. This comment has been addressed in a previous reply to comment 10.

**General comments**

*Multidisciplinary framework:*

*Most of the subsections need a clear description of the data collected, the methods and software used, the uncertainties, etc. This is particularly the case for the geomatic survey, where much basic information is missing, such as: processing steps, number of GCPs, software used, types of errors, etc.*

We thank you for highlighting this aspect. This very relevant comment was raised by the Reviewer 1 and the new version of the manuscript was edited to include all the basic information previously missing.

*Discussion and conclusion:*

*The discussion seems quite confused with different terms and should be organised into two or more central ideas. I would suggest adding a discussion of uncertainty in your data. It also needs to be better placed in the context of Rutor Glacier and its proglacial system. What do we know that we did not know before?*

Thank you for raising this question, we would like to remark that this work aims to present the methodology with which to study proglacial areas. Focusing on the multidisciplinary approach and not on processes analysis, it provides exploratory products useful for deepening and understanding the processes occurring in the Rutor proglacial areas.

**Detailed comments:**

**1. Line 2: "***This is a bit vague, can you please specify what you mean by "changes in the water"?***"** with reference to "changes in the water"

Thank you for raising this question. In order to be more specific, we have changed the text in line 2 as follows: "The recession of Alpine glaciers causes an increase in the extent of proglacial areas and leads to changes in the water discharge and sediment balance (morphodynamics and sediment transport)."

**2. Lines 7-8:** *"uncrewed and crewed aerial surveys"* with reference to "both uncrewed (drone) and crewed aerial photogrammetric flights"

Thank you for your comment. We have changed the sentence under consideration (line 7) accordingly to your suggestion.

**3. Line 14:** *"Snow cover and glaciers"* with reference to "snow, ice"

Thank you for your comment. We have changed the sentence in line 15 considering also the 4[th] comment as follows: "Global warming is entailing a rapid decline of the cryosphere globally. Mountain snow cover and glaciers, respond directly and rapidly to climate change making them key indicators of global warming."

**4. Line 15: "***In fact, the response of mountain permafrost to climate change is less obvious and direct than that of snow cover and glaciers.***"** with reference to "responds directly and rapidly to climate change"

Thank you for highlighting this aspect. Permafrost does respond to climate change but, as you pointed out, its response is less obvious and direct than that of snow cover and glaciers. Therefore, we decided to not mention it.

**5. Lines 15-16:** "*That is too general, better say snow cover and glaciers.*" with reference to "the cryosphere"

Thank you for your comment. We have changed the text in lines 15-16 as suggested: "The decline of snow cover and glaciers exposes more land and water surfaces to solar energy, leading to decreasing albedo and to weathering, resulting in increased erosion."

**6. Lines 16-17:** "*Glaciers are significant agents of erosion. In your specific context, which is more significant in terms of erosion, glaciers or weathering processes?*" with reference to "resulting in increased erosion."

Thank you for raising this valuable question. The objective of our paper is to showcase a multidisciplinary approach and present the data obtained through surveys and field monitoring activities. Investigating the specific contribution of weathering to erosion would necessitate a more in-depth analysis, which we acknowledge as a potential avenue for future research.

**7. Line 18:** "*mountains*" with reference to "mountain"

Thank you. We have addressed the suggested correction.

**8. Line 22:** "*The end of the Little Ice Age (LIA) in the Alps occurred approximately in 1850.*

*Ivy-Ochs S, Kerschner H, Maisch M, Christl M, Kubik PW, Schlüchter C. Latest Pleistocene and Holocene glacier variations in the European Alps. Quat Sci Rev. 2009;28(21–22):2137-2149. doi:10. 1016/j.quascirev.2009.03.009*" with reference to "With LIA being a cooler period in the Holocene, lasting from years 1300s to 1950s"

Thank you for the comment. We have modified the date as suggested and the reference accordingly.

**9. Lines 23-24:** "*This statement is missing a reference*" with reference to "At present, most Alpine glaciers are not in equilibrium with the current climate, so they are undergoing a dramatic mass loss."

Thank you for pointing that out. We have added the following reference:

"Sommer, C., Malz, P., Seehaus, T.C. et al. Rapid glacier retreat and downwasting throughout the European Alps in the early 21st century. Nat Commun 11, 3209 (2020). https://doi.org/10.1038/s41467-020-16818-0"

**10. Lines 52-53:** "*What type of glacier response are you referring to? Please provide specific terminology for clarity.*" with reference to "Glacier response to regional and local climate is heterogeneous in space and time"

Thank you for highlighting this point. We have been more specific and we have changed lines 52 and 53 as follow: "Glacier retreat in response to the local climate is heterogeneous in space and time  and so is the water regime."

**11. Lines 65-66:** "*Please list the morphological and glaciological characteristics that make this glacier one of the most representative glaciers.*" with reference to "The Rutor Glacier is considered one of the most representative glaciers due to its geographical position and its morphological and glaciological characteristics"

Thank you for your comment. The Rutor glacier has a rich and well-documented history, and changes in the proglacial area due to rising temperatures had already been taking place for almost 200 years. Due to the orography of the proglacial area, the glacier tongue has split into 3 tongues that respond to rising temperatures differently. As already documented by Villa et al. (2007) and confirmed by our monitoring activities, the right tongue is retreating faster than the other two. Compared to Alpine glaciers, the Rutor

glacier is among those with the largest surface area and is the third largest in the Aosta Valley (GLIMS dataset).

We have modified the description of the glacier to encompass the most essential information according to the study presented here. Below are the changes we implemented in lines 64-75:

"The Rutor glacier lies at the head of the Dora Baltea Valley in La Thuile, near the French-Italian border in northwestern Italy. It is mainly oriented to the northwest and at an altitude ranging from 2540 m a.s.l. to 3486 m a.s.l., with an average altitude close to the average value for Alpine glaciers, as retrieved from the Global Land Ice Measurements from Space (GLIMS) database (GLIMS Consortium, 2005; Raup et al., 2007). The Rutor glacier is among the glaciers with the largest surface area in the Alps, and it is the third largest glacier in the Aosta Valley (GLIMS Consortium, 2005; Raup et al., 2007). At present, it has a surface area of 7.5 km2 and its front is formed by three tongues (Figure 1) that were once united (Figure 2). Villa et al. (2007) determined the Rutor glacier retreat and volume changes from the mid-19th century to 2004. Since 2005, the regional environmental protection agency (ARPA) of Valle d'Aosta has been monitoring the mass balance of the Rutor glacier, which with the exception of the years 2013, 2014 and 2016 has always been negative, resulting in a cumulative mass balance from 2005 to 2017 of -12252 mm w.e. (ARPA Valle d'Aosta, 2014). Since its maximum extent in LIA (Orombelli, 2005; Villa et al., 2007), the glacier has lost approximately 34% of its surface area. The retreat and lowering of the glacier surface are not uniform but more pronounced in the eastern tongue (Villa et al., 2007). "

**12. Line 68:** *"Please indicate this toponomy on the main figure 1."* with reference to ""Vedettes du Rutor""

Thank you for pointing that out. Considering the intention to change the text as stated in the reply to the previous comment, it may no longer be necessary to indicate the position of the "Vedettes du Rutor" in Figure 1 (a). Below, however, is the updated image with the position of the "Vedettes du Rutor" additionally indicated.

**13. Figure 1. a):** *"This inset is quite small and unreadable. Please enlarge it to be useful."*

Thank you for your comment. We have modified Figure 1 as follows:

[Figure]

**14. Figure 1. a):** *"the Rutor glacier outlines has to be reviewed. part of the outlines is on France. this is another glacier according to the GLIMS dataset."*

We appreciate you bringing this to our attention. We have updated the image considering the information from the dataset you indicated. The updated image can be seen in the reply to the comment above.

**15. Caption of Figure 1. a):** *"This is a hillshade obtained from the 2008 DSM."* with reference to "Digital Surface Model (DSM)"

Thanks for highlighting this aspect. We have modified the caption of figure 1.a) as follows:" Hillshade based on the Digital Surface Model (DSM) as of 2008 of the Rutor glacier and the L4 lake catchment (SCT Geoportale, Regione autonoma Valle d'Aosta). The upslope area of L4 outflow (hatched area with continuous black lines) has been mapped using the 2008 model of Valle d'Aosta (SCT Geoportale, Regione autonoma Valle d'Aosta). The inset shows the location in Italy."

**16. Caption of Figure 1. a):** *"Missing reference to this dataset."* with reference to "2008 model of Valle d'Aosta."

Thank you for pointing this out. We integrated the caption with the reference, this change can be seen from the answer to the previous comment.

**17. Line 70:** *"Here you are discussing mass loss, but the following sentence only provides information about changes in area. Please, provide better statements to support the dramatic mass loss."* with reference to "Due to global warming, the Rutor Glacier has gone through a dramatic mass loss"

Thank you for your comment. This remark was addressed in comment 11.

**18. Line 85:** *"'ice-marginal' or 'ice-contact' lakes"* with reference to "are attached to the glacier lobe".

Thank you for bringing this to our notice. As with the description of the glacier, we have decided to modify the description of the proglacial area in order to summaries it by removing details that are superfluous for the purposes of the presented study. Therefore, the lakes referred to in the comment are not mentioned in the updated text and consequently it is not be necessary to define them with the proper terminology as suggested. The text in lines 84-87 has been modified as follows: "Of the many lakes formed in the Rutor proglacial area, we have focused on those with the largest surface area. Of these, the one situated furthest upstream is lake L1 and is fed by the right tongue (Figure 1).

**19. Lines 99-98:** *"This is a valuable graphic resource that could benefit from further refinement. Would you consider shortening the URL link for ease of access?"* with reference to "https://poli.maps.arcgis.com/apps/instant/3dviewer/index.html?appid=0c63aa5cc0e8436ca1e7cd3dc214 bc27(last access: 17 January 2023)"

Thank you for your suggestion. The URL link has been shortened as follows: https://arcg.is/Tyeju0

**20. Lines 113-126:** *"This entire paragraph requires restructuring in terms of its content. The current references are intercalated, but their relationship is not clear."* with reference to "Guillon et al. (2018) combined sedimentary measurements with precipitation data to understand present-day suspended sediment storage and erosion processes during one melt season in the Bosson glacier proglacial area. They measured water depth and turbidity, deriving water discharge and suspended sediment concentration respectively, in three different stations within the proglacial area. Orwin and Smart (2004) characterized a proglacial channel over a 9-week ablation period by continuously measuring the water depth and turbidity in nine different gauging stations distributed within the proglacial area. Their study confirmed that sediment yield varies spatially and temporally within a proglacial area. Delaney et al. (2018) assessed erosion rates and processes in Griesgletscher's proglacial area. That glacier is located near a hydropower infrastructure so the catchment has been monitored annually since 1986. To determine volume changes and assess sediment processes in

Griesgletscher's proglacial area they used digital surface models (DSMs), reservoir bathymetry and a glacial-hydrological model (GERM). Water discharge measurements were determined by the reservoir's water level. Guillon et al. (2018) and Orwin and Smart (2004) measured both discharge and turbidity at different locations in the proglacial area, providing an explanation for the variation in space and time of proglacial suspended sediment flux but they did not assess the landscape evolution of the geomorphological features in proglacial areas. However, although Delaney et al. (2018) identified the sediment processes in the proglacial area using DSMs, they measured water discharge only at the basin outlet."

Thank you for bringing this to our attention. In this section, our goal is to showcase case studies of monitoring activities in proglacial areas. The paragraph has undergone revisions to include crucial details, resulting in the removal of some information and the addition of new elements. The revised paragraph is provided below:

"The following paragraph provides a concise overview of the monitoring methods used in three distinct studies concerning different proglacial areas.

Guillon et al. (2018) combined sedimentary measurements with precipitation data to understand present-day suspended sediment storage and erosion processes during a melt season. They measured water depth and turbidity, deriving water discharge and suspended sediment concentration respectively, in three different stations. Orwin and Smart (2004) characterized a proglacial channel over a 9-week ablation period by continuously measuring the water depth and turbidity in different gauging stations distributed within the proglacial area. Confirming that sediment yield varies spatially and temporally within a proglacial area. Delaney et al. (2018) assessed erosion rates and processes in an alpine proglacial area through digital surface models (DSMs), reservoir bathymetry and a glacial-hydrological model (GERM). Water discharge measurements were determined by the water level at the reservoir located at the basin outlet. The first two reported studies (Guillon et al., 2018; Orwin and Smart, 2004) provided an explanation for the variation in space and time of proglacial suspended sediment flux but they did not assess the landscape evolution of the geomorphological features in the whole proglacial area, whereas in the latter reported study (Delaney et al., 2018) the sediment processes in the whole proglacial area was identified but the water discharge was directly measured only at the basin outlet. The studies presented are of important value for the understanding of paraglacial dynamics, but there is a lack of studies in the literature involving repeated surveys (e.g. photogrammetric flights) and continuous monitoring (e.g. flow measurements) at several points in the proglacial and glacial area."

**21. Line 114:** *"whareabout this glacier is located?"* with reference to "Bosson glacier proglacial area"

Thank you for your comment. Considering the revisions made in response to Comment 20, we are of the opinion that specifying the location of the mentioned area is no longer essential in the paragraph.

**22. Line 117:** *"Do we have knowledge of this proglacial region? Could you please provide more specificity?"* with reference to "within the proglacial area"

I appreciate your feedback. Given the modifications made as per Comment 20, we find it unnecessary to specify the location of the mentioned area in the paragraph.

**23. Line 119:** *"Switzerland?"* with reference to "Griesgletscher's proglacial area"

Thank you for your input. Taking into account the changes made in response to Comment 20, we have determined that detailing the location of the mentioned area in the paragraph is no longer required.

**24. Line 133:** *"I would reframe this into aerial surveys (both crewed and uncrewed)."* with reference to the subsection title "Geomatic survey"

Thanks for highlighting this: nevertheless, we prefer to stick with this wording as the work described includes activities (i.e., markers materialization and measuring with GNSS positioning) that, even if realized in support to aerial surveys, require different techniques in the geomatics domain (detailed at line 155 and 156). Additionally, this wording would grant consistency with the other sections' titles, stressing the importance of the multidisciplinary approach. Nevertheless, we have modified the section title as follows: Geomatic survey: aerial acquisitions and GNSS positioning.

**25. Line 134:** *"What are the various geomatics techniques? Currently, only the photogrammetric technique is being employed."* with reference to "different geomatics techniques"

Thank you for your question. To complement comment 24, GNSS positioning was used also to measure bathymetric profiles.

**26. Line 137:** *"uncrewed aerial vehicle (UAV)"* with reference to "drone"

Thank you for bringing this to our notice. The text has been changed referring to uncrewed and crewed aerial photogrammetric flights (the drone word has been removed)

**27. Lines 138-142:** *"This should be merged into one paragraph."* with reference to

"photogrammetric flights as well as topographic measurements in the field. As far as the 2020-2021 period is concerned, the following surveys have been carried out:

– Aerial flights: 30th September 2020 and 13rd September 2021, over the whole area

– Drone flights: 9th July 2021 and 20th July 2021, over the proglacial area

A summary of the flight coverage and technical features is included in Table 2."

Your observation is noted; thank you. The text has been simplified as following: "As far as the 2020-2021 period is concerned, the surveys described in Table 2 have been carried out."

**28. Line 154:** *"Indicate how many points were installed"* with reference to "a set of"

Thanks for highlighting this. The number of markers has now been detailed in the text, line 154 has been changed as follows: "During the 2021 activities in the field, a total of 32 artificial photogrammetric markers, either squared (0.5 m x 0.5 m) plastered markers or crosses painted on stable rocks, were positioned (or painted) and measured with a Real-Time Kinematic (RTK) and static Global Navigation Satellite System (GNSS) positioning approach, using 3 Spectra Precision SP80 GNSS receivers (static data have been processed with RTKLIB software)."

**29. Line 154:** *"These are better known as Ground Control Points (GCPs)."* with reference to "artificial photogrammetric markers"

Thank you for your comment. We would opt to keep the original wording to avoid confusion between markers (either artificial or natural) and their specific use as GCPs or CPs, as introduced to address also one of the comments from Reviewer 1 (see sections 2.2.1 and 3.1)

**30. Line 155-156:** *"please indicate the manufacturer of the instruments and software used."* with reference to "a Real-Time Kinematic (RTK) and static Global Navigation Satellite System (GNSS) positioning approach"

Thanks for highlighting this. The GNSS receivers are 3 Spectra Precision SP80 and static data have been processed with RTKLIB.

**31. Line 156-159:** *"indicate the location of these markers (GCPs) on Figure 1 or another relevant figure."* with reference to "The markers were distributed on the periglacial area (to ensure stability over time), around L4 and along the L1 until the glacier front on the eastern tongue. Moreover, around the top part of the glacier area, a set of 1 m x 1 m makers were positioned during the September 2021 campaign to enable a straightforward identification on aerial images."

Your observation is noted; thank you. The location of 2021 GCPs and CPs has been added in figure 7. Furthermore, this dataset is now uploaded in Zenodo at following link: https://zenodo.org/records/10074530

**32. Line 160-161:** *"This statement is somewhat misleading. The DJI Phantom 4 drone cannot perform direct georeferencing due to the lack of high accuracy in its GPS and inertial measurements units. Therefore, to process the imagery captured with a DJI Phantom 4 drone, it is usually necessary to complement it with Ground Control Points (GCPs)."* with reference to "Unlike drone flights which were oriented exploiting a direct georeferencing approach, the camera positions of aerial flights were not geo-tagged with proper accuracy."

Thank you for your comment. We would like to point out that only one of the drone surveys has been oriented by means of a Direct Georeferencing approach (having used a DJI Phantom 4 with RTK capabilities and a GNSS base station positioned on a point of known coordinates and sending RTK corrections through RTCM radio transmissions). The highlighted sentence has been amended accordingly, including details related to the Direct Georeferencing approach.

The manuscript has been integrated with the following paragraph: "A GPC/CP based approach has been used also for UAV surveys, except for one UAV survey where, exploiting the RTK capabilities of the UAV GNSS receiver, a direct georeferencing approach has been adopted (considering it was not possible to place markers in the glacier front for safety reasons). Direct Georeferencing refers to the orientation of remotely sensed imagery without using GCP, exploiting Real Time Kinematic (RTK) or Post Processing Kinematic (PPK) approaches. The RTK- or PPK-based approach enables the generation of metric products with 3D positional precision and accuracy in the range of few centimeters (Teppati Losè et al., 2020b; Chiabrando et al., 2019; Teppati Losè et al., 2020a)."

**33. Lines 163-164:** *"Where are the large number of ground control points (GCPs) located? Please provide a figure depicting their locations and include the resulting file with x, y, and z values in the assets section of the manuscript."* with reference to " Due to a large number of well-distributed ground control points,".

Thanks for your question. As explained in the answer to comment 31, the location of 2021 GCPs and CPs has been added in figure 7. Furthermore, this dataset is now uploaded in Zenodo at following link: https://zenodo.org/records/10074530.

**34. Lines 167-170:** *"I'm sorry, but this came a bit out of the blue. Please provide a context for the use of the Pleiades images in this manuscript."* with reference to "To assess the advantages and disadvantages of a multiplatform, multiscale and multitemporal analysis, a Pleiades very-high-resolution satellite stereo-pair acquired in 2017 was also used. The satellite multispectral imagery (including visible and near-infrared data with a nominal GSD of 0.71 m resampled to 0.50 m) was processed to extract two orthoimages and one DSM".

Thank you for your comment. As addressed in the 12[th] reply to Reviewer 1, the focus of the manuscript is on aerial dataset, therefore we are not going to discuss the analyses based on satellite imagery. Accordingly, line 167-170 and 290-293 related to satellite imagery were removed

**35. Line 171:** *"The manufacturer of the instrument and the software code used to process the raw data should be specified."* with reference to section 2.2.2 Geophysical survey.

Thank you for bringing this to our notice. The manufacturer of the instrument and the software used have been specified. The updated sentence is: "The GPR antenna, manufactured by IDS, had a central frequency of 200 MHz, which provides the best possible resolution while avoiding the energy dispersion that occurs in water at frequencies higher than 200 MHz (Bradford et al., 2007). The GPR system was installed on an inflatable rowing boat and the boat was moved to cover the whole area of the lake. The GPR sections acquired were processed according to a set of standard processing steps, performed in ReflexW software (Sandmeier, 2021; Vergnano et al., 2023), and reported in Appendix B."

**36. Figure 3:** *"Provide source and date of the orhotphoto."* with reference to Figure 3.

Thanks for your comment. We have changed the caption of figure 3 as follows: "Aerial orthophoto of the Rutor proglacial area acquired on 13/09/2021 and the snout of the Rutor eastern tongue. The red polygon in the upper left orthophoto shows the position of the area enlarged in the right figure. The lakes (L1, L2, L3 and L4), the gauging stations and the geophones network are indicated."

**37. Lines 221-225:** *"This should be included in a paragraph or table."* with reference to "The upslope areas of the 4 sensors installed are:

– 5.3 km2 for L1 gauging station,

– 12.6 km2 for L2 gauging station,

– 4.9 km2 for L3 gauging station,

– 18 km2 for L4 gauging station."

Your observation is noted; thank you. We have included this in a paragraph.

**38. Line 226:** *"Which photogrammetric flight do you mean (date)?"* with reference to "the photogrammetric flight"

Thanks for your comment. We have changed the text in line 226 as follows: "Since the area covered by the photogrammetric flights (2020 and 2021) excluded a portion of the upstream area of L1 and L3 gauging stations, these areas were determined using the 2008 DSM of Aosta Valley (SCT Geoportale, Regione autonoma Valle d'Aosta).

**39. Lines 267-268:** *"At this stage, and with inadequately described methods of obtaining 3D data, this statement cannot be supported. There are several factors that contribute to the accuracy of the 3D data that need to be described and accounted for. For instance, could you provide an overview of the photogrammetric processing used, including the number of GCPs utilised, the RMSE of the resulting model, and a report on the procedure undertaken?"* with reference to "The use of a Rover Base System and the presence of measured markers enabled the extraction of 3D data with centimetric accuracy (including the vertical component)."

Thanks for highlighting this aspect. Section 3.1 has been edited following Reviewers' comments, including a table with the details of planimetric and altimetric residuals on GCP and CP for all the dataset.

**40. Figure 8:** *"Please improve the colour scale in Figure 8a. Additionally, there appears to be no discernible change between 2020-2021 in Figure 8b."*

Your observation is noted; thank you. We have modified Figure 8a (Figure 9a in the current version) introducing a bivariate scale and the Limit of Detection (95%) thresholds. As far as figure 8b is concerned, changes are not visible due to the vertical axis scale: the larger scale profiles in figure 8c over a limited portion of the section is aimed at highlighting variations in the most relevant area. The updated figure can be seen in the 10$^{th}$ reply to Reviewer 1.

**41. Lines 289-293:** *"Before concluding this, more information is needed regarding the errors of each DSM and the corresponding limit of detection."* with reference to "Additionally, a comparison with the 2008 DSM shows a lowering of glacier surface up to 50 meters in glacial front areas. As far as very high-resolution satellite stereo pairs are concerned, they enable the extraction of 3D information with a lower vertical accuracy (metric level) with respect to aerial and drone data. Nevertheless, the coverage of a much larger area (in the range of hundreds of square kilometres) enables a multiscale and multiplatform approach to identify the most critical areas where to focus the monitoring activities in the field (Macelloni et al., 2022; Giulio Tonolo et al., 2020)."

Thanks for your comment. The text in Section 3.1 has been integrated with a discussion on the suggested LoD approach at line 285: "When comparing two DSMs (i.e. Difference of DSMs, DoD) it's crucial to distinguish the information (actual vertical displacement) from the noise. To this purpose, the Limit of Detection approach has been adopted. The vertical error of the two DSMs propagates when calculating their difference. From the RMS of the DSM, it is possible to calculate the RMS of the difference exploiting the error propagation theory. The vertical precision (based on CPs) of the DSMs is 11.1 cm for 2021 and 16.7 cm 2020. The DoD LoD at 68% confidence level is 20 cm ($\sqrt{11.1^2 + 16.7^2}$ ). Figure 8 (a) shows the differences between the 2021 and 2020 DSMs adopting a LoD threshold of 95% = 40 cm."

**42. Line 301:** *"which software and method did you used for interpolated the bathymetric data?"* with reference to "interpolated to produce a bathymetry map".

Thanks for your question. We have explained it in line 300 as follows: "The x-y-z locations of the first interface, representing the lake bottom, detected in all the GPR sections, were interpolated with a linear triangulation-based method (griddata function of MATLAB) to produce a bathymetry map."

**43. Line 307:** *"How did you arrive at the identification of glacial till at this location? It is important to note that till is a strictly glacial sediment with genetic implications."* with reference to "possibly a heterogeneous glacial till.".

Thank you for the report, we do not have sufficient evidence to state that the sediment could be a heterogeneous glacial till. Therefore, we have removed the sentence.

**44. Lines 377-378:** *"Please provide all the orthophotos and DSMs used. Aerial, drone and satellite images."* with reference to "The orthophotos and DSMs database related to the 2020 aerial survey is available on the Zenodo repository at https: //zenodo.org/records/8089499 (Corte et al. (2023d));".

We appreciate you bringing this to our attention. The 2020 aerial and 2021 UAV orthophotos and DSMs have been added as Zenodo repository. Satellite images have not been addressed in the manuscript.

**45. Lines 379-380:** *"Please, also provide the outlines of the glacier from Figure 1."* with reference to "The orthophotos and DSMs database related to the 2021 aerial survey is available on the Zenodo repository at https://doi.org/10.5281/zenodo.7713299 (Corte et al. (2023b))".

Your observation is noted; thank you. We have updated the dataset with the outlines of the glacier.

---

## Author Response (AR2)

Responses to Reviewers' Comments for Manuscript essd-2023-94

**Multitemporal characterisation of a proglacial system: a multidisciplinary approach**

Addressed Comments for Publication to

Earth System Science Data

by

Elisabetta Corte, Andrea Ajmar, Carlo Camporeale, Alberto Cina, Velio Coviello , Fabio Giulio Tonolo, Alberto Godio, Myrta Maria Macelloni, Stefania Tamea, and Andrea Vergnano

Dear Dr. James Thornton,

We extend our sincerest gratitude to the Editor and the esteemed Reviewers for dedicating their time and expertise to assess the revised version of the manuscript, "Multitemporal characterisation of a proglacial system: a multidisciplinary approach" with manuscript number essd-2023-94. Your insightful comments and constructive feedback have enriched the quality and depth of our work. We are grateful for the opportunity to undertake another revision. The Editor willingness to facilitate further improvements are genuinely appreciated. We approach this revision with enthusiasm and commitment to address all the concerns raised, striving to enhance the clarity, coherence, and impact of our work.

A summary of main modifications and a detailed point-by-point response to the comments from the Editor and Reviewers 1 and 2 are given below.

Sincerely,

Elisabetta Corte, Andrea Ajmar, Carlo Camporeale, Alberto Cina, Velio Coviello , Fabio Giulio Tonolo, Alberto Godio, Myrta Maria Macelloni, Stefania Tamea, and Andrea Vergnano

**Note:** To enhance the legibility of this response letter, all the editor's and reviewers' comments are typeset in boxes.

**Authors' Response to the Editor**

> **General Comments.** The revised manuscript has now been re-reviewed by the original referees. Whilst Reviewer #1 is now largely satisfied that the manuscript is suitable for publication, Reviewer #2 still has major concerns about the lack of a dedicated section on data quality assurance / evaluation / uncertainty (especially with regards to the drone data). Therefore, in your next revision, please kindly ensure that thorough details and explanations of the methods are provided throughout, and that a section regarding the quality assurance / evaluation of the UAV data in particular (and other data, as necessary) is included, supported by appropriate and rigorous statistics.

**Response:** We appreciate the opportunity to address the concerns raised.

In our latest revision, we have diligently incorporated dedicated sections focusing on data quality evaluation, and uncertainty, targeting the aerial data. A thorough methodological section based on a rigorous statistical analysis has been included in section 2.3 (and in a new Annex), while the related detailed results have been integrated and thoroughly discussed in section 3.2 to demonstrate the robustness of our findings. We are committed to ensuring the strength and integrity of our manuscript, and we hope that these revisions meet the expectations outlined. Once again, we appreciate the constructive feedback and the opportunity to enhance our work.

**Authors' Response to Reviewer 1**

> **General Comments.** I carefully read the authors' response to my comments and looked at the manuscript with the tracked changes. The authors have made a great effort to respond to my comments, and the article has benefited greatly from this work. As I pointed out in my previous paper, I consider this work a very good contribution and it demonstrates the importance of studying the functioning of proglacial margins with a multidisciplinary approach. The results presented here will allow further investigation of the Rutor Glacier and its proglacial margin, and the papers that follow will certainly be of interest to the geomorphological community.

**Response:** We are grateful for the extremely constructive comments and suggestions provided by Reviewer 1.

The authors agree the previous version of the manuscript benefited greatly from the reviewers' comments, in terms of paper structure, correction of errors, and integration of additional information to streamline the understanding from the reader perspective.

> **General Comments.** The paper is now stronger, more informative, and better structured. However, I still have some concerns about the error assessment in the DSMs, but it may be that I am too picky.

**Response:** Thank you again for your thorough review of our revised manuscript and for your constructive feedback.

Regarding your concerns about the error assessment in the DSMs, we truly appreciate your diligence in scrutinizing this aspect of our work. We anticipate that a new section focused on the assessment of the error in elevation differences (referred to as DoD in the manuscript) has been added, according to requests from the reviewers and editor.

We are confident that this new version should clarify the quality assessment of the data, especially in terms of evaluation of the DoD (precision and possible systematic errors) and its LoD.

> **General Comments.** The authors present a first DSM assessment in Table 4. They calculate the vertical and horizontal RMS of the GCPs and checkpoints, but it would have been more informative to include the mean error and the standard deviation of error as well (this would allow characterizing systematic and random errors in the DSMs).

**Response:** Thank you for the comment.

Table 4 (table 5 in the revised manuscript) have been revised accordingly, detailing the mean values and the standard deviations (for the $\Delta X$, $\Delta Y$, $\Delta Z$ residuals) for both GCP and CP. The outcome of the analysis suggests the absence of meaningful systematic errors.

> **General Comments.** Then, the authors calculate such statistics in Table 5 by using the delta of Z of a number of stable points, however these results are not informative since those stable points are not well scattered across the study zone.

**Response:** Thank you for the comment.

The area was selected since it is the only large area the authors are confident is stable in the two dates, being the rocky areas around the glacier partially covered by snow. Nevertheless, i) the area has been revised and divided in three smaller polygons to exclude fluvial areas and water bodies and ii) as suggested below by the reviewer, these stable areas have been used only to showcase a possible alternative approach to estimate the LoD of DoD (while the LoD based on CP has been used in Figure 9).

**General Comments.** By looking at the DoD in Figure 9a, it appears that the DSMs are of good quality, and no apparent systematic deformation is occurring. However, without a deeper error assessment some doubts might arise. Building on my own experience in error assessment in DSMs and DoDs, I would suggest two paths to assess the DSMs.

1. Assess the DSMs by using your independent checkpoints, which are widely distributed across the study area, although their number is low (but this is OK in such extreme and dangerous environments). This analysis would inform if your DSMs are impacted by random errors (std of error) and systematic errors (mean error). In the presence of systematic errors, check if those errors have some spatial structure (e.g., systematic tilt, which I formerly named datum shift), and if any correct it in respect to your checkpoint network (if possible). Once the presence of systematic errors is assessed (and corrected), use the std of error to calculate your Limit of Detection (LoD), either at 68 or 95 confidence limits. In this way, you do not violate the assumption of the error propagation theory which states that the errors in your DSM should be random, independent and gaussian. I suspect that this approach would provide a large LoD, but this is fine.

**Response:** Thank you for the comment.

We confirm that we adopted this approach, which is now clearly explained in the new ad-hoc section 2.3.1 "DSM Validiton: DoD and LoD estimation".

**General Comments.** 2. Assess your reference DSM (i.e., 2021) by using your independent checkpoints, and check for systematic errors. In the presence of systematic errors, check if those errors have some spatial structure (e.g., systematic tilt), and if any correct it in respect to your checkpoint network (if possible). This would inform about the quality of your reference. Then create a new set of checkpoints based on a number of widely distributed stable points (e.g., bed-rock, etc.). Calculate the shift between the reference (2021) and the DSM of 2020, and assess the presence of random and systematic errors. In the presence of systematic deformations such as tilt, correct them. Once the presence of systematic errors is assessed (and corrected), use the std of error of your stable points to calculate the LoD. Both ways are widely accepted, although (1) is probably more robust although your checkpoint population is somehow little. I would suggest discussing this with the editors, and get to know what their expectations in terms of error assessment are. I want to emphasize that my suggestions should not be seen as criticism, but as genuine suggestions to improve the quality of the article, which - as I have already said - in my opinion is already good. I am of the opinion that the authors can respond to my concerns fairly quickly and I don't think another round of review by the referees is needed.

**Response:** We truly appreciate Reviewer 1's comment, the extremely detailed feedback and the constructive suggestions.

We confirm that they have been exploited to improve the quality of the manuscript. It was an opportunity to further discuss among the authors pros and cons of the two different approaches that can be used to estimate the LoD of DoD. We are confident that this updated version should clarify the quality assessment approaches in terms of evaluation of the DoD (precision and possible systematic errors) and its LoD.

> **General Comments.** I know the challenges of collecting (and processing) this amount of data in these extreme environments, therefore I compliment the authors for having produced and discussed this remarkable dataset.

**Response:** Thank you for your acknowledgment of the challenges associated with collecting and processing data in extreme environments. We greatly appreciate your recognition of our efforts in producing and discussing the dataset presented in our manuscript.

> ### Comment 1
> Line 192: "During the 2021 field activities, a total of 32 artificial photogrammetric markers"
> Following Table 2, you deployed 33 markers. Either the text or the table are wrong.

**Response:** Many thanks for highlighting this source of misunderstanding.

The number of markers has been verified to be 32, which is the precise count. However, the total number of ground control points (GCPs) and checkpoint points (CPs) utilized for images acquired in 2021 amounts to 34. This discrepancy arises because two markers used for the aerial dataset were also leveraged for the UAV dataset, as elaborated upon in the manuscript.

**Comment 2**

Lines 223 – 225: "Outwash plains, which may have been affected by geomorphological changes (e.g. due to erosion and water deposits) between the time of the surveys, were also considered stable areas."

I do not see the point of using such points if you want to assess the error in your datasets. Outwash plains are extremely unstable as you pointed out. What is the rationale of choosing points within the outwash plain? You bring in this point later in the paragraph, stating that using such points may have worsen your statistics, which again questions the rationale of choosing them. Could you please clarify? How many stable points did you select?

**Response:** Thank you for pointing out this aspect.

Even if the approach based on CPs is used to estimate the LoD, we do agree with the comment. The stable areas have been refined accordingly, excluding the outwash plains and the water surfaces. Due to the steep slopes and the snow cover, especially for the year 2020, it was not possible to add other stable areas around the glacier (the main reason why the approach based on CP has been used).

> ### Comment 3
>
> Lines 352 – 354: "A standard Structure-from-Motion (SfM) photogrammetric approach was adopted, following a consolidated workflow (i.e. interior and exterior orientation, camera calibration, dense point cloud generation, DSM and orthomosaic generation) using the software Agistoft Metashpe."
>
> I would move this sentence to methods. I am a bit picky here, but could you please clarify what consolidated workflow means. I guess you mean the estimation of the internal/external orientation parameters through the alignment in Metashape, then markers, re-fitting through a bundle adjustment (camera calibration), and so on. Is this the workflow? Perhaps, it would be useful for future readers to specify the parameters you used within the bundle adjustment (e.g., focal length, principal point offset, etc.).

**Response:** Thank you for your valuable suggestions.

The sentence has been moved to the method section as suggested. As for the consolidated workflow, we have further specified it as follows:

The dataset acquired during the UAV and aerial photogrammetric flights were processed to obtain a 3D model of the terrain and additional cartographic products, i.e. orthophotos and DSMs. A standard Structure-from-Motion (SfM) photogrammetric approach was adopted, following a consolidated workflow using the software Agistoft Metashape, i.e.:

- Image alignment, to estimate interior/exterior (relative) orientation parameters, generating a relative sparse point cloud using feature detection and matching and SfM-based bundle block adjustment with self-calibration.

- Collimation of Ground Control Points (GCPs, not relevant in case of a direct georefencing approach), to re-estimate interior/exterior (absolute) orientation parameters refining the SfM-based bundle block adjustment with self-calibration, to generate a georeferenced sparse point cloud.

- Evaluation of residuals on GPC and Check Point (CP) and iteration of the previous step in case of anomalies in the residuals.

- Generation of a dense point cloud.

- Generation of a Digital Surface Model (with respect to a cartographic plane) and Orthoimagery.

Considering it was only a suggestion, we eventually opted to not include the specific estimated parameters for each photogrammetric dataset. This decision was based on the fact that we already provided several technical details regarding the photogrammetric flights and their general processing. Rather than detailing a long list of the aforementioned estimated parameters, which is not common even in Geomatics manuscripts, we focused, as suggested, on the assessment of the elevation datasets.

**Comment 4**

Table 4: See my general comments about including the mean error and std of error.

**Response:** Thank you for the comment.

We confirm that Table 4 (Table 5 in the revised manuscript) has been revised accordingly, detailing the mean values and the standard deviations (for the $\Delta X$, $\Delta Y$, $\Delta Z$ residuals) for both GCP and CP.

**Comment 5**

Line 365: "about 0.2 m"

Consider using 20 cm instead of 0.2 m.

**Response:** Thank you for your suggestion.

The text has been changed to '20 cm' instead of '0.2 m' as recommended.

> **Comment 6**
>
> Lines 377 – 379: "Glacier surface elevation differences were estimated by subtracting 2021 DSM to 2020 one, to quantify ablation and displacement (Table 5)"
>
> I suspect that the reference to Table 5 is wrong here.

**Response:** Thank you for the comment.

Our apologies for the typo. You are correct, the reference should be to Figure 9. The text has been updated accordingly to reflect this correction.

> **Comment 7**
>
> Lines 380 – 385: The introduction of a LoD is indeed very good, however there is no reference to the works of Brasington et al. (2000) and Lane et al. (2003), both published in ESPL, that are the foundation of the LoD theory in DoDs (in my opinion). Furthermore, you use the RMS of Table 4 that was previously noted as accuracy and here as precision. When calculating the LoD, one should use precision i.e. the standard deviation of error as illustrated in both Brasington et al. (2000) and Lane et al. (2003) papers.

**Response:** Thanks for referencing these two important manuscripts related to considering the LoD when performing DoD.

They are properly referenced in the current version of the paper as follows:

[1]  S. N. Lane, R. M. Westaway, and D. Murray Hicks, "Estimation of erosion and deposition volumes in a large, gravel-bed, braided river using synoptic remote sensing," *Earth Surf. Process. Landf.*, vol. 28, no. 3, pp. 249–271, 2003. DOI: `https://doi.org/10.1002/esp.483`. eprint: `https://onlinelibrary.wiley.com/doi/pdf/10.1002/esp.483`.

[2]   J. Brasington, B. T. Rumsby, and R. A. McVey, "Monitoring and modelling mor-
      phological change in a braided gravel-bed river using high resolution gps-based
      survey," *Earth Surf. Process. Landf.*, vol. 25, no. 9, pp. 973–990, 2000. DOI: `https:`
      `//doi.org/10.1002/1096-9837(200008)25:9<973::AID-ESP111>3.0.CO;2-Y`.
      eprint: `https://onlinelibrary.wiley.com/doi/pdf/10.1002/1096-9837%`
      `28200008%2925%3A9%3C973%3A%3AAID-ESP111%3E3.0.CO%3B2-Y`.

We also confirm that the terms accuracy and precision are now adopted consistently
throughout the paper.
* * *
**Comment 8**

Table 5: I appreciate the choice of including Table 5 in the results, however I
think that the way it is presented now does not provide an informative assessment
since the stable points cover a fraction of the study area and they are not widely
distributed across the study zone. I acknowledge the difficulty of selecting stable
points in such unstable environments, but you might find stable zones even in close
proximity of the glacier (e.g., bed rock) so that your assessment becomes more
informative. Please refer to my general comments at the beginning of this report,
and if possible calculate the statistics again to leave no room for any doubt.
* * *
**Response:** Thank you for the comment.

We confirm that the stable areas have been refined (three different polygons have been
identified), excluding the outwash plains and the water surfaces. Table 5 (Table 6 in the
revised manuscript) was obviously revised accordingly.

Due to the steep slopes and the snow cover, especially for the year 2020, unfortunately it
was not possible to add other stable areas around the glacier. Therefore, this analysis
was only used to showcase an alternative approach to the one based on CP to estimate
the LoD of DoD.

**Comment 9**

Line 386: "Figure 9 (a) shows the differences between the 2021 and 2020 DSMs adopting a LoD threshold of $95\% = 40$ cm" Since you used the 95% LoD in your DoD (Figure 9a) I would remove the description of the 68%. I do not see any point of including the calculation of the 68% if you do not use it in further analysis.

**Response:** Thank you for the comment.

The reference to 68% is now excluded from Figure 9. It is still used in the theoretical description of the approach and while commenting on the results to clarify how the 95% LoD is calculated.

**Comment 10**

Figure 9a: The DoD is now clearer with the bivariate scale and the LoD. I would suggest using the color white for changes that fall within the LoD or even use the orthomosaic as background (with LoD range in transparent), so that significant changes are easier to see. Furthermore, consider using more classes to be more informative, and perhaps use better contrasting colors. I would also suggest reversing the scale, since now it shows that the glacier termini gained in Z between 2020 and 2021. This results from subtracting the DSM of 2021 from the DSM of 2020 instead of doing the opposite, that is subtracting 2020 from 2021.

**Response:** Thanks for all the suggestions.

Figure 9 has been thoroughly revised accordingly and we do agree that it is now more informative and easier to interpret (Figure 1).

[Figure]

Figure 1: (a) 2021-2020 DoD. The white lines refer to the cross-sections A-A', B-B', and C-C', whose 2020 (red) and 2021 (blue) elevation profiles are shown in panels (b), (c), and (d) respectively.

**Authors' Response to Reviewer 2**

> **General Comments.** Thank you for giving me the opportunity to review this paper for the second time. I would like to thank the authors because this manuscript has been improved compared to the first version submitted.

**Response:** We appreciate the reviewer recognised the effort

> **General Comments.** However, even though more details and explanations have been proposed, I think there are still some major methodological weaknesses (especially on the UAV-derived datasets) and underlying problems in this work that, from my point of view, could question the acceptance of the manuscript.

**Response:** Thank you for the comment.

We are confident that this new version will clarify the concerns that have been raised. Nevertheless, it's important to clarify that our primary focus is on analysing the photogrammetric datasets based on the crewed aerial surveys rather than the uncrewed aerial vehicles (UAV) surveys. The DoD analysis has been carried out on the aerial datasets since the UAV surveys cover only a small portion of the glacier and were conducted solely in 2021, which doesn't align with our multitemporal comparison of 2021 versus 2020. We hope this clarification addresses any concerns you may have had.

**General Comments.** The manuscript lacks a comprehensive quality assessment section (after the results), especially for the UAV data. This issue was raised during the first round of review, but no real attempt was made by the authors to address it. Without this, and without validation of the data obtained, the usefulness of such a rich dataset is not assured. The quality of the results is also highly dependent on the data acquisition and therefore requires proper planning and description. For example, image overlap, which is a fundamental factor in any photogrammetric approach, is not even mentioned (and this is just one example).

**Response:** Thank you for the feedback

We have introduced a comprehensive section (2.3.1) dedicated to quality assessment, encompassing both theoretical aspects applicable to UAV and crewed aerial surveys and a results section primarily (3.1) centred on crewed aerial DSM (for the aforementioned reasons). Our emphasis lies in evaluating the quality of Difference of DSMs (DoD), with a specific focus on introducing, discussing, and applying the concept of Limit of Detection (LoD) of DoD.

Additionally, it is important to emphasize that this manuscript adopts a multidisciplinary approach, extending beyond the realm of Geomatics. We have included detailed planning activities related to the UAV surveys to provide a comprehensive understanding. The focus, however, remains on the rigorously assessed final products, as previously mentioned.

**General Comments.** If the authors want to resubmit their paper, they will of course have to be more careful about the statistics used, especially when quantifying precision, accuracy, systematic errors in general, systematic errors in UVA-based models according to photogrammetric reasoning. More and clearer statements are needed, backed up by solid statistics and numbers. For example, I do not believe in the elevation changes obtained. The LoD analysis (map visualization) is not achieved (Figure 9).

**Response:** Thank you for the comment.

We are confident that the current revision is properly addressing all the aforementioned topics.

---

## Author Response (AR3)

**Multitemporal characterisation of a proglacial system: a multidisciplinary approach**

Addressed Comments for Publication to

Earth System Science Data

by

Elisabetta Corte, Andrea Ajmar, Carlo Camporeale, Alberto Cina, Velio Coviello , Fabio Giulio Tonolo, Alberto Godio, Myrta Maria Macelloni, Stefania Tamea, and Andrea Vergnano

**Note:** To enhance the legibility of this response letter, all the editor's and reviewer's comments are typeset in boxes.

**Authors' Response to the Editor**

> **General Comments.** Dear Authors,
>
> I have now obtained one further constructive review report. Please kindly address all points raised and submit a revised version of your manuscript.
>
> I will then be able to take a final decision.
>
> Best regards,
>
> James Thornton

**Response:**

Dear Dr. James Thornton,

Thank you for providing us with the additional constructive review report. We appreciate the opportunity to address the points raised and will promptly work on revising our manuscript accordingly.

Best regards,

The Authors

**Authors' Response to Reviewer**

> **General Comments.** Dear Authors, Thank you for giving me the chance to read and review your paper and data. I think it should be published as it provides valuable information that can help us better understand how proglacial areas work.

**Response:**

Dear reviewer, thanks for the time dedicated to reading, reviewing and sharing constructive comments and suggestions on the manuscript and underlying data. The author's team is extremely satisfied by reading that you deem the manuscript is worth publishing since it can support the understanding of proglacial areas.

> **General Comments.** The dataset consists of airborne orthomosaics and DEMs that cover glaciers and the proglacial area, along with other data from the area like hydrometric, bathymetric, and seismic data. The only downside is that the dataset only includes data from one year (2020-2021).

**Response:** Thank you for the comment.

We confirm that the current dataset is covering the period 2020-2021. Considering that the Glacier Lab of the Politecnico di Torino is still active, and therefore the Rutor Glacier will almost certainly still be monitored, in case of new dataset acquisitions, we will publish them as soon as processed and validated.

> **General Comments.** The authors improved their manuscript in response to previous reviews.

**Response:**

We deeply appreciate the recognition of the effort in revising the initial version and the structure of the manuscript to improve it, according to the comments and suggestions from the previous Reviewers.

**Comment 1**

However, I still have some suggestions related to SfM data processing and presentation of the results:

1. Please ensure that you report the parameters used in the Structure-from-Motion (SfM) processing, as well as the assessment of each stage of processing, either in the text or as an appendix. It is recommended that you follow the detailed guidelines provided for reporting results of SfM photogrammetry in geosciences James et al. (2019) [1]. Several publications have reported the influence of processing parameters in Structure-from-Motion software on the quality of DEMs and orthomosaics - they can significantly affect the accuracy of the final products, particularly the filtration and optimization stages. Therefore, it is essential to report all processing parameters.

**Response:** Many thanks for your suggestions

- We deem that the most efficient approach for sharing all the processing parameters in a transparent and detailed way is to add, in the supplementary material if the Editor agrees (considering they are two PDF documents of about 8 pages each), all the relevant processing reports of the SfM processing software, focusing on the crewed aerial flights used for the DoD analysis. Those reports provide the requested details (including tables, maps and, plots when relevant) about survey data, camera calibration, camera locations, ground control points and checkpoints, Digital Elevation Model and, the processing parameters. The following sentences were added: at line 161: "The photogrammetric processing reports generated by Metashape software for the two crewed aerial flights (the only ones

used for elevation analyses) are available in the supplementary material section (`https://doi.org/10.5281/zenodo.11144390`, **Corte2023_Geom4**). These reports include information on processing parameters settings, survey data details, camera calibration, camera locations, ground control points and check points"

- As far as the recommendation to follow the guidelines proposed by James et al (2019)[1], we confirm that they have been explicitly or implicitly followed and clearly discussed in the manuscript. Especially the considerations related to points 7 (Results - Error reporting), 9 (Results - control and independent check measurements), 10 (Split data tests), 11 (Management of systematic error), 12 (Residual uncertainty) are extensively presented and discussed in the current version of the manuscript according to the comments from previous reviewers (section 2.3.1, section 3.2 and Appendix B). Accordingly, the following sentence has been added at line 193 at the beginning of section 2.3.1: "To properly assess the photogrammetric results (i.e. in this specific case the DSMs and their differences generation), it is necessary to apply "suitable statistics to identify systematic error (bias) and to estimate precision" and to propagate "uncertainty estimates into the final data products" (James et al, 2019 [1])".
* * *
**Comment 2**

2. Following the previous comment, please consider adding raw (i.e. unprocessed) images to the described dataset. This would enable future users to process the images using different approaches and parameters, especially if new data becomes available or new processing methods arise.
* * *
**Response:** Thank you for the comment.

We understand the potential benefits of sharing raw data, such as crewed aerial images, despite, as mentioned in a previous response to the reviewers, the focus of this manuscript is on the rigorously assessed final products. Regrettably, the licensing terms for the

crewed aerial images do not allow their public re-sharing as open data. However, we remain open to consider ad-hoc access requests, which can be sent directly to the authors.
* * *
**Comment 3**

3. During the initial stage of SfM processing (photo alignment), consider co-aligning surveys from different years to reduce uncertainties in resultant DEMs. Please refer to the approach proposed by Cook and Dietze (2019)[2]; De Haas et al. (2021)[3], Nota et al. (2022)[4]. That could significantly reduce uncertainties in resultant DEMs, especially for the area lacking GCPs.
* * *
**Response:** Thank you for your valuable suggestions.

We confirm that the coregistration of the aerial survey was a priority, as explained in lines 168-171 and 181-184 of the manuscript. Nevertheless, we have not used the co-alignment approach proposed by Cook and Dietze (2019) [2] due to the changes that occurred between the two acquisitions and the limited extent of stable areas, as stated in lines 201 and 411 of the manuscript. This issue is explicitly discussed in the "Potential limitations" section of the above-mentioned paper: "If the appearance of the area changes too much between surveys or if too much of the area of interest has changed, sufficient tie points may not be generated, as described above. Therefore, well-distributed stable areas with a consistent appearance are required for successful alignment"

> **Comment 4**
>
> 4. In order to determine if there are any systematic errors on CP, it would be helpful to examine the spatial distribution of the errors. For instance, if most of the GCPs and CPs are located in the proglacial area, then it is likely that the uncertainties will be lower in this area compared to the upper part of the glacier. I suggest adding a map that displays errors (X, Y, Z) on CPs, so the reader can identify any spatial trends.

**Response:** Thank you for the comment.

As mentioned in the answer to comment number 1, we are going to include in the supplementary material, if the Editor agrees, the detailed photogrammetric processing reports, which include the figure "GCP locations and error estimates" with the spatial distribution of GCPs and CPs including the related error estimates represented by color-coded ellipses (where the altimetric error is represented by the ellipse color and the planimetric error components are represented by the ellipse shape and orientation). An example is reported in the figure below for easy reference.

> **Comment 5**
>
> 5. In the manuscript, you stated that LoD was set to 0.44 m; however, in the map (Figure 9), values between -0.5 and +0.5 m are indicated as no change (i.e. transparent). Please, correct it.

**Response:** Thank you for your comment.

The image has been re-generated accordingly (see below). Additionally, the sentence in line 418 of the manuscript has been changed as follows: "Notably, values falling within the range of -44 cm to +44 cm are omitted from figure 2(a)) , as they fall within the LoD for the DoD calculated based on CPs".

[Figure]

Fig. 3. GCP locations and error estimates.
Z error is represented by ellipse color. X,Y errors are represented by ellipse shape.
Estimated GCP locations are marked with a dot or crossing.

Figure 1: Figure included in the detailed photogrammetric processing reports: GCP locations and error estimates.
* * *
**Comment 6**

6. Page 20, l. 411 "this study has opted to use the DoD value derived from CP analysis, which stands at 44 cm" should probably read "LoD" not "DoD"
* * *
**Response:** Many thanks for spotting this typo.

The typo has been fixed as suggested.

[Figure]

Figure 2: (a) 2021-2020 DoD. The white lines refer to the cross-sections A-A', B-B', and C-C', whose 2020 (red) and 2021 (blue) elevation profiles are shown in panels (b), (c), and (d) respectively.

[1]  M. James, J. Chandler, A. Eltner, *et al.*, "Guidelines on the use of structure from motion photogrammetry in geomorphic research," *Earth Surf. Process. Landf.*, vol. 44, Apr. 2019. DOI: `10.1002/esp.4637`.

[2]  K. L. Cook and M. Dietze, "Short communication: A simple workflow for robust low-cost uav-derived change detection without ground control points," *ESurf*, vol. 7, no. 4, pp. 1009–1017, 2019. DOI: `10.5194/esurf-7-1009-2019`.

[3]  T. Haas, W. Nijland, B. McArdell, and M. Kalthof, "Case report: Optimization of topographic change detection with uav structure-from-motion photogrammetry through survey co-alignment," *Front. Remote Sens.*, vol. 2, p. 626 810, Feb. 2021. DOI: `10.3389/frsen.2021.626810`.

[4]  E. Nota, W. Nijland, and T. Haas, "Improving uav-sfm time-series accuracy by co-alignment and contributions of ground control or rtk positioning," *Int. J. Appl. Earth Obs. Geoinf.*, vol. 109, p. 102 772, May 2022. DOI: `10.1016/j.jag.2022.102772`.

---

## Author Response (AR4)

**Multitemporal characterisation of a proglacial system: a multidisciplinary approach**

Addressed Comments for Publication to

Earth System Science Data

by

Elisabetta Corte, Andrea Ajmar, Carlo Camporeale, Alberto Cina, Velio Coviello , Fabio Giulio Tonolo, Alberto Godio, Myrta Maria Macelloni, Stefania Tamea, and Andrea Vergnano

**Note:** To enhance the legibility of this response letter, all the editor's comments are typeset in boxes.

**Authors' Response to the Editor**

> **General Comments.** Dear authors, Thank you for responding to the comments of the third reviewer. I fully support your decision to add the processing parameters to the SI.
>
> In my view, the manuscript is now suitable for publication, pending your consideration of one minor point.
>
> Congratulations and best wishes,
>
> James Thornton

**Response:**

Dear Dr. James Thornton,

We express our deepest gratitude for generously dedicating your time and expertise to evaluate the revised version of our manuscript, "Multitemporal characterisation of a proglacial system: a multidisciplinary approach" with manuscript number essd-2023-94.

We sincerely appreciate the opportunity to conduct another revision. The insightful remarks and constructive feedback have enhanced the quality of our manuscript.

We are extremely grateful for your decision to accept our manuscript. We are honored and delighted by the positive feedback from the reviewers and your approval of our work.

Best regards,

The Authors

> **General Comments.** Dear authors,
>
> Regarding Comment #2 of the final reviewer, I appreciate that licensing terms may restrict you from sharing the raw data. However, since this is an important point for readers to be aware of (and to some extent affects the reproducibility of the work), I would ask you to consider a sentence to the manuscript explaining why the raw data cannot be shared.

**Response:** Thank you for your suggestion

We added in the Data availability section at line 516 the following sentence:

"The images used for the creation of the photogrammetric products used for the DoD analysis are not available in the datasets due to licensing restrictions. The ownership of the images is not exclusive to the authors but is shared with the organization responsible for their acquisition, Digisky. For this reason, it is not possible to share this dataset as well. To obtain the images, the purposes of the request need to be analyzed, and an agreement on their use must be signed directly with the authors and Digisky. We are open to sharing the dataset if it is agreed upon with Digisky."

> **General Comments.** I am also slightly confused about the last part of your response: "However, we remain open to consider ad-hoc access requests, which can be sent directly to the authors." I would expect that if the data cannot be shared openly due to the licence, then they could not be shared following an ad hoc request (e.g. email) either. Please could you kindly clarify?
>
> Many thanks for addressing this final point, and congratulations for your persistence once again.
>
> James Thornton

**Response:** Thank you for your request for clarification

We apologize for our incomplete response.

The images we used for the creation of the photogrammetric products are owned by us and Digisky, therefore the licensing terms for the crewed aerial images do not allow their public re-sharing as open data. Data sharing must be agreed upon between the image owners (us and Digisky) and the party wishing to use them. Consequently, to obtain the images, the purposes of the request need to be analyzed, and an agreement on their use must be signed directly with us and Digisky.

Our availability was about our openness regarding data sharing and thus to submitting any ad-hoc requests to Digisky and requesting the establishment of an agreement with a third party. However, unfortunately, we are currently not authorized in any way to share the images.

We apologize again for the lack of clarity in our previous response.